# A flexible device to produce a gas stream with precisely controlled water vapour mixing ratio and isotope composition based on microdrop dispensing technology

Harald Sodemann[1,2], Alena Dekhtyareva[1,2], Alvaro Fernandez[2,3,4], Andrew Seidl[1,2], and Jenny Maccali[2,3,5]

[1]Geophysical Institute, University of Bergen, Norway
[2]Bjerknes Centre for Climate Research, Bergen Norway
[3]Department of Earth Sciences, University of Bergen, Norway
[4]Andalusian Institute of Earth Science, CSIC-University of Granda, Spain
[5]SFF Centre for Early Sapiens Behaviour (SapienCE), University of Bergen, Norway

**Correspondence:** Harald Sodemann (harald.sodemann@uib.no)

**Abstract.** Here we describe a versatile device to produce a gas stream with precisely controlled water vapour mixing ratio and stable water isotope composition based on microdrop dispensing technology. To produce a moist air stream, the microdrop dispensing technology ejects micrometer-size water droplets that completely evaporate into a stream of carrier gas heated to 60 deg C. By precisely controlling the contribution of water standards from two dispenser heads into a carrier gas stream, the device allows to set the air-vapour stream to any isotope ratio between two end member waters. We show that Allan deviation of the water vapour mixing ratio is 10 ppmv over more than 24 h, and reaches 0.004 ‰ for $\delta^{18}$O and 0.02 ‰ for $\delta^2$H for a flow rate of 40 sccm. Tests with flow rates from 40–250 sccm shows an increase of the Allan deviation with higher flow rates. Tests with mixing standard water from two dispenser heads shows a linear mixing across a range of water vapour mixing ratios from 1'000 to 20'000 ppmv. In addition to presenting the design and several performance characteristics of the new system, we describe two application examples. First, we utilise the device to determine the water vapour mixing ratio – isotope ratio dependency, a common artefact of water vapour isotope spectrometers. Second, we utilise the device to provide a constant background stream of moist air for fluid inclusion water isotope analysis in calcite samples from stalagmites. The observed flexibility and precision of the device underpins its usefulness and potential for a wide range of applications in atmospheric water vapour isotope measurements. Future developments could focus on reducing the number of manual interventions needed to clear dispenser heads from gas bubbles, and the provision of a water vapour stream at flow rates of up to several liters per minute.

## 1 Introduction

The composition of stable water isotopes in ocean and surface waters, land ice, and precipitation has long been used to extract valuable information about the climate system (Horita et al., 2008; Galewsky et al., 2016). Since the advent of instruments based on Cavity-Ring Down Spectroscopy (CRDS) for water isotope analysis (Kerstel et al., 2006; Lee et al., 2005; Gupta

et al., 2009), water vapour can be measured continuously at a time scale of seconds, providing for example insight into the details of weather systems (e.g., Graf et al., 2019; Thurnherr et al., 2020). To match the precision and accuracy of Isotope-Ratio Mass Spectrometry (IRMS), laser-based isotope analysers need frequent calibration of the raw data to account for instrumental drift, and for the characterisation of spectroscopic baseline effects (Sturm and Knohl, 2010; Aemisegger et al., 2012). Currently, variability of the gas stream produced by the calibration system, both in terms of mixing ratio and isotope composition, together with inlet and instrument characteristics, are important contributors to the total uncertainty of atmospheric water vapour isotope measurements. Separating different contributions of uncertainty is in particular critical at low humidities, such as for airborne measurements (Sodemann et al., 2017) and in cold environments (Casado et al., 2016; Seidl et al., 2023). Correction functions for spectroscopic effects causing a dependency on both water vapour mixing ratio and isotope composition require a detailed characterisation of each analyser(e.g., Weng et al., 2020; Thurnherr et al., 2020). The necessary, precisely controlled gas stream over sufficiently long averaging times requires an external vaporisation device with flexibility in terms of mixing ratio and isotope composition, and flow rates.

The characteristics and properties of such an external vaporisation device are critical to obtain accurate and precise measurements of the water vapour isotope composition. A range of devices with distinct designs have so far been used to generate a vapour stream for instrument characterisation and calibration. The bubbler design consists of a dry air stream that is percolated through a sufficiently large water reservoir at a defined temperature. The emanating water vapour has then an isotope composition in isotopic equilibrium with the liquid for the given temperature (e.g., St Clair et al., 2008). Despite its overall simplicity, there are several drawbacks with bubbler designs. First, precise temperature control is required to predict the isotope composition of the water vapour from equilibrium fractionation. Additionally, a reservoir of up to several liters of water may be needed to limit the impact of drift from the changing isotope composition in the liquid over time. Depending on the measurement platform and ambient conditions, handling of such amounts of liquid may be a hindrance during field deployments.

Another design concept to circumvent complications from isotope fractionation involves the complete evaporation of liquid water into a heated cavity. Such systems are for example commonly used for liquid sample analysis, where an autosampler injects about $2\,\mu$l of water into a cavity heated to $110\,°C$ (e.g., van Geldern and Barth, 2012). Such systems only produce a vapour stream of about $10\,min$, which is sufficient for liquid sample analysis, but not sufficient for longer-term instrument characterisations. Another design that allows for generation of a continuous vapour stream with complete evaporation is the water vapour isotope standard source (WVISS, Los Gatos Research, USA) using droplets emanating from a nebulizer head. The WVISS allows for production of a vapour stream over several hours for one standard at a time. In addition to manual intervention for changing standards, the system requires external modifications to reach lower mixing ratios, and the achieved precision of the water vapour mixing ratio can be a limitation for some applications (Aemisegger et al., 2012).

A further design concept involves continuously injecting a small amount of standard water into a heated cavity by means of thin needles, such as used in the Standard Delivery Module (SDM, Picarro Inc, Sunnyvale, USA). Built as an external module added to the liquid water analysis setup of these analysers, the SDM provides a vapour stream at a typical flow of $35\,$sccm, supplied from water standard reservoirs by two syringe pumps. While overall a reliable field calibration device, the mechanically operated syringe pumps can be prone to failure of moving parts, and air bubbles in the lines can lead to strong

oscillations in the mixing ratio, requiring manual intervention. Both can be important limitations for calibration in remotely-operated measurement setups (Bonne et al., 2014). A further design for water vapour generation with complete evaporation that has been used for example in providing a constant background humidity in fluid inclusion analysis involves, and a peristaltic pump that delivers a water droplets into an $N_2$ gas stream heated to 140 °C (Affolter et al., 2014). A similar design has been

used in the continuous analysis of water from ice cores on a melting bed, where the liquid is provided through a capillary to a heated oven (Gkinis et al., 2010) or from a circular nebuliser that produces a spray of $1.5\,\mu$m-size droplets (Jones et al., 2017). Both designs have been operated over extended time ranges (e.g., Bonne et al., 2019), but have not been constructed for flexibility, such as to regulate the mixing ratio of water vapour in the gas flow over a wide range.

The precise control of mixing ratio is in particular important for the provision of a water vapour stream background humidity

in fluid inclusion analysis. The principle of fluid inclusion analysis relies on voids within the carbonate matrix of stalagmites that regularly preserve remnants of cave drip waters. Since these waters are the relics of past precipitation, their oxygen and hydrogen isotope compositions can be used to reconstruct past changes in cave temperature (e.g., McGarry et al., 2004; Meckler et al., 2015; Fernandez et al., 2023; Maccali et al., 2023) and changes in the amount and/or source of precipitation (Fleitmann et al., 2003). Due to the small size of inclusions (<100 $\mu$m) and the low amount of water that is commonly present

in stalagmites (0.05 to 0.5 weight%; Affolter et al. (2014)), isotope measurements are made on the water that is released when large aliquots (>50 mg) of carbonate are crushed in a heated apparatus. Due to the small water amounts, analysis of fluid inclusions in stalagmites requires a background vapour stream with precisely known isotope composition and mixing ratio as carrier gas. This known background enables to separate the superimposed $\delta^{18}$O and $\delta^2$H signals of the water released from calcite cavities during the crushing.

Another design concept with both, complete evaporation of standard water and fine-grained control of the mixing ratio is based on the dispensing of micrometer-size droplets (St Clair et al., 2008). Microdrop systems are essentially ink jet printer heads that eject droplets from a glass capillary at a defined size by means of piezoelectric stimulation. Thereby, the head is filled or emptied by pressure applied to the head space of the reservoir. During dispensing of droplets, the liquid is supplied to the printer head by capillary forces. The particular advantages of a microdrop system in the context of vapour stream generation

is the precise control of the amount of water released from the dispenser head by modification of voltage and frequency (Iannone et al., 2009). Release of individual water droplets at a defined size enables fast and complete evaporation without fractionation artefacts. Sayres et al. (2009) applied a bubbler and a microdrop system with a high flow rate of dry gas (100 slm) to achieve a stable flow at low humidity (<200 ppmv) needed for calibration of cavity-based absorption instrument used for upper troposphere and lower stratosphere measurements. Sturm and Knohl (2010) built a prototype water vapour generator

based on microdrop technology, that required manual change of standards, but the design has not been developed further for water vapour isotope analyses since. Therefore, there is currently no single system available that in a flexible way provides the combination of a precise stream of water vapour across a range of water vapour mixing ratios and isotope compositions as well as operation at various flow rates between 50 and 250 sccm.

Here we present a new, flexible and versatile calibration system for CRDS vapour isotope analysers based on microdrop

dispensing technology, which allows to produce a precise vapour stream over a range of mixing ratios and flow rates. As a key

innovation, we use two dispensing heads simultaneously, providing to our knowledge the first system that allows to generate a vapour stream with any isotope composition along the mixing line of two standards. Due to its flexibility, the device is suitable for a range of applications, including instrument characterisation, calibration of water vapour isotope measurements, and as a component in specific analytical setups, such as a crushing line for fluid inclusion isotope analysis in cave deposits (Affolter et al., 2014). In addition to presenting the design and performance characteristics of the new system, we describe here two different application examples of increasing complexity, namely the characterisation of the mixing ratio – isotope ratio dependency of CRDS analysers, and the provision of background humidity for stalagmite fluid inclusion isotope analysis.

## 2 Application requirements, design objectives, and specifications

Our primary design objective is to create a single, highly flexible device for the generation of a water vapour isotope stream that can be used in a variety of different applications with CRDS analysers. Application (i) is to provide a constant background humidity with precisely controlled mixing ratio $x$ and isotope composition in an analytical setup, specifically for fluid inclusion water isotope analysis. In this application, a vapour stream with precise mixing ratio (SD(30 min)<20 ppmv) at a set isotope composition should be provided for several hours to days. Application (ii) is to provide a vapour stream for calibration with flow rates varying between 35 and 250 sccm. This range of flow rates is required to produce the vapour stream required by analysers that run in different flow configurations, such as the standard low-flow configuration of Picarro analysers at 35 sccm, and the flux configuration with flow rates of about 150 sccm for aircraft measurements (Sodemann et al., 2017). At even larger flow of about 300 sccm through the cavity of the CRDS analyzer, flow rate limitations of calibration systems emerge as an important challenge (Thurnherr et al., 2020; Bailey et al., 2023). Even larger flow rates that would also allow to characterise the response times of entire inlet lines with additional flush pumps used in semi-permanent installations for water isotope analysis (e.g., Steen-Larsen et al., 2013; Bonne et al., 2014; Galewsky et al., 2016) would require downstream dilution with a dry carrier gas. We leave such larger flow rates to future applications. Application (iii) is to providing a wide range of isotope composition and mixing ratios for instrument characterisation. Characterisation of analysers in terms of their mixing ratio – isotope ratio dependency requires to independently step through mixing ratio and isotope composition (Weng et al., 2020). This is facilitated greatly by an on-line, adjustable mixing of the evaporated water between two end-member standards. To cover the typical ambient mixing ratios encountered in mid-to high latitudes from the surface to high-elevations, the range of mixing ratios should encompass at least 500 to 25000 ppmv. Furthermore, the entire system should be small, portable and robust enough to be part of a field installation, for example to set up in a 19 inch rack mount alongside the analyser on a ship or some measurement station.

These application requirements are in their combination not met by any single existing calibration system, and would require either a combination or modification to currently available systems. In particular the ability to set the isotope composition within a range is not available from current devices. For example, the SDM provides a more limited range of humidities (6'000–24'000 ppmv according to manufacturer specifications, even though lower humidities are possible), and at most 20 min of operation before a new cycle is started. The SDM is also limited to two standard waters and allows no easy exchange or

mixing of different standards. To some degree, autosampler injections can be used to cover a range of isotope composition over
different mixing ratios. However, this involves significant manual intervention and preparations, as different mixtures between
water standards have to be prepared and analysed beforehand, and injection amounts have to be adjusted to obtain the desired
range of water vapour mixing ratios in the vapour stream from the vapouriser module (Weng et al., 2020). The availability
of one flexible device for several applications holds the promise of simplifying workflows, reducing error sources, and thus
contribute to better data quality.

In summary, we chose the following specifications of the water vapour generator:

1. To produce a water vapour stream with mixing ratios between 500 and 25000 ppmv;

2. To provide gas flow rates of between 35 and 250 sccm;

3. To maintain a precicely regulated, drift-free water vapour stream over hours to days;

4. To allow for continuous mixing of the isotope composition between two end members;

5. To be portable enough to be setup with a CRDS analyser during a field installation.

## 3  Device components and setup

At its core, the calibration device consists of a vertical evaporation chamber where the vapour stream is generated (Fig. 1).
The evaporation chamber is a custom modification of a 300 ml sample cylinder (Swagelok Inc., USA, Part No. 304L-HDF4-
300-T) with 1/4 inch NPT threads on both ends (Fig. A1). Two stainless steal tubes of 30 mm length and 10.1 mm inner
diameter have been welded horizontally to the upper part of the evaporation chamber, facing each other at an angle of 60 °C
and with a vertical offset of 10 mm (Fig. A1b). The two tubes reach 3 mm into the interior of the chamber, and each holds a
dispenser head (Microdrop GmbH, Germany, Part Nr. MK-K-130-020) with an inner nozzle diameter of $50\mu$m for liquids with
a viscosity below 20 mPas. For our prototype design, we chose an unheated head with a medium-size nozzle, similar to the
design of St Clair et al. (2008). Each dispenser head (DH) is connected to one 12 ml glass vial that holds a liquid water standard
and is mounted next to the evaporation chamber. Both DHs and the standard vials are connected to a control device that among
other parameters controls the piezo-electric pulse voltage and duration, the pulse frequency, and headspace pressure for both
DHs. In order to reduce memory effects from the retention of water vapour on the walls inside the evaporation chamber, the
assembly has been treated with a hydrophobic coating (SilcoNert, SilcoTec Inc., USA).

Dry air (synthetic air) or $N_2$ is introduced as a carrier gas at the lower end of the evaporation chamber (Fig. 1). The carrier
gas is heated during passage of 60 cm of 1/4 inch SS tubing that bends between two brass plates that are heated on the outside
by heat trace covered with metal mesh (Watcom Inc, USA). The heat trace wraps around both the gas heating assembly and the
evaporation chamber, and is controlled to 60 °C. Temperatures near the DHs are monitored to remain below 60 °C to minimize
evaporation of liquid directly from the DH, which could possibly interrupt the dispensing.

A manual valve provides either $N_2$ or dry air from a gas tank or other source to the calibration unit. At the entry into the heated tubing, a mass flow controller (GFC 17, Alborgh Inc, USA) regulates the flow rate electronically to a set value between 0 and 500 sccm. The experiments presented here were either performed with high-purity grade $N_2$ (Nitrogen 5.0, purity >99.999 %; Praxair Norge AS, x<5 ppmv), synthetic air (synthetic air 5.5, purity 99.9995 %; Praxair Norge AS, x<5 ppmv), or air sourced from a dry air generator with added drying cartridges (MT-400, VWR, USA, m<100 ppmv). Typically, the carrier gas was supplied with a pressure of <1 bar upstream of the mass flow controller.

At the upper outlet of the evaporation chamber, different applications tap the vapour stream from a 1/4 inch SS connection with Swagelok compression fitting. Depending on the application, the vapour stream was directly provided to the CRDS analyser, or led into an analytical assembly upstream of the analyser. Typically, an open split allowed for excess vapour to be vented into the room, with a line leading to the Wave-Length Monitor (WLM) port of the Picarro analysers.

A custom software written in python running on the Picarro analyser controls the flow rate of the mass-flow controller, the dispenser heads, and temperatures via a digital/analog interface (U12, LabJack Inc., USA). The software allows to manually fill, empty and modify settings for each DH, including dispensing frequency, and to step through an automated sequence with a specific frequency at which each dispenser head injects droplets into the evaporation chamber.

## 4 Operating principle and procedures

The operating principle of the microdrop calibration device is based on the precisely controlled injection of liquid water droplets into the evaporation chamber. The number of droplets at a specific size ejected into the evaporation chamber, which is flushed at a specific flow rate, results in a moist air stream with a water vapour and isotope ratio adjusted to the desired application.

### 4.1 Droplet generation and droplet evaporation

The microdrop dispensing technology with the DHs used here allows to generate droplets with diameters ranging between 35.0 to 90.0 $\mu$m in diameter, depending on fluid properties (Table 2). The droplets can be ejected from the nozzle into the evaporation chamber with frequencies between 1 and 1500 Hz. As the droplets are ejected with about $2\,\mathrm{ms}^{-1}$ into the heated dry carrier gas flow, they rapidly evaporate before reaching the bottom of the evaporation chamber. The default factory calibration parameters can be modified to obtain different drop sizes. Thereby, the drop size range is more limited than the theoretically available range of drop sizes, as it depends on the nozzle diameter, fluid viscosity, and piezoelectric pulse voltage and duration parameters. We observed that, in line with manufacturer specifications, only at specific combinations of settings for the piezo-electric pulses (defined by voltage and duration) a single droplet, rather than a jet or sequence of drops was ejected. Higher voltages thereby tended to provide a droplet stream that operated over longer times before it could stop randomly, typically to the build-up of gas bubbles. The piezo-electric parameters needed to be determined for each DH in a relatively time-consuming procedure using a separate setup with a high-speed camera. According to manufacturer information, unless when operating at much higher voltages, the piezo-electric characteristics of each DH are thereby expected to be constant over time. Figure 2 gives examples for well-formed (panel a) and jet-like (panel b) droplet generation. Drop-like ejection is important for the linearity

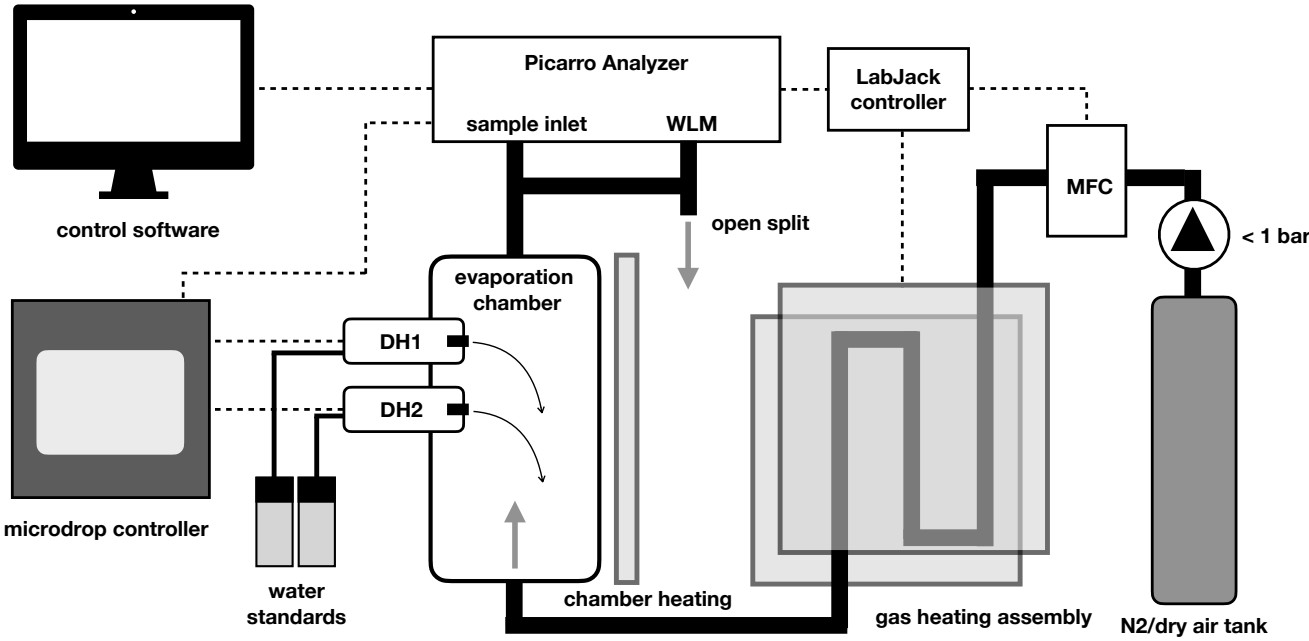

**Figure 1.** Schematic of the microdrop calibration device components. Thick black lines indicate tubing for gas flow. Dashed lines indicate electrical signal flow. Transparent grey areas denote heating elements. For further details see text.

of the DH performance across frequencies and flow rates (Sec. 4.2). After suitable settings for the drop size had been found, the piezo-electric parameters were not modified further. When a decrease of the DH performance or efficiency was observed, this was in most cases due to the formation of gas bubbles which could be removed by repeated fill/empty cycles. The typical and theoretical limits for the specified rate of flow rates at the given drop size of about $70\,\mu$m are given in Table 1. With simultaneous operation of two DHs, a duplication of the mixing ratio is possible.

## 4.2 Linearity and effective drop size

Accurate control of the size of drops ejected from each DH is an important basis for mixing between water standards later on. As each dispensed drop adds a defined volume of water to the airstream, an ideal DH would provide a linear correspondence between the dispensing frequency and the mixing ratio of the resulting vapour stream. We can use the relation between flow rate, frequency and mixing ratio to compute the effective drop size of the DHs for a given voltage setting, pulse length, and run. To this end, we first calibrate the raw mixing ratio signal measured by the Picarro with a calibration curve obtained from

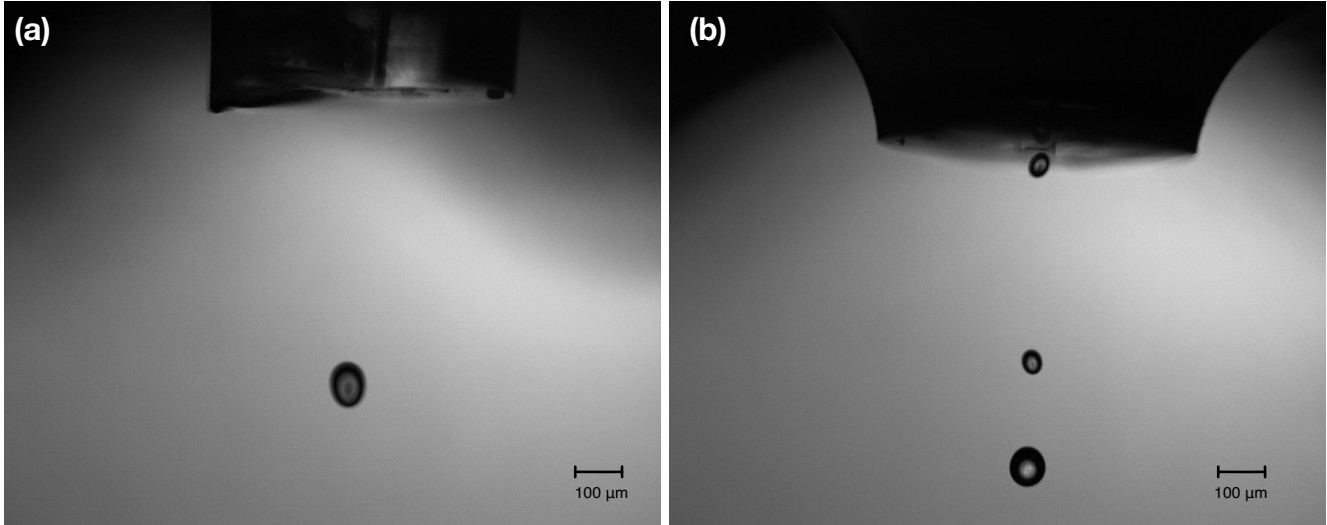

**Figure 2.** Pictures of microdrop dispenser heads in operation with different droplet examples. (a) Single drop emanating from DH with serial number 1016 at 85 V and 16 $\mu$s pulse width. (b) Jet-like drops emanating from DH with serial number 1015 with 85 V and a pulse width of 100 $\mu$s. Images have been obtained with a time-synchronized USB-b/w-camera with 10x objective (part #MD-O-539-USB, Microdrop GmbH, Germany).

**Table 1.** Mixing ratio and relative humidity at 60 °C of the air-vapour stream calculated for a typical and maximum range of dispensing parameters resulting in a drop size of 70 $\mu$m. A typical range of mixing ratios through the L2130-i and L2140-i analyzers is between 200–40000 ppmv at a flow range of 35–300 sccm.

| Frequency (Hz) | Carrier gas flow (sccm) | Mixing ratio (ppmv) | Relative humidity (%) |
|---|---|---|---|
| 10 | 50 | 795 | 0.7 |
| 10 | 250 | 160 | 0.1 |
| 1000 | 50 | 79511 | 72.2 |
| 1000 | 250 | 15902 | 14.5 |

a dewpoint hygrometer (Optisonde, GE Inc., USA). Then we compute the mass flux of the carrier gas at a given temperature, and obtain the effective drop size from the water mass contained in the air-gas mixture for a given frequency.

Day-to-day standard deviations of drop size are typically only a few micrometers (Fig. 3a, red and blue histogram), using dispensing parameters 82 V, 18 $\mu$s for DH1 and 82 V, 19 $\mu$s DH2. Median drop sizes are 64 $\mu$m for DH1 and 54.5 $\mu$m for DH2. The difference between the two DHs is a consequence of the DH geometry, which only allows to stably dispense single drops for a discrete sub-set of sizes that are particular for each head. Figure 3b shows the mixing ratio resulting for different

**Table 2.** Performance characteristics and calibration parameters for 3 MK-K-130 DHs used in the microdrop device. As DH3 was purchased as a spare at a later time, we herein focus on results obtained with DH1 and DH2. Factory calibration parameters can be modified to obtain different drop sizes.

| DH | 1 | 2 | 3 |
|---|---|---|---|
| Serial number | 1015 | 1016 | 1032 |
| Nozzle diameter | 50 $\mu$m | 50 $\mu$m | 50 $\mu$m |
| Frequency range | 1–1500 Hz | 1–1000 Hz | 1–1000 Hz |
| Tubing | 10 cm PTFE | 10 cm PTFE | 11.5 cm PTFE |
| Calibration voltage | 51 V | 58 V | 54 V |
| Calibration pulse length | 39 $\mu$s | 23 $\mu$s | 26 $\mu$s |
| Calibration drop size | 68 $\mu$m | 74 $\mu$m | 75 $\mu$m |

dispensing frequencies. Very limited scatter is apparent for both heads that represents day-to-day variability. Due to the different drop sizes, both DHs have different efficiencies, quantified by the slope of a linear fit. DH1 (blue) has for the given settings an efficiency of 207 ppmv Hz$^{-1}$, compared to 130 ppmv Hz$^{-1}$ for DH2 (blue). The results in Fig. 3b are obtained for well-tuned dispensing settings, where discrete bubbles were ejected from both DHs (compare Fig. 2b). If the system is operated using dispersion parameters that lead to jets, the performance characteristics degrade markedly, both in terms of linearity and day-to-day reproducibility (not shown). Overall, this analysis demonstrates that once suitable dispensing parameters have been determined, the microdrop device produces a well-constrained linear relation between the amount of liquid released from a dispensing head at a given frequency and the water vapour mixing ratio in the air stream.

### 4.3 Liquid water standard preparation

The water standards used here are secondary laboratory standards calibrated at FARLAB, University of Bergen on the VSMOW-SLAP scale with primary standard material provided by IAEA, according to their recommendations (IAEA, 2009). We used a depleted standard (GLW, -40.10±0.03 for $\delta^{18}$O; -308.8±0.1 for $\delta^2$H) and a standard close to local meteoric waters (DI2, -7.64±0.02 for $\delta^{18}$O; -49.8±0.3 for $\delta^2$H). Prior to filling DH reservoirs, the standard waters were filtered using a 0.2 $\mu$m PTFE filter (Part No. 514-0066, VWR, USA). This procedure is recommended by the manufacturer to prevent clogging of the DH capillary and PE tubing, and thus damage to the DHs. Furthermore, gas bubbles forming in the capillary are a major factor that can lead to arbitrary stopping of dispensing. Effective de-gassing is particularly important since our water standards are stored in stainless steel containers pressurised with Argon at about 1 bar. We initially removed dissolved gas from the liquid by suspension of the vials in an ultrasonic bath for about 15 min. After discussion with Microdrop GmbH, we found application of a vaccuum in a closed syringe dramatically increased effectiveness of the de-gassing, resulting in regularly uninterrupted dispensing of more than 24 h at a time.

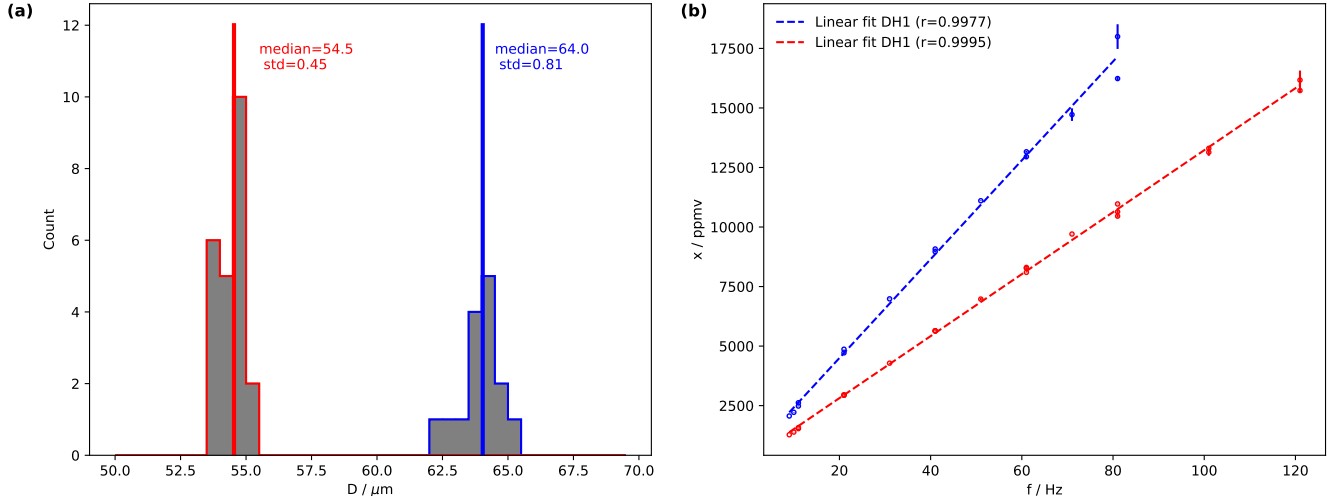

**Figure 3.** Performance of two dispenser heads at fixed droplet generation parameters and variable frequency. (a) Histograms of droplet size computed for DH1 from the dispensing parameters 82 V, 18 $\mu$s (blue) and for DH2 for 82 V, 19 $\mu$s (red) with median and standard deviation. Linearity of water vapour mixing ratio (ppmv) versus dispensing frequency (Hz) for DH1 with dispensing parameters 82 V, 18 $\mu$s (ref) and for DH2 with 82 V, 19 $\mu$s (blue). Measurements have been performed on a Picarro L2140-i analyser (serial no. HKDS2038).

## 4.4 Operating procedures

Before starting to operate the microdrop device, the standards need to be filtered and degassed, filled into reservoirs, and the
225 device heated to the operating temperature of 60 °C (typically <30 min) with gas flow enabled (>35 sccm). This heating time will allow the device to reach a residual background mixing ratio of well below 100 ppmv. Thereafter, the standard operating procedures for the calibration device to produce a vapour stream include the following steps:

1. Remove DHs from evaporation chamber

2. Empty and fill DH with standard water using software control, wait for holding pressure to stabilise

3. Test DH operation by ejecting onto a lint-free piece of cloth, lit by a bright light source on a dark background at high dispensing frequency (1000 Hz)

4. Stop dispensing and insert DH in evaporation chamber, attach and seal DH

5. Start dispensing at desired frequency from software.

These steps need to be followed at the start of a measurement sequence, but also each time when DH operation stops
accidentally (Sturm and Knohl, 2010). Critical aspects during this normal operation sequence, and error sources leading to interruptions or poor calibration system performance are discussed in Sec. 8.

## 5  Device characteristics

This section presents typical results for the device characteristics regarding the linearity of the DH performance, stability of the calibration air stream, and ability of the system to generate mixtures of different standard waters at different flow rates.

### 5.1  Short-term stability

Stability of the water vapour mixing ratio and isotope composition of the calibration device are important in isotope analysis, and need to be quantified. In order to assess stability of the microdrop vapour device across a range of time scales for a continuous period of operation, we use the Allan deviation, which is given by the square root of the Allan variance (Allan, 1966; Sturm and Knohl, 2010):

$$\sigma^2(\tau) = \frac{1}{2n} \sum_{i}^{n} \left( y_{i+1}(\tau) - y_i(\tau) \right)^2 \tag{1}$$

hereby, $\tau$ is the averaging time, $y_i$ is the average value of measurements in an averaging interval $i$, and $n$ is the number of averaging intervals for a given averaging time $\tau$.

The minimum of the Allan deviation indicates the optimal averaging time, where the highest measurement precision can be achieved. While measurement precision is primarily a property of the CRDS analyser, the vapour stream needs to be sufficiently stable to enable analyser characterisation. Thus, variability of the vapour stream needs to be minimal to identify instrument characteristics, rather than calibration system characteristics from the analysis of the Allan variance. The Allan deviation has been assessed from a constant operation of the microdrop device, dispensing at 24 Hz with DH1 into a $N_2$ stream of 40 sccm during about 28 h, resulting in an average mixing ratio of about 10'440 ppmv. For averaging times below 10 s, the Allan deviation of the mixing ratio is less than 2 ppmv, and remains below 10 ppmv for all remaining averaging times (Fig. 4, red line). The maximum variance of about 10 ppmv is reached after 400 s. We compare this to a measurement sequence obtained with the SDM and the same standard with daily calibrations during 25–29 March 2021. In total five calibration sequences have been combined to a 1:45 h time series for the assessment of the Allan deviation of the same analyser with an SDM (Fig. 4, black lines). Thereby, we acknowledge that the SDM is not designed for mixing ratio stability, but for the stability of the isotopic signal. For mixing ratio, the SDM has a factor of 10 or more lower precision than the microdrop device, with an overall minimum of 10 ppmv. For $\delta^{18}O$, the Allan deviation with the SDM is initially lower than for the microdrop device, but reaches a minimum of only 0.05 ‰ after about 4 s, whereas the microdrop device provides the highest precision of 0.004 ‰ after about 1000 s (Fig. 4b). For $\delta^2H$, the findings are similar, with a minimum in the Allan deviation for the SDM at 4 s with about 0.3 ‰, compared to below 0.02 ‰ after 1000 s with the microdrop device (Fig. 4c). The results obtained for the microdrop device are in a very similar range as with a capillary vapour generator obtained by Steig et al. (2014) and for the Allan deviation generated by WVIA in Aemisegger et al. (2012) for the same type of analyser. From this analysis, we note that the microdrop device is able to provide a vapour stream with properties that allow to determine the precision of the analyser consistent with literature. The SDM vapour stream used here was substantially more noisy, and does not appear to be equally

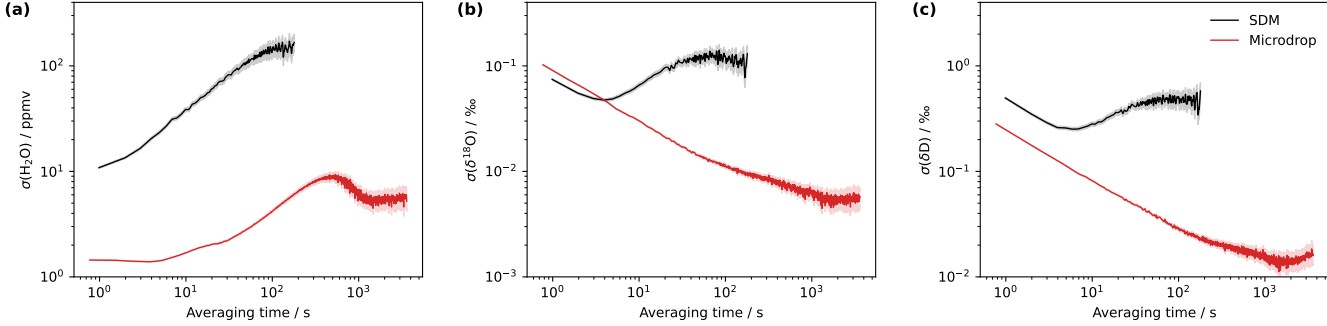

**Figure 4.** Allan deviation for (a) mixing ratio (ppmv), (b) $\delta^{18}O$ (‰) and (c) $\delta^2H$ (‰) by the microdrop device (red) and an SDM (black) on analyser HKDS2039 with standard GLW. The microdrop measurements were performed in the laboratory on 12–14 August 2023. The SDM measurements were combined from a set of shorter daily calibration periods obtained from 25–29 March 2021.

useful for analyser characterisation. These findings underline that it is important to consider the contribution of the vapour generation device when characterising the short-term stability of a CRDS water isotope analyser, in particular for applications that require precise control of the mixing ratio, such as fluid inclusion analysis.

### 5.2 Long-term stability

Stable background water vapour mixing ratios and $\delta^{18}O$ and $\delta^2H$ values are particularly important for accurate and precise fluid inclusion data. We evaluated the long-term stability of the microdrop system as part of a crushing line setup (Sec. 7). From 16 analytical sessions with the crushing line over a period of 5 weeks, we obtain $H_2O$ concentrations between 9550 and 13500 ppmv for 37 different occasions (Table A1). We selected a duration of 15 min to calculate short-term stability of each fluid inclusion sample peak measurement in that period (see Sec. 7). As a measure of long-term stability, we find an average standard deviation of 11.6±6.6 ppmv for water mixing ratio, 0.10±0.01 ‰ for $\delta^{18}O$, and 0.65±0.02 ‰ for $\delta^2H$ values on a 15 min time scale in this humidity range (Fig. 5). These standard deviations are much lower than typical sample peak heights during fluid inclusion measurements (see Sec. 7 and Fig. 13 below) as required for reliable fluid inclusion analysis (Affolter et al., 2014; Dassié et al., 2018; de Graaf et al., 2020). The long-term stability obtained here is also in a similar but slightly larger range as the short-term stability (Fig. 4). The larger variability on longer time scales can be induced by different factors in the analytical line setup and the microdrop device, including variation in dispenser head performance due to potential bubble formation, build-up of residual in the crusher line, among others (Sec. 7 and 8).

### 5.3 Variation of carrier gas flow rate

The currently used mass flow controller allows to regulate the flow rate in a range of 0–500 sccm. When using a Picarro L2140-i analyser in low-flow mode, about 40–50 sccm of gas flow are required to avoid leakage of ambient air from the open split. Importantly, with larger gas flow, the residence time of the vapour stream in the microdrop device decreases, and flow

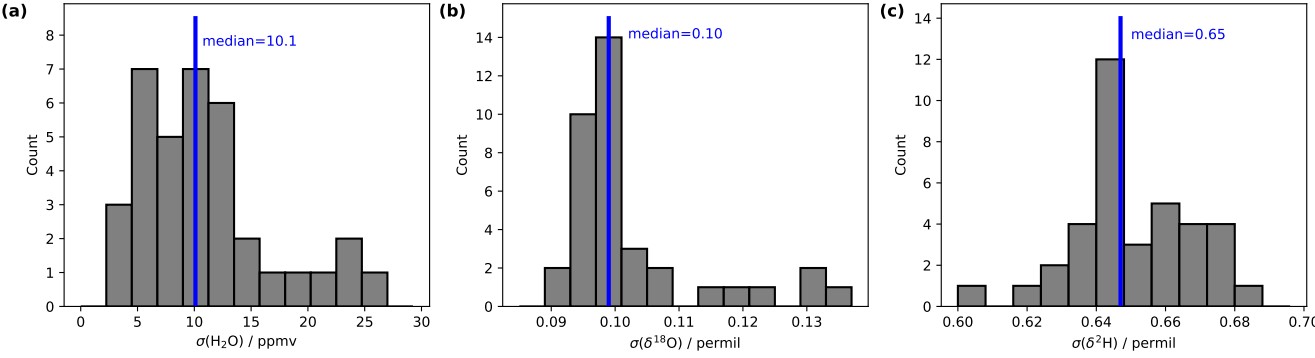

**Figure 5.** Long-term stability of the vapour stream assessed during a range of measurement days in Oct-Nov 2020. Histograms show standard deviations on (a) water vapour mixing ratios (ppmv), (b) $\delta^{18}O$ (‰) and (c) $\delta^2H$ (‰) values, calculated over 15 min long time intervals (n=37).

can become turbulent in the evaporation chamber. Thus, higher flow rates can lead to better mixing conditions, and offset the falling of droplets to a larger degree. On the other hand, slight overpressure may build up in the evaporation chamber, which counteracts the operation of the DH. We therefore investigated the performance of the device in terms of the Allan deviation (Eq. 1) for water vapour and the isotope composition at three different flow rates. To this end, DH1 was operated with laboratory standard DI2 for about 3-hour long segments with $N_2$ flow rates of 70, 140 and 250 sccm, and respective dispensing frequencies of 120, 210 and 350 Hz during 08–12 March 2022. The mixing ratio in all three setups was between 18'000 and 20'000 ppmv . The Allan deviation for 70 and 140 sccm is very similar for both mixing ratio (Fig. 6a) and the two isotope values (Fig. 6b,c). At averaging times of 30 s and longer, the Allan deviation becomes about 2 times larger for 140 sccm (black line) than 70 sccm (red line). At a flow rate of 250 sccm (blue line in Fig. 6), the Allan deviation is larger for the mixing ratio throughout the range of averaging times (Fig. 6a), while for both isotope species, the Allan deviation only increases for averaging times of more than 10 s. This indicates that the microdrop device can produce a usable vapour stream at flow rates of up to 250 sccm. However, the mixing ratio is more variable, and an additional mixing chamber may be required downstream to further stabilise the vapour stream.

## 5.4 Mixing of water from two dispenser heads

Due to the two independently operated DHs, it is possible to produce a vapour stream that is freely mixed two water standards that have large differences in isotopic composition. We have tested the linearity of the mixing across a range of frequency settings using standards DI2 and GLW. Using frequency ratios $r = DH2/(DH1 + DH2)$ of 0.0, 0.2, 0.3, 0.4, 0.5, 0.6 and 1.0, we step through a range of frequencies for both dispenser heads. For example, to obtain a mixing ratio of 5000 ppmv at $r = 0.2$, we dispense with a frequency of 5 Hz from DH2 and 20 Hz from DH1 (ignoring sensor head efficiency for simplicity at this point). The mixing between DHs is to first order linear (Fig. 7a). In particular for lower mixing ratios, deviations become apparent (color shading). Both the efficiency of the dispenser heads (Sec. 4.2), and the combination of the frequency of each

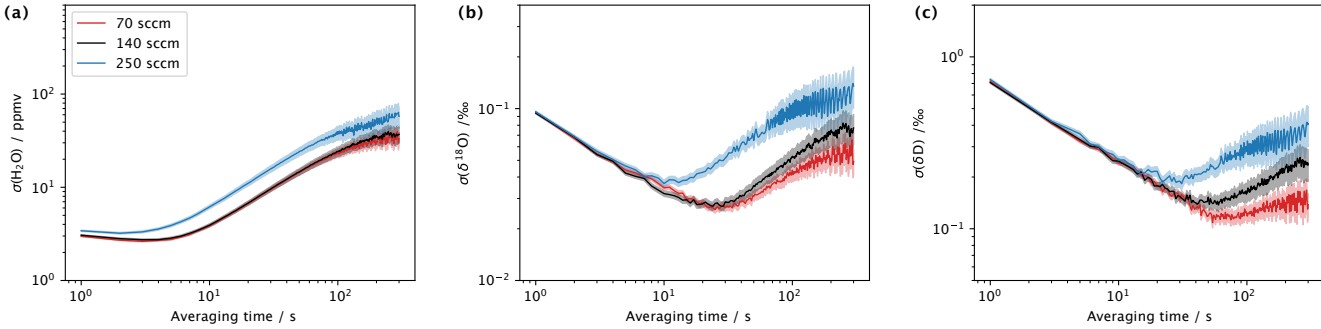

**Figure 6.** Comparison of Allan deviation for (a) water vapour mixing ratio (ppmv), (b) $\delta^{18}$O (‰), and (c) $\delta^2$H (‰) at flow rates of 70 sccm (red), 140 sccm (black), and 250 sccm (blue) obtained from a sequence of measurements at mixing ratios of 18'000-20'000 ppmv during March 2022.

DH, which is an integer number in the current version of the control software, need to be taken into account for precisely regulating the mixture. At lower humidity than 5000 ppmv, some of the deviations are due to spectroscopic baseline effects. In the experiments presented here, the lowest RMSE between the predicted and actual isotope composition was obtained with mixing ratios of 20'000 ppmv (Fig.7b), with 0.36 ‰ for $\delta^{18}$O (red crosses) and 1.46 ‰ for $\delta^2$H (black dots). At lower humidity, the RMSE was typically around 1–2 ‰ for $\delta^{18}$O and 10–15 ‰ for $\delta^2$H. With corrections to the efficiency of the heads and more suitable combinations of dispensing frequency and gas flow, we expect that a more precise control of the mixing between end members can be achieved. Nonetheless, we conclude that by varying the dispensing frequency at a fixed frequency ratio, and in addition changing the flow rate, a vapour stream with given isotope ratio can be produced over a wide range of mixing ratios. Thus, the microdrop vapour generator can be used to quantify for example spectroscopic baseline effects for nearly arbitrary positions in the mixing ratio – isotope ratio space (Sec. 6).

## 6 Application 1: Characterisation of the isotope composition-mixing ratio dependence of CRDS analysers

We now present a first application example for the use of the microdrop vapour generator. In this application example, the device is directly connected to the input port of the analyser with an open split (Fig. 1). We then exploit the capability of the calibration system to independently vary both mixing ratio and isotope ratio, and thus obtain a detailed characterisation of the mixing ratio – isotope ratio dependency. As noted above, this dependency is a measurement artefact that results from spectroscopic effects, such as uncertainties from the spectral fitting and the correction of baseline effects (Sturm and Knohl, 2010). In particular at water vapour mixing ratios below about 10000 ppmv, the raw measurements need to be corrected for this artefact. Early studies detected first a mixing ratio dependency (Lis et al., 2007; Sturm and Knohl, 2010; Steen-Larsen et al., 2013; Bastrikov et al., 2014). Based on a systematic investigation with different water standards, Weng et al. (2020) detected in addition a systematic dependency on the isotope ratio, and proposed corresponding correction functions. The shape of this

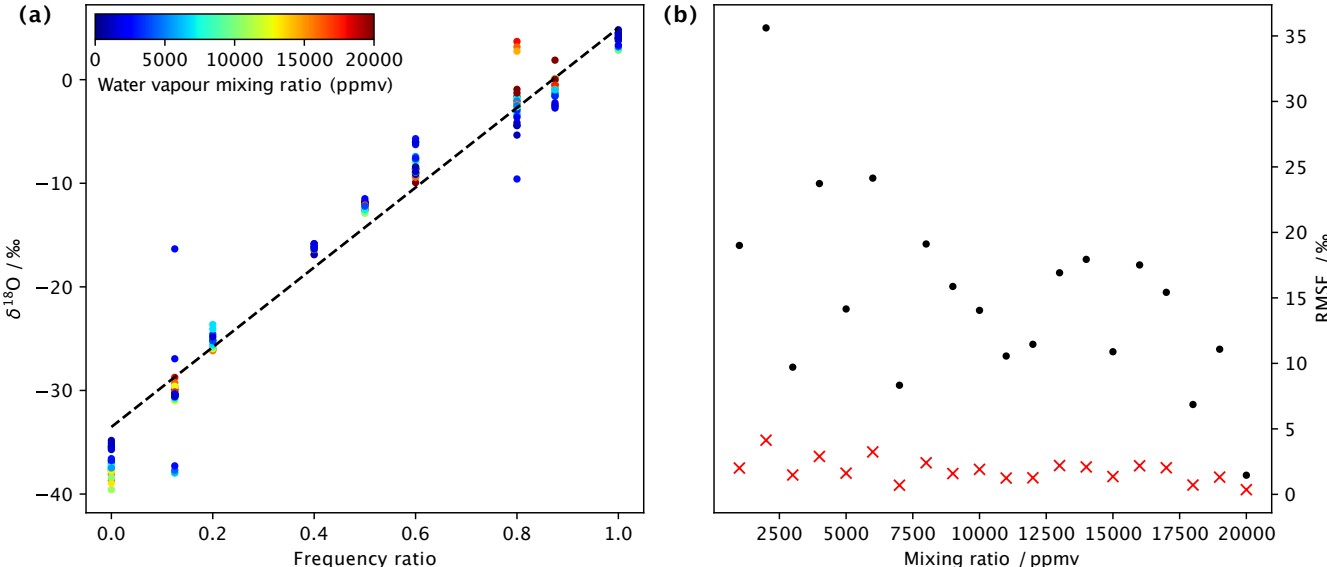

**Figure 7.** Stability of isotope standard mixing across range of humidities at an averaging time of 10 min. (a) Mixing line between two end members (laboratory standards DI2 and GLW) as a function of the ratio of dispensing frequency for $\delta^{18}$O (‰) at different mixing ratios (ppmv, color shading). Dashed black line is a linear regression to all available data points. (b) Root-mean square error (?), calculated from a linear regression in comparison to the data points for $\delta^{18}$O (red crosses) and $\delta^2$H (black dots) mixing line for categories of different mixing ratio at a 1000 ppmv interval.

measurement artefact specific for each analyzer, and also the used carrier gas (Aemisegger et al., 2012; Bailey et al., 2015). As electronics and other components age, this dependency needs to be obtained at regular intervals to maintain high data quality for water vapour isotope measurements.

The correction procedure of Weng et al. (2020) allows to correct the uncalibrated isotope measurements for the combined water vapour mixing ratio and isotope ratio dependency, in particular for low mixing ratios. However, the characterisation experiments are time consuming with existing water vapour generators, require manual preparation and calibration of additional standard waters, and are thus usually based on a limited number of standards (5 in the case of Weng et al. (2020)). The microdrop calibration device presented here alleviates these difficulties by allowing to create an arbitrary mixture of standard waters at a specific isotope composition and mixing ratio, thereby providing the mixing ratio – isotope ratio dependency for a given analyser in high detail, specifically in regions of the mixing ratio – isotope ratio space where the artefact is particularly pronounced.

## 6.1 Experimental setup

The microdrop vapour generator was connected directly to the inlet of the Picarro analyser (L2130-i, Ser. No. HIDS2254 or L2140-i, Ser. No. HKDS2038) with a stainless steel T-piece. Thereby, the overflow port was connected to the WLM Purge Port, secured by a check valve (Part Nr. SS-2C-1/3, Swagelok Inc, USA) to prevent leakage of ambient air into the analyser inlet line. To achieve the desired humidity, the DH frequency was stepped up and down within a range of typically 1 to 100 Hz, while the flow rate of the dry matrix gas was set to 70, 100, 200 and 300 sccm.

Mixtures of the end member standards were injected by altering the DH frequency ratio for each of the two heads. For example, a 50 % contribution of enriched standard from DH1 (raw $\delta^{18}$O=2.6 ‰ and $\delta^2$H=12.1 ‰ at approximately 20000 ppmv) and 50 % of depleted standard (raw $\delta^{18}$O=-39.7 ‰ and $\delta^2$H=-311.9 ‰ at approximately 20000 ppmv) from DH2 would result in an intermediate raw mixed signal (raw $\delta^{18}$O=-16.0 ‰ and $\delta^2$H=-130.5‰ at approximately 20000 ppmv), taking into account the efficiency of the two DHs (see Sec. 4.2).

Each sequence typically covered a humidity range from about 20000 to 500 ppmv. An example for a typical humidity sequence during such a run shows the end of an initial drying of the evaporation chamber to reach sufficiently low background mixing ratio between 13:30 and 13:35 down to <100 ppmv (Fig. 8a, label A). Then, the sequence starts by flushing the evaporation chamber with moist air at a frequency ratio of 0.6 at a maximum frequency of 50 Hz for 5 min until 13:40 (label B), followed by another drying sequence to remove some vapour remaining from the previous run together with the new vapour until 13:45 (label C). Then, at 13:45, the first step starts dispensing from both DH at frequency ratio 0.6 (label D, blue dashed line). To allow for a more quick transition between different mixing ratio steps, the first seconds at the start of each step the target frequency have been exaggerated. For example, when going from 45 to 39 Hz with DH1, a 19 Hz frequency was used for 6 s to provide a faster step down (Fig. 8, label D). Then, the frequencies of both DHs were kept constant for 10 min, providing flat segments of water vapour mixing ratio (Fig. 8a, label E) and isotope ratios (Fig. 8b,c). For the characterisation of the mixing ratio – isotope ratio dependency, the initial 5 min of each segment were removed, while the following 5 mins were retained and averaged (Fig. 8, red segments). The precision of water vapour mixing ratio during the retained segments were typically 8.2 ppmv (median of one-sigma standard deviation for retained 5 min periods).

## 6.2 Obtaining an isotope composition-mixing ratio dependency correction for a CRDS analyser

To obtain the isotope composition-mixing ratio dependence functions for the same CRDS analyser as in Weng et al. (2020) (L2130-i, Ser. No. HIDS2254), experiments with five different DH frequency ratios (100-80-60-40-20) covering the humidity range from 450 to 25000 ppmv have been conducted. Each experiment was repeated at least three times to obtain statistical significance of the measured results. Then, averages and standard deviation for each humidity step identified as in Sec. 6.1 were calculated. After an arbitrary stop of one of the DHs manual intervention was required. In some cases, leakage of water vapour from the second, stopped dispenser head required filtering for outliers. Here, measurement points for which the isotopic composition differs from the expected value at 20000 ppmv by more than 20 % have been excluded as outliers. Furthermore, the points for the mixing ratio above 5000 ppmv for which the standard deviation of the $\delta^{18}$O-value exceeded the 75th quantile

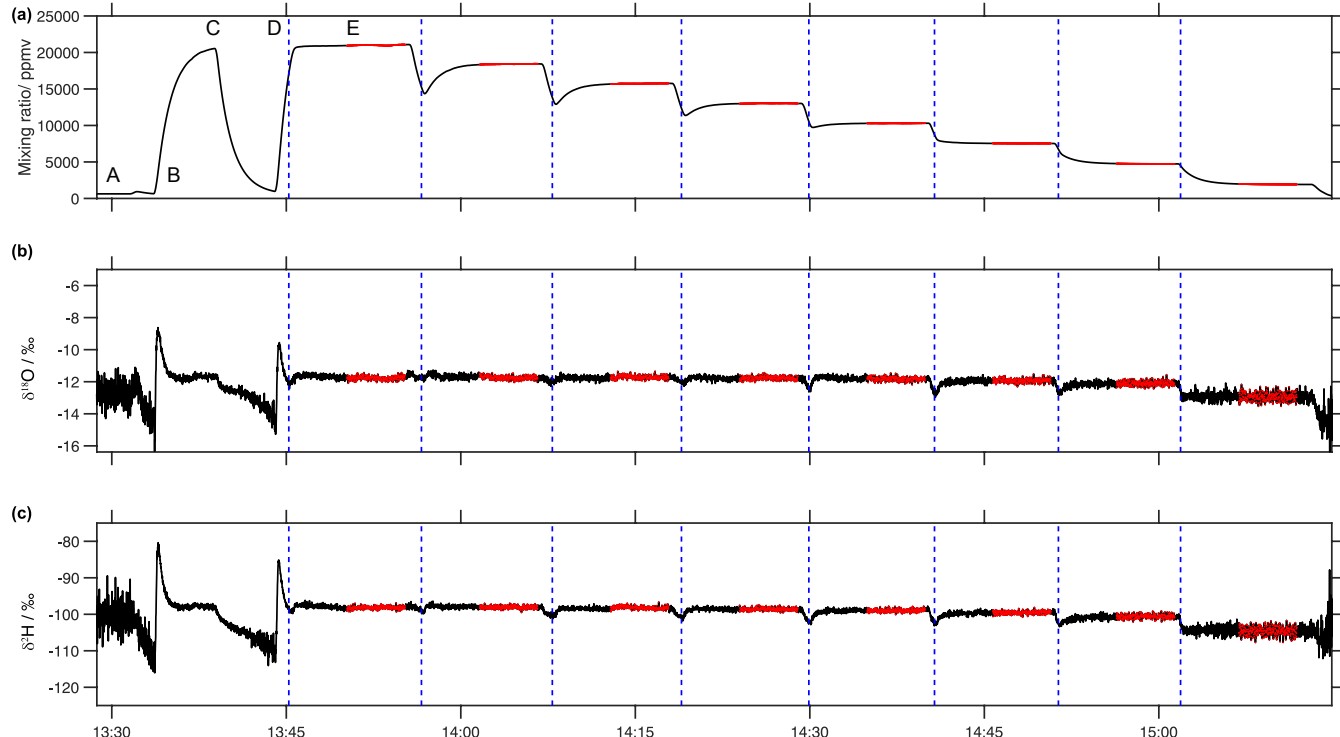

**Figure 8.** Typical sequence of (a) water vapour mixing ratio (ppmv), (b) $\delta^{18}$O (‰) and (c) $\delta^2$H (‰) during a calibration run on the microdrop device. Example depicts a run with Picarro analyser HKDS2038 using synthetic air as a carrier gas, and a mixture of standards (60% DH1 and 40% DH2) on 04 June 2021, with 8 steps of 10 min duration (marked with dashed vertical blue lines) and 5 min stable period within each step (marked with red colour) used to calculate the average values and standard deviation of stable water isotopes and mixing ratio.

of the standard deviation of $\delta^{18}$O for all measurements have been removed. These outliers occurred due to strong variability in mixing ratio during the step or due to unstable functioning of one or both of the DHs, possibly due to formation of gas bubbles
inside the DHs. With more effective de-gassing of the standards, the number of stopped runs leading to outlier removal has been substantially reduced. The remaining 203 data points, supplemented by the measurement points of Weng et al. (2020), were then used to construct the mixing ratio – isotope ratio dependency correction (Fig. 9, blue points). As in previous studies, we use a reference value $\delta_{\mathrm{ref}}$ at 20'000 ppmv to determine the deviation $\Delta\delta_{\mathrm{corr}}$ of the raw isotope measurements from the reference value in units of ‰ on a $\delta$ scale for mixing ratio $x$:

$$380 \quad \Delta\delta_{\mathrm{corr}}(\delta, x) = \delta(x) - \delta_{\mathrm{ref}}. \tag{2}$$

While Weng et al. (2020) used hyperbolic functions of mixing ratio fitted to measurement points with a set of water standards, we use a new approach that is more flexible in terms of utilising data points that are mixed from different waters by the

microdrop device. To this end, we first remove the dominant first-order dependency on the mixing ratio by a log-transformation of the mixing ratio to $\ln(x)$:

$\quad \Delta\delta_{\mathrm{corr}}(\delta, \ln(x)) = \delta(\ln(x)) - \delta_{\mathrm{ref}}.$ (3)

To take all measurements into account, we obtain $\delta_{\mathrm{ref}}$ from a 2nd-order polynomial fit to each measurement sequence at a specified isotope ratio. Next, we fit a regular second or third order two-dimensional polynomial to the set of deviations. We find that different analysers require different polynomials for fitting in the $\Delta\delta$ and the $\ln(x)$ direction. For the four analysers investigated at FARLAB, a 2nd-order polynomial was sufficient along the $\ln(x)$ axis. Two of the four analysers had better
fitting results with a 3rd-order polynomial in the $\Delta\delta$ direction than with a 2nd-order polynomial. An example for analyser HIDS2254 shows that the fitted surfaces are to a large degree consistent with the measurement data (Fig. 9).

The 2D polynomial fit is not constrained to zero at the reference level and thus result in an offset at $\delta_{\mathrm{ref}}$. Therefore, as a final processing step to avoid false corrections at reference humidity, we subtract the deviation from zero of the fitted surface at reference humidity along each isotope ratio. Thereby, the RMSE of the correction surface in comparison to the measurement
points increases (Table 3). For $\delta^2\mathrm{H}$, the RMSE increase is smaller than for $\delta^{18}\mathrm{O}$, where the RMSE is very similar to the correction surface of Weng et al. (2020). Despite the different approach compared to Weng et al. (2020), the shape of the correction surface is to first order remarkably consistent with previously published results for the same analyser (Fig. 10a,b; solid and dashed contours). Compared to measurements, the root-mean square error of the correction obtained in this study is 1.15 ‰ for $\delta^2\mathrm{H}$ and 0.33 ‰ for $\delta^{18}\mathrm{O}$ with adjustment for zero at reference mixing ratio (Table 3). At lower mixing ratios, the
RMSE is largest, with 1.57 ‰ for $\delta^2\mathrm{H}$ and 0.43 ‰ for $\delta^{18}\mathrm{O}$ for the range of 0–5000 ppmv.

Due to the mixing capability of the microdrop device, it is possible to make additional measurements in-between available laboratory standards, and without mixing additional waters beforehand. We also extended the range of the correction to between 5 and -40 ‰ for $\delta^{18}\mathrm{O}$ and to 10 and -350 ‰ for $\delta^2\mathrm{H}$ using additional secondary standards. Notably, the microdrop device allows to obtain the data points needed for such a correction function in a semi-automated fashion and with large flexibility.
For example, based on this first screening of the dependency for this analyser, it would be possible to zoom into particularly critical regions of a given analyser's dependency function, such as the region with mixing ratios below 2000 ppmv in the $\delta^{18}\mathrm{O}$ dependency for the current device (Fig. 10b, red shading).

The surface fitting approach presented here also allows to quantify the uncertainty of the surface fits, thus providing access to the contribution from the mixing ratio – isotope ratio correction to the total uncertainty of a final data set. Here, we used
a Monte-Carlo approach to determine the standard deviation of the correction values. Thereby, a bootstrap resampling with 50 repetitions was used to draw non-unique samples from the entire dataset. The standard deviation of the 50 realisations are largest at the edges of the correction surface, with values of up to 0.6 ‰ for $\delta^2\mathrm{H}$ (Fig. 10c) and 0.2 ‰ for $\delta^{18}\mathrm{O}$ (Fig. 10d). Near the center of the correction functions, the uncertainty is much smaller than the correction itself, as constrained by the precision of the sub-sampled microdrop-generated dataset.

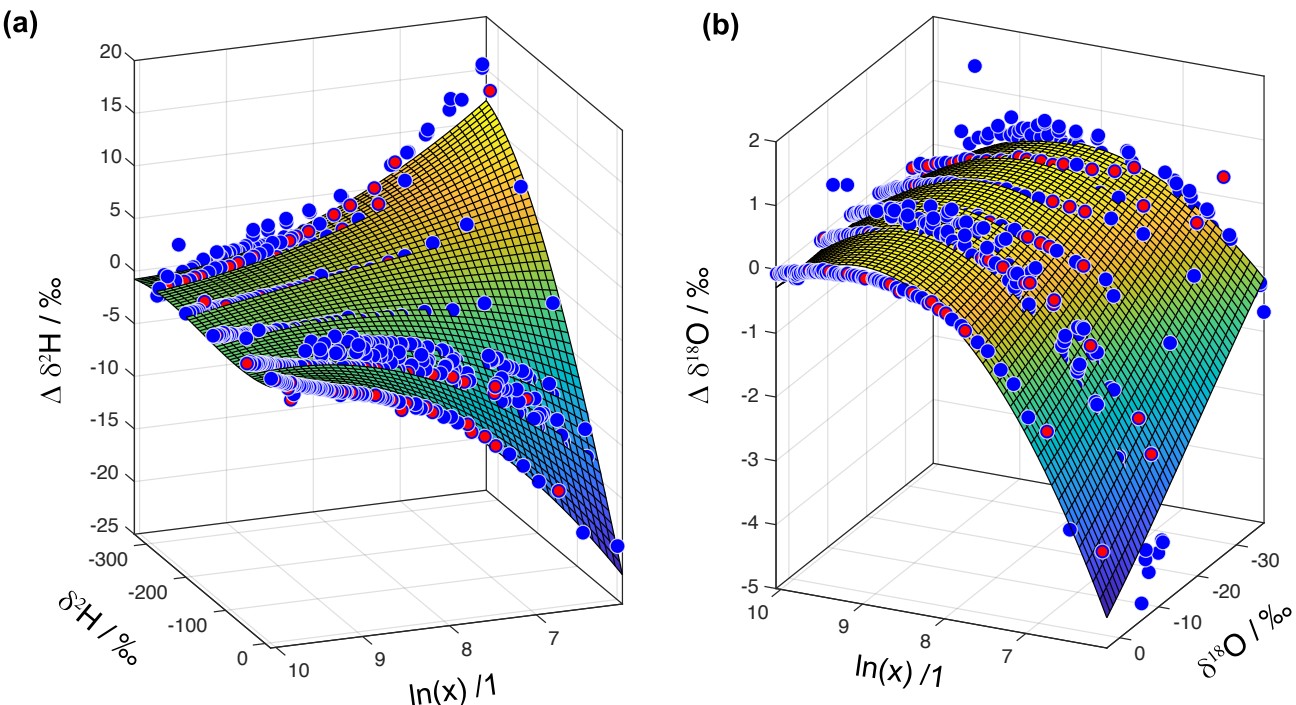

**Figure 9.** Surface fitting for instrument HIDS2254 for (a) $\delta^2H$ and (b) $\delta^{18}O$ as a function of ln(mixing ratio). Blue dots show measurement points obtained from microdrop device to constrain the surface fit. Red dots are measurement points obtained from autosampler injections (Weng et al., 2020)

.

**Table 3.** Difference between data points obtained from microdrop measurements and different mixing ratio – isotope ratio correction surfaces, quantified as root-mean square error (RMSE).

| Method | Mixing ratio range | RMSE $\delta D$ (‰) | RMSE $\delta^{18}O$ (‰) |
|---|---|---|---|
| Weng et al. (2020) | all data | 4.5797 | 0.4348 |
| polynomial fit | all data | 1.0230 | 0.2908 |
| polynomial fit* | all data | 1.1514 | 0.3259 |
| polynomial fit* | <5000 ppmv | 1.5714 | 0.4313 |
| polynomial fit* | 5000-10'000 ppmv | 0.8368 | 0.2514 |
| polynomial fit* | >10'000 ppmv | 0.6441 | 0.2054 |

*with zero adjustment for reference mixing ratio

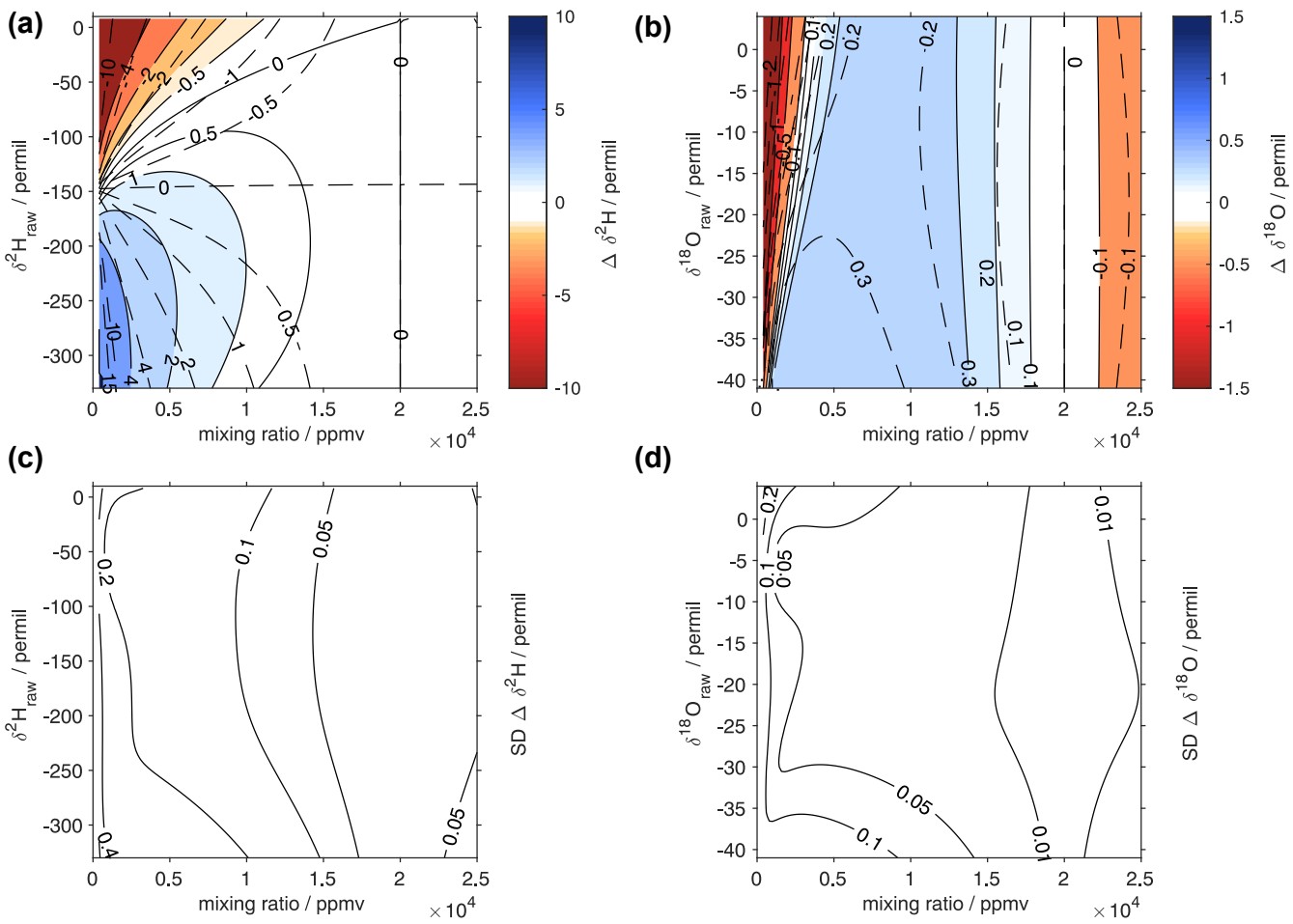

**Figure 10.** Surface function of the isotopic deviations for (a) $\delta^{18}$O and (b) $\delta^2$H based on the isotope composition – mixing ratio dependency of Picarro analyser HIDS2254 (Picarro L2130-i). The horizontal axis is the raw mixing ratio, and the vertical axis shows the raw isotope composition at 20'000 ppmv. Shading and solid contours with numbers indicate the isotopic deviation of $\Delta\delta$. Dashed contours show the correction functions of Weng et al. (2020). Lower row: standard deviation of correction function obtained from a Monte-Carlo approach (bootstrap resampling) for (c) $\delta^2$H and (d) $\delta^{18}$O (‰) using 50 random non-unique resamplings of the entire dataset.

## 7 Application 2: $\delta^{18}$O and $\delta^2$H values of fluid inclusions in stalagmites

We now present an example for an analytical set-up in the laboratory where the microdrop device produces a constant vapour stream with known mixing ratio and isotope composition for fluid inclusion analysis from stalagmites. Recently, Affolter et al. (2014) introduced an online method based on CRDS where the water isotope content of fluid inclusions is measured on a standard water background, which is subtracted after peak integration. The result is higher sample throughputs and better

reproducibilities than without a moist background (Affolter et al., 2014; Dassié et al., 2018; de Graaf et al., 2020). Recent work indicates that if a moist background is constantly provided to analytical devices (stainless steel lines and injection devices), memory effects during the analysis of liquid samples can be substantially reduced (de Graaf et al., 2021).

In all these applications, the precision of the background water vapour and its isotopic composition is critical for the analytical uncertainty of samples. Since the background is subtracted from the sample peak, large variations in the background (mixing ratio and isotopes) are propagated directly into the measured values. de Graaf et al. (2020) argue that even minor instabilities in the background can lead to a loss in measurement precision. Another problem that has been encountered with peristaltic pumps is drift of the background concentration during measurements (Weissbach et al., 2023, their Fig. 2). Here we describe a setup with the microdrop device for background vapour provision and the corresponding analytical procedures to obtain accurate and precise fluid inclusion $\delta^{18}O$ and $\delta^2H$ measurements.

## 7.1 Measurement and data reduction procedures

An analytical line for the sample crushing device (crusher) was built following the design described by Affolter et al. (2014) and de Graaf et al. (2020) (Fig. 11). The upper end is connected to the microdrop device, and the downstream end is connected directly to the CRDS analyser (Ser. No. HIDS2254, L2130-i, Picarro Inc) with a union tee fitted with a 1/16 inch capillary as open split. A check valve (Part No. SS-2C-1/3, Swagelok Inc., USA) was installed before the injection port to prevent backflow of sample waters. The injection port consists of a union tee fitted with a rubber septum. The last component is a particle filter (7 $\mu$m, Part No. SS-2F-7, Swagelok Inc., USA), which was installed to prevent sample powder from reaching the analyser. All parts of the line are heated to 120 °C with the use of an oven (FP 53, Binder GmbH, Germany) or silicon rubber heating tapes (EHG series, Watlow, USA). The crusher is made entirely of stainless steel and follows the design by de Graaf et al. (2020) with some modifications (Fig. 11). Two Viton O-rings are used to seal the device from atmospheric gases. One O-ring is placed on top of the base, and the other is placed on the piston. The piston is threaded to allow up and down movements.

At the start of an analytical session, the $N_2$ flow into the microdrop device is set to 90 sccm, whereby the excess above the flow of 35 sccm used by the analyser is vented through the open split (Fig. 11a). The DH frequency is then set to 60–120 Hz to produce a constant mixing ratio of typically 10'000 – 15'000 ppmv, depending on DH efficiency. A sample is subsequently loaded into the crusher (Fig. 11b), and a period of about 20 min is needed for the background $H_2O$ mixing ratio and $\delta^{18}O$ and $\delta^2H$ values to stabilise (Fig. 13, grey area). Once the background is stable, the oven is opened and the thread of the crusher is turned to pulverise the sample, which causes the fluid inclusions waters to evaporate instantaneously (Fig. 13, green dot denoted 'start'). During the release of sample water into the background air stream, we note an initial dip to more depleted isotope composition (Fig. 13b,c). Following Affolter et al. (2014), we speculate that this dip stems from kinetic fractionation during the adsorption of water molecules from the background vapour stream onto the newly exposed surface area of the calcite sample. Immediately thereafter, the water peak from the sample arrives on top of the background, here with an enriched signature in the sample. In order to quantify the total water released from the sample, integration is done between start and end of the sample peak (Fig. 13a, dashed black arrow between green dots). Typically, about 10–12 min are needed to measure a complete sample peak.

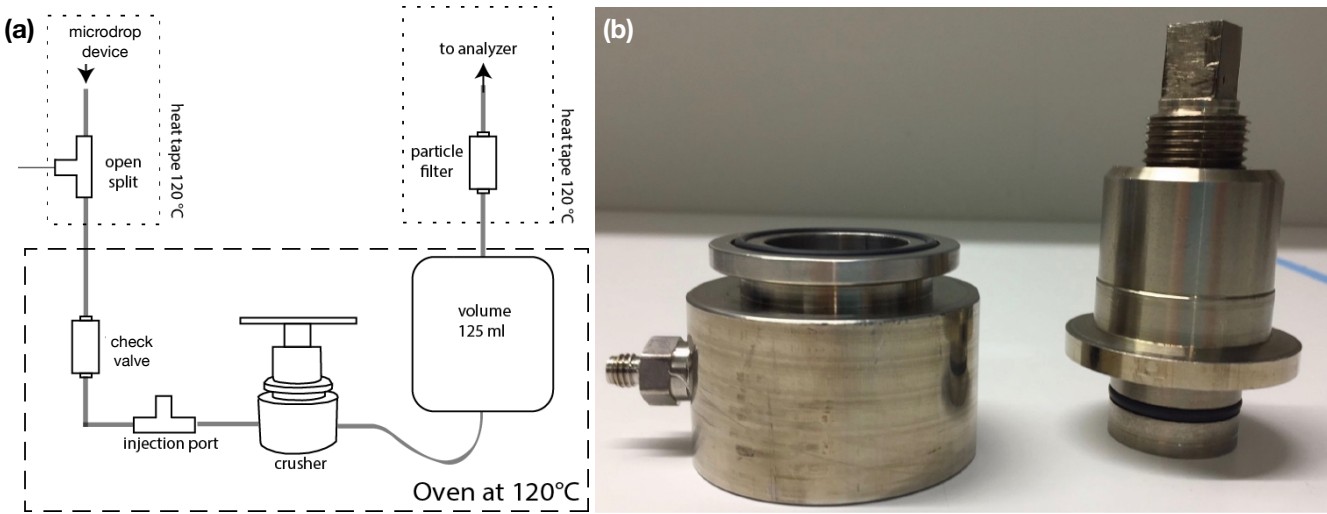

**Figure 11.** Setup of the crusher system. (a) Schematic of the analytical line for the crusher system. (b) Photograph of the crusher device, built after de Graaf et al. (2020).

After the water is measured by the analyser, the data is processed with a series of MATLAB scripts that: (i) correct $\delta^{18}$O and $\delta^2$H values for their humidity dependence (see Sec. 6), (ii) integrate peaks and subtract background values, and (iii) normalise $\delta^{18}$O and $\delta^2$H values to the VSMOW scale and calculate sample $H_2O$ concentrations with standards of known compositions. Mixing ratio – isotope ratio corrections have not been previously performed in fluid inclusion CDRS data, which may explain why some laboratories observe a relationship between water amounts and isotope ratio that varies for different standards (e.g., de Graaf et al., 2020). From a sequence of water standard DI2 injected at different amounts and on different water vapour mixing ratios for the background, we see that the mixing ratio – isotope ratio correction clearly translates into measurable signal differences, even when operating not far from the recommended operating range of the analyser (Fig. 12). Some samples with higher water amounts could require even lower background mixing ratios than the 10'000 ppmv applied here, which would lead to an even larger need to correct for spectroscopic baseline effects during peak integration.

During sample analysis, we integrate the area under the sample peak and subtract the background. This was done following the algorithms described in Affolter et al. (2014). Briefly, raw isotope and $H_2O$ ppmv data (peak and background) are first passed through a running mean filter with a 10 s window to smooth out high frequency variability. The algorithm then finds the start of the peak, which is defined when the rate of change in the water concentration exceeds 5.5 ppmv s$^{-1}$. Since the rate quickly increases and then turns negative at the top of the peak, the end of the peak is defined after the rate returns to positive values. Next, background values are calculated by averaging values before and after the peak; this was done to account for possible drift in the background isotope composition. After background subtraction, raw sample $\delta^{18}$O and $\delta^2$H values were normalised to the VSMOW scale with three different in-house standards, which were previously calibrated against international standards. For calibration, a range of water amounts (0.05 to $\sim$1.5 $\mu$l) were injected with the aid of autosampler GC syringes

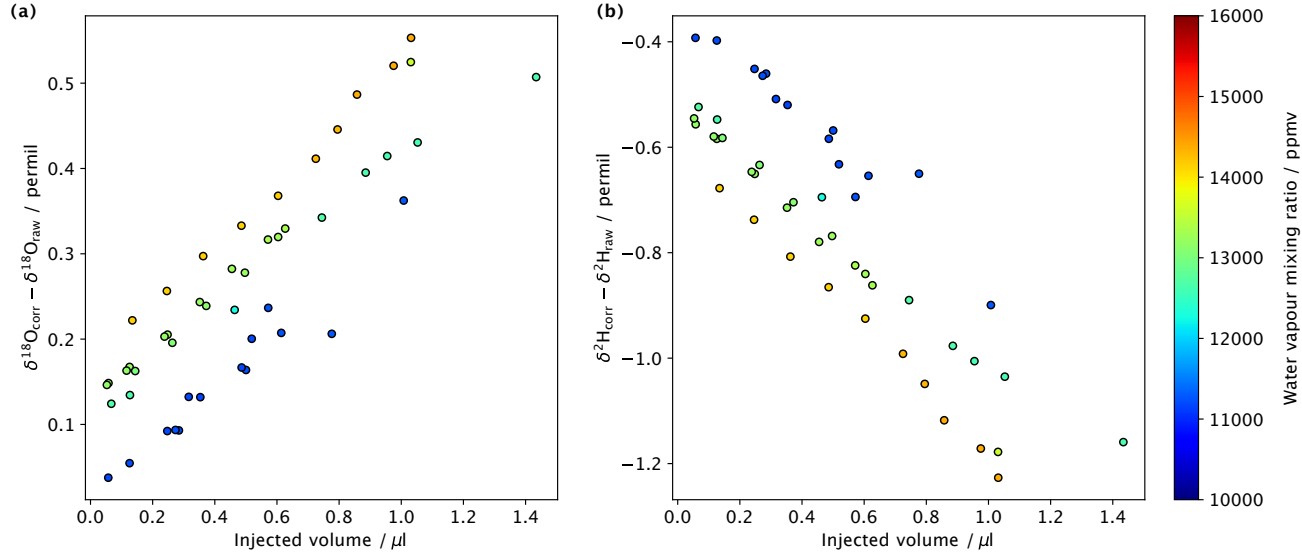

**Figure 12.** Difference between raw measurements and measurements corrected for spectroscopic baseline effects with the water vapour mixing ratio – isotope ratio dependency from liquid injections of water standard DI in the crushing line on different backgrounds (shading) for (a) $\delta^2$H (permil) and (b) $\delta^{18}$O (permil)

.

(0.5 $\mu$l SGE Analytical Science and 1 $\mu$l Thermo Scientific), and their mean values were used to build a transfer function (Fig. A2). This step ensures that samples and standards receive identical treatments, in particular as the crusher device can contain calcite remnants and additional components, such as the particle filter (Fig. 11). Finally, the amount of water released by the samples was obtained from a transfer function, that had been constructed previously from a series of injections of known amount of standard water.

## 7.2 Data accuracy and reproducibility

To estimate the reproducibility of our analyses we calculated pooled standard deviations every 0.1 $\mu$l (Fig. A3). This was done with the injections used for scale normalisation. We observe that standard deviations are very similar for samples larger than 0.3 $\mu$l (<0.3‰ for $\delta^{18}$O and <1.3 ‰ for $\delta^2$H), increase for smaller samples, and are the largest for samples smaller than 0.1 $\mu$l (0.7‰ for $\delta^{18}$O and 3.4 ‰ for $\delta^2$H). These reproducibilities are in the range of what has been observed in similar set-ups (Affolter et al., 2014; Dassié et al., 2018; de Graaf et al., 2020).

As a final test of data quality, we measured two samples that have known isotopic compositions. One of the samples consists of 24 aliquots of 0.2 to 0.6 $\mu$l of DI standard water that were sealed in borosilicate glass capillaries and crushed in the analytical line (Fig. 14c). Replicates of this sample are designed to mimic real fluid inclusions since, unlike injections, they receive the same treatment (i.e., water is released in the same location by turning the thread of the crusher device; Weissbach (2020)). We

find that the mean value of these analyses ($\delta^{18}$O=-7.5±0.3‰ and $\delta^2$H=-50.9±1.2‰; ±1$\sigma$) are statistically indistinguishable from their assigned values (Fig. 14a). The second sample (Fig. 14b) is an aliquot of a natural carbonate (Semproniano travertine) that has been previously measured in another laboratory. A total of 4 aliquots of this sample were analysed, and their mean values ($\delta^{18}$O=-4.3±0.3‰ and $\delta^2$H= -34.3±1.7‰;±1$\sigma$) are statistically indistinguishable from the results obtained by de Graaf et al. (2020) ($\delta^{18}$O=-4.6±0.32‰ and $\delta^2$H= -33.4±0.9‰; ±1$\sigma$, n=4).

In summary, the microdrop vapour generation device enabled a two-fold improvement of the application setup for the measurement of fluid inclusions. First, the characterisation of the analyser allows to correct for the mixing ratio – isotope ratio dependency of the analyser signal, providing a more accurate integrated signal of each sample than without this correction. With the variable water amounts and isotope composition in each sample, and a variable range of set background mixing ratios, this appears as an important additional processing step. Second, the precise background signal of the microdrop device with short-term variance of ∼10 ppmv on the time scale of the handling of the crushing device provides a noise-free environment to separate the signal of the fluid inclusion water reliably from the background water stream. In combination, both aspects demonstrate the value of the microdrop technology for specific water isotope measurement applications.

## 8 Critical operational aspects and potential error sources

During the operation procedure (Sec. 4), and during operation of one of the application examples (Sec. 6), a number of problems may occur that interrupt or disturb experiments. While in general working reliably over hours up to days, these problems are important to be aware of, and one needs to search for remedies in future development of the calibration system.

Potential problems ensue from undetected jets of droplets, that can lead to non-linear or noisy DH characteristics (Sec. 4.2). Finding suitable parameters for single droplet formation, rather than a jet of droplets, is cumbersome and time-consuming process, but should be a rarely repeated procedure for each DH that is utilised.

During droplet generation, several problems and interruptions can occur. Sturm and Knohl (2010) already reported that the droplet generation stopped occasionally, probably due to formation of bubbles in the injector as the liquid warms up. Therefore, gas removal in an ultrasonic bath and under vaccuum in the headspace (Sec. 4.4, step 2) before starting dispensing is imperative for reliable dispensing operations. We found that in particular a procedure where vaccuum is briefly applied to the headspace of the liquid makes gas removal dramatically more effective, and prolongs operating times from hours to days.

A further cause for stopping droplet generation can be clogging due to fine particles or residue of salts building up at the dispenser head. While salts can be removed by cleaning procedures, a clogged dispenser head may have to be replaced at some point. Filtering of the standard liquid (Sec. 4.4, step 1) is therefore imperative. While it is conceivable that direct evaporation from the DH capillary may occur at high temperatures and low frequencies, thereby stopping DH operation, more evidence is needed to confirm this potential error source.

A final parameter that could influence DH operation is the holding pressure. The holding pressure is a slight underpressure created by the microdrop controller in the standard vials to prevent liquid from running freely into the evaporation chamber. While typically a holding pressure of about -10 Pa was employed, it could be worth exploring dispensing behaviour for lower

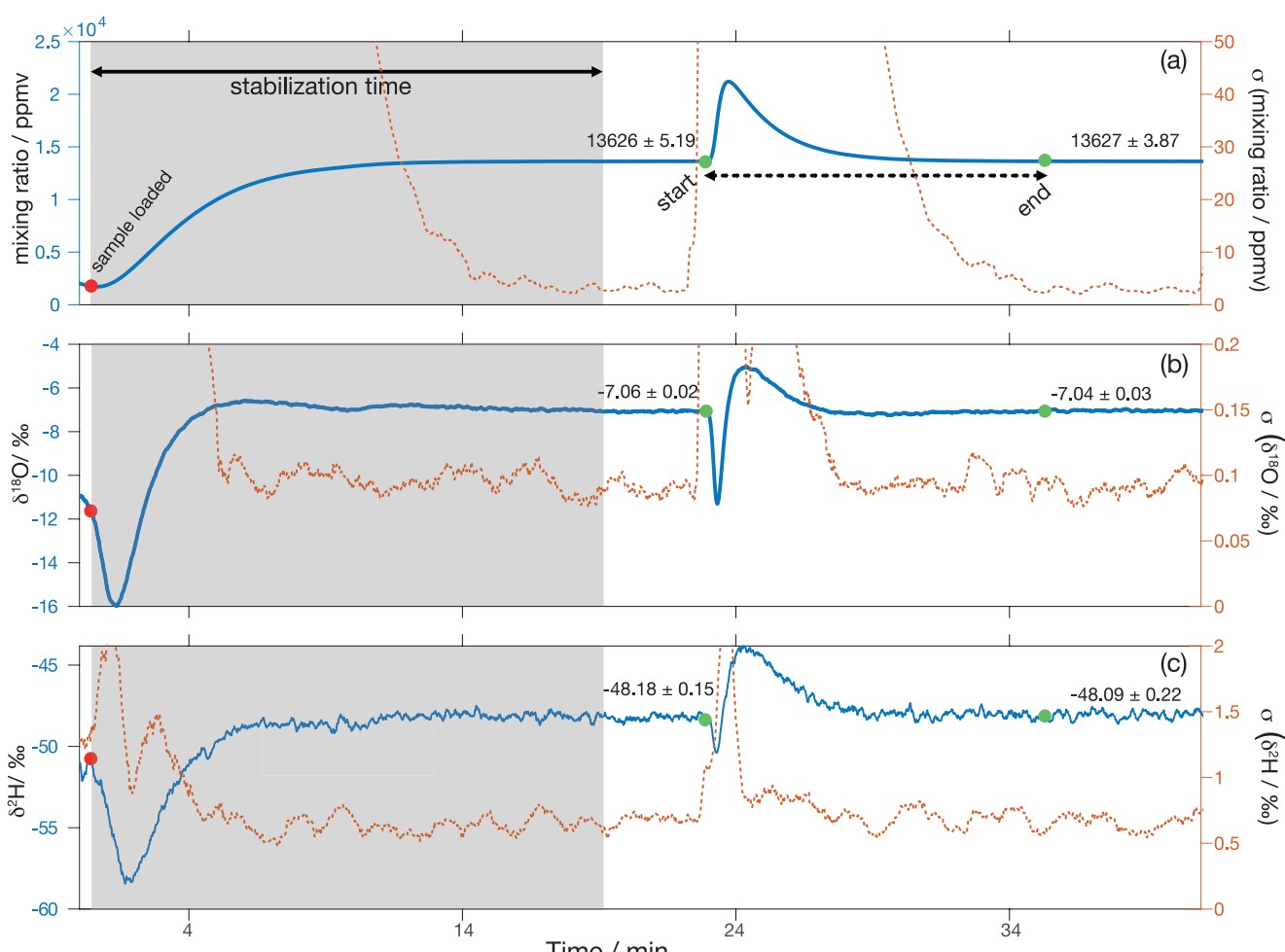

**Figure 13.** Example of a typical analysis with the crusher and microdrop device. Approximately 18–20 minutes are needed for (a) mixing ratio, and the two water isotope species (b) $\delta^{18}O$ and (c) $\delta^2H$ to stabilise after a sample has been loaded into the crusher device (red circle, ∼1 minute). The left axis shows average raw measurements acquired at 0.9 Hz filtered through a 30 s running mean filter, and the right axis shows the standard deviations of the running mean. Background average values before and after the peaks are shown in all panels. Peak values are integrated between the green markers, which show the start and the end of the peak.

holding pressure. This is in particularly important for our current design with horizontally mounted DHs, where gravitational forces are not aligned with the capillary axis. At higher flow rates, overpressure may build up in the evaporation chamber, which counteracts dispensing by pushing liquid back into the capillary. It may therefore be beneficial to determine the holding pressure relative to the pressure inside the evaporation chamber.

During operation of both dispenser heads, the discrete frequency settings for each DH cause limitations of the mixing ratios between both standard liquids. For example, if both DHs are to contribute no more than 5 drops per second in total, obtaining

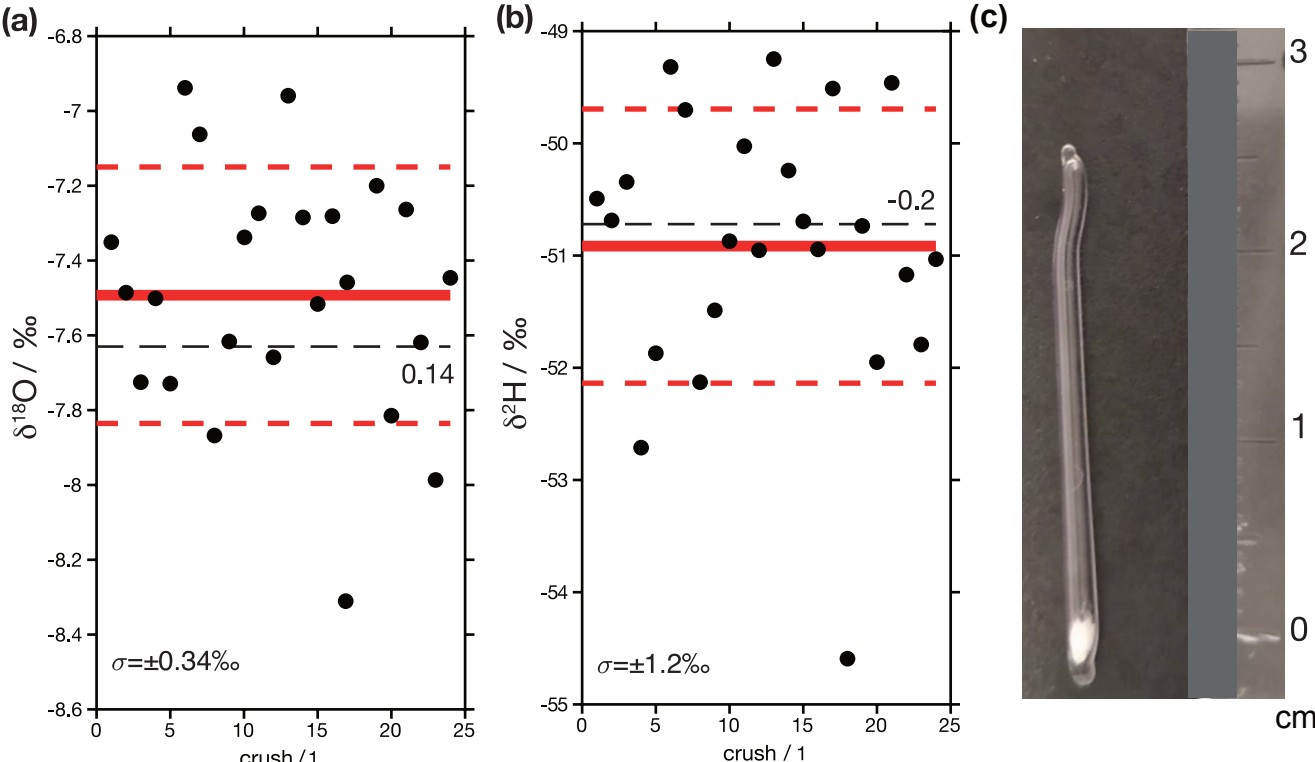

**Figure 14.** Results from crushed glass capillaries loaded with DI2 standard. (a) $\delta^{18}$O values. The red line shows the mean value of the replicates, and the dashed red lines are the $\pm1$ standard deviation. The black dashed line is the assigned value of the standard, which is 0.14 ‰ lower than measured mean value. (b) $\delta^2$H values. In this case the assigned value of the standard is 0.2 ‰ higher than the measured mean value. (c) Photograph of a loaded glass capillary. These are built from the capillaries of borosilicate glass pipets. The bottom is sealed and loaded with quartz wool, and water is injected into the wool. The top of the pipet is then sealed. For measurements, the entire pipet is then crushed in the crushing unit.

a partitioning of 1:2 between both heads would require non-integer trigger frequencies, which are currently not allowed by the microdrop controller software. To circumvent this problem, the calibration system can be operated at higher gas flow rates and corresponding higher frequencies, or the controller software needs to be modified.

A limitation for field operations is the need to refill the calibration standard reservoirs at regular time intervals. The currently used 12 ml vials hold only about 1/4 of the storage bags of the SDM (45 ml). For longer field deployments, a simple modification would be to use larger glass vials with the same thread size as reservoirs for the DHs. In the current design, manual interventions are needed to restart the dispensing heads, preventing unsupervised operation over prolonged time periods.

    An important limitation of the current setup is the maximum flow rate of about 250–500 sccm. Even larger flow rates
would be needed to characterise the response times of entire inlet lines with additional flush pumps used in semi-permanent

installations for water isotope analysis (e.g., Steen-Larsen et al., 2013; Bonne et al., 2014; Galewsky et al., 2016). While we have not tested such an application here, it is technically fairly straightforward to create higher flow rates by diluting the water vapour stream from the microdrop device with dry carrier gas, for example from a gas tank or a dry air generator, or by modifying the tubing, pressure control, gas heating, and dispensing frequency for larger flow rates. Such an extention of the current design, and corresponding characterisation studies are left for future work.

## 9 Conclusions and implications

Here we describe a new device for the generation of a vapour stream of a specified mixing ratio and water isotope composition based on microdrop dispensing technology. As a key innovation, we operate two dispenser heads in parallel to enable creating a vapour stream with any value along a mixing line between two water standards. We characterise the microdrop vapour generator in terms of the precision of the vapour stream mixing ratio and isotope composition on long and short time scales. Short-term uncertainty of the vapour stream, quantified as Allan deviation, is on the order of 2–10 ppmv for $H_2O$, up to 0.004 ‰ for $\delta^{18}O$ and about 0.02 ‰ for $\delta^2H$ for averaging times of 1000 s. These values are substantially better than a comparable set of measurements obtained with the SDM, which we use as the commercially available benchmark for characterisation of the microdrop device. The long-term precision of 15 min measurement intervals is on the order of 10 ppmv for $H_2O$, 0.10 ‰ for $\delta^{18}O$, and 0.65 ‰ for $\delta^2H$. These estimates are not substantially affected by either flow rate or dispensing frequency. Simultaneous operation of two dispenser heads provides a linear mixing between two standard waters. The general characteristics of the design in terms of response to flow rate and simultaneous operation of the dispenser heads thus demonstrate that the device functions overall according to specifications set forth in Sec 2.

We use the microdrop vapour generator in two application settings. First, we obtain a semi-automated characterisation of the mixing ratio – isotope ratio dependency for an analyser across a wide range of $\delta$ values, more precisely and with lower effort than previously possible. Using a simplified fitting procedure, we obtain a 2-dimensional correction function. Along the $\ln(x)$ axis, a 2nd-order polynomial was sufficient for all four analysers investigated here. Along the $\Delta\delta$ axis, for two of the four analysers a 2nd-order polynomial was sufficient, while the other two had better fitting results with a 3rd-order polynomial. The correction function shows similar overall shape and characteristics for the same analyser over time as in Weng et al. (2020), confirming the stability of this analyser characteristic over months to years. In a second application, we use the microdrop vapour generator in an analytical setup with a crusher device for the analysis of fluid inclusion isotope composition of stalagmite samples. Analysis of standard sample material confirms that the overall analysis works correctly. The high precision of the background humidity from the microdrop vapour generator is a valuable asset in obtaining precise results from the crusher line. Importantly, the availability of a mixing ratio – isotope ratio characterisation for the analyser enables correction of this analyser artefact during peak integration, a factor which has been neglected in previous fluid inclusion studies.

The main advantages of the design are, besides the high precision of the signal, its flexibility in terms of mixing ratio and isotope ratio, absence of moving parts, and low power consumption. Challenges are the occasional stopping due to bubble formation in the dispenser head that requires manual intervention and makes proper de-gassing of the standard liquid imperative.

In addition to a higher degree of automated operation, transfer to a more robust, field-deployable setup in protective housing would clearly be an advantage over the current prototype version. Nonetheless, our successful demonstration of the overall design has several implications for the water vapour isotope measurements: (i) The availability of a source of standard vapour that works at different flow rates, mixing ratios, and stable isotope ratios offers a range of possible applications, from regular calibration and instrument or inlet system characterisation, to more specialised operations as part of an analytical setup in one device. (ii) With the availability of a precise vapour generator, commercial CRDS analysers can be more commonly characterized in terms of the mixing ratio – isotope ratio dependency, contributing to better data quality of vapour measurements, in particular at low water vapour mixing ratios. (iii) The highly precise continuous stream of water vapour from the microdrop device contributes to lower uncertainty during crushing applications on a moist background gas stream.

One aspect that has not been explored here is the characterisation of inlet lines for water vapour isotope measurements. The inlet tubing can affect the isotope composition, for example by memory effects from different materials, depending on flow rates and heating temperatures. Inlet lines could be characterised by providing known pulses of water vapour mixing ratio and isotope composition on the inlet system. This would then enable to optimise the extraction of geophysical signals from measurements. However, substantially larger flow rates of several liters per minute than tested with the current microdrop device are generally required for inlet characterisations. The microdrop device may be able to fill this important gap in water vapour isotope measurements during future follow-up work.

*Code and data availability.* The data set and program code are available from the authors on request.

**Appendix A:  Crusher application details**

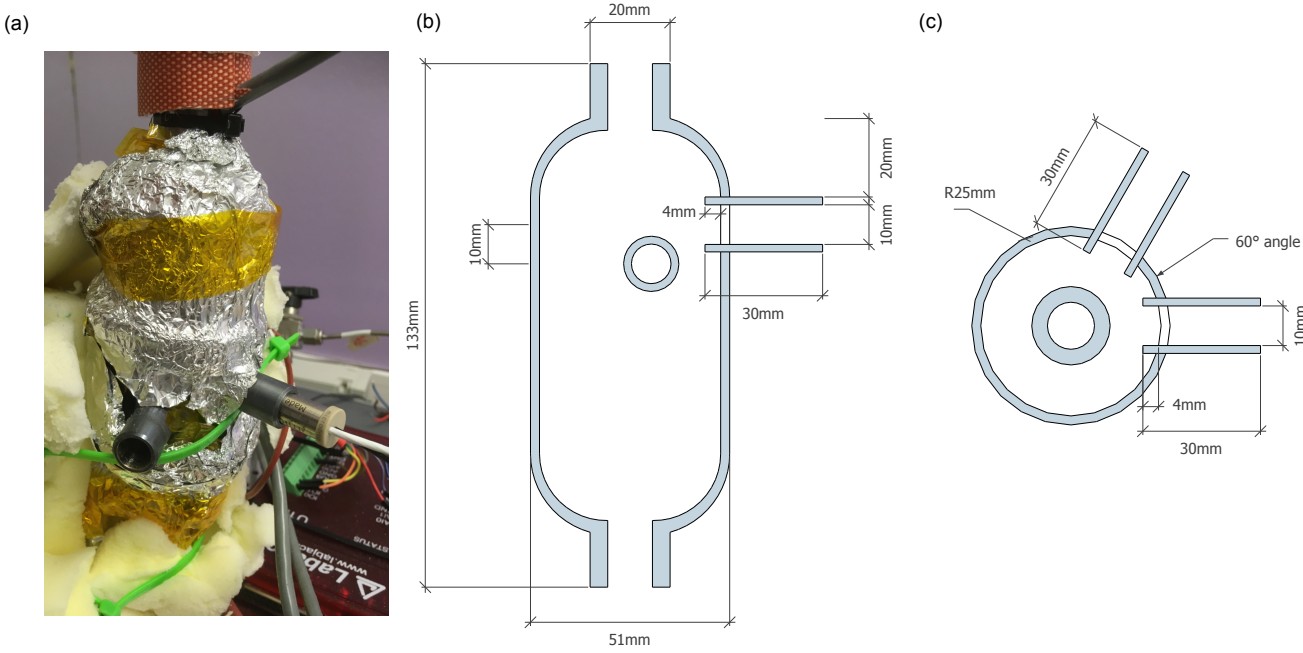

**Figure A1.** Photograph and drawings of the evaporation chamber with dispenser head ports. (a) Photograph of the dispenser head ports on the evaporation chamber with DH1 inserted. Insulating material has been removed to expose the dispenser head ports. Heat tape is covered by aluminium foil. Plan drawing of (b) side view and (c) top view.

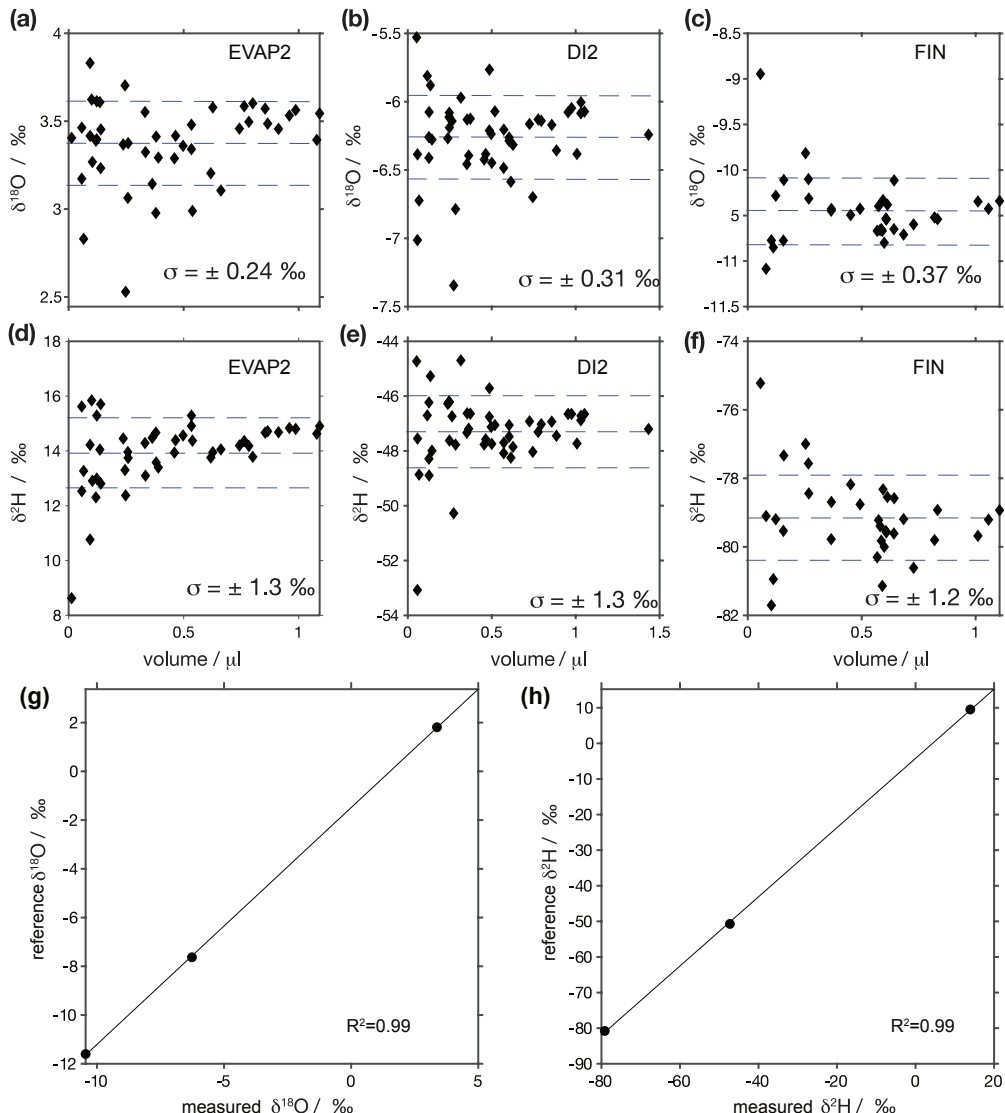

**Figure A2.** Sample waters are calibrated with three inhouse standards EVAP2 (1.77±0.02 for $\delta^{18}$O; 10.0±0.1 for $\delta^2$H), DI2 (-7.64±0.02 for $\delta^{18}$O; -49.8±0.3 for $\delta^2$H), and FIN (-11.66±0.02 for $\delta^{18}$O; 80.8±0.4 for $\delta^2$H) which have previously been normalised to VSMOW-SLAP scale from liquid injections on CRDS analysers using reference waters obtained from IAEA. Panels (a-f) show $\delta^{18}$O and $\delta^2$H values for injections of different volumes. Dashed lines show the mean values and standard deviation across all water volumes, with 1-$\sigma$ standard deviation stated in each panel. Panels (g) and (h) show transfer functions constructed from panels a-f with the mean measured vs. assigned values for $\delta^{18}$O and $\delta^2$H. Note that the y-axes differ between panels.

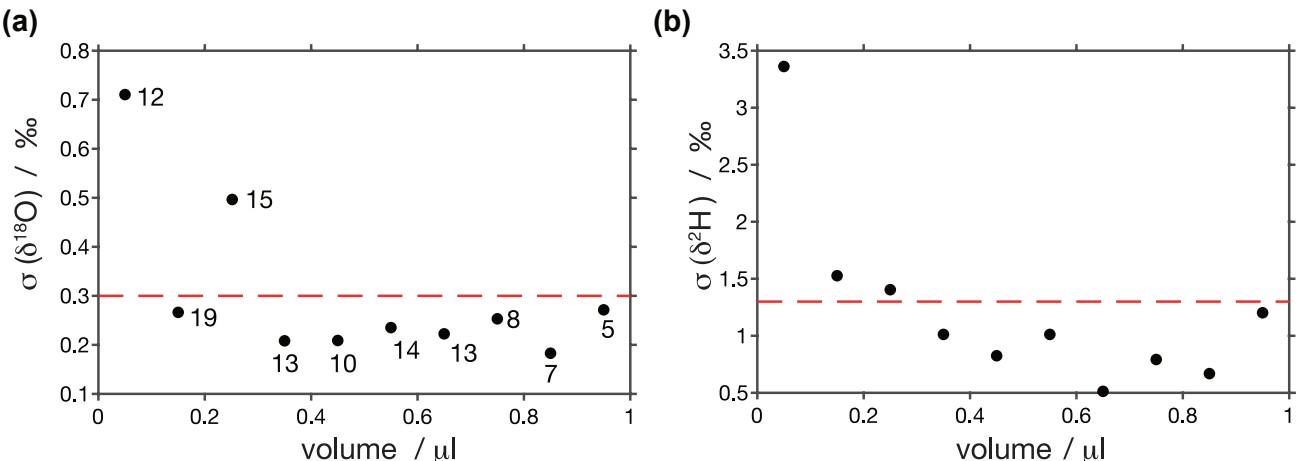

**Figure A3.** Pooled variance of injections calculated every $0.1\,\mu l$. Reproducibilities for samples larger than $0.3\,\mu l$ are $<0.3‰$ for (a) $\delta^{18}O$ and $<1.3\,‰$ for (b) $\delta^{2}H$ and increase for smaller samples. Circles are labelled with number of injections used to calculate pooled variance at each point.

**Table A1.** Date, time, and mixing ratio of the 37 segments used for the short-term stability assessment. Time is the begining of the 15 min interval used to calculate standard deviations in water background and isotope values.

| Date | Time | Mean(H$_2$O) | SD(H$_2$O) | SD($\delta^{18}$O) | SD($\delta^2$H) |
|---|---|---|---|---|---|
| 2020-10-02 | 12:51:07 | 11524 | 29.8 | 0.133 | 0.661 |
| 2020-10-06 | 17:02:47 | 11266 | 13.6 | 0.114 | 0.668 |
| 2020-10-09 | 12:27:45 | 11193 | 26.6 | 0.135 | 0.673 |
| 2020-10-11 | 15:39:44 | 11154 | 10.5 | 0.098 | 0.618 |
| 2020-10-14 | 10:51:25 | 11456 | 10.6 | 0.099 | 0.648 |
| 2020-10-14 | 13:00:50 | 11444 | 4.2 | 0.097 | 0.647 |
| 2020-10-14 | 14:37:03 | 11478 | 4.6 | 0.091 | 0.633 |
| 2020-10-19 | 10:14:18 | 9608 | 10.3 | 0.099 | 0.632 |
| 2020-10-19 | 12:18:44 | 9558 | 5.8 | 0.099 | 0.630 |
| 2020-10-19 | 13:01:35 | 9561 | 3.9 | 0.097 | 0.627 |
| 2020-10-20 | 10:37:04 | 11672 | 6.5 | 0.100 | 0.671 |
| 2020-10-20 | 12:23:13 | 11640 | 14.0 | 0.102 | 0.662 |
| 2020-10-22 | 13:03:44 | 12026 | 10.1 | 0.096 | 0.655 |
| 2020-10-22 | 14:30:00 | 12050 | 13.1 | 0.098 | 0.642 |
| 2020-10-23 | 11:18:42 | 12079 | 7.3 | 0.095 | 0.644 |
| 2020-10-23 | 12:36:16 | 12080 | 9.1 | 0.097 | 0.644 |
| 2020-10-23 | 13:44:41 | 12060 | 8.5 | 0.101 | 0.641 |
| 2020-10-27 | 13:29:00 | 12064 | 11.6 | 0.098 | 0.653 |
| 2020-10-27 | 17:46:53 | 12208 | 8.8 | 0.099 | 0.665 |
| 2020-10-28 | 10:19:02 | 12170 | 9.7 | 0.094 | 0.602 |
| 2020-10-28 | 11:36:25 | 12092 | 12.7 | 0.099 | 0.644 |
| 2020-10-28 | 14:30:23 | 11981 | 10.0 | 0.102 | 0.640 |
| 2020-10-28 | 15:28:51 | 11901 | 11.8 | 0.105 | 0.640 |
| 2020-10-30 | 11:34:02 | 12286 | 18.5 | 0.118 | 0.661 |
| 2020-11-02 | 11:09:05 | 12327 | 11.9 | 0.098 | 0.646 |
| 2020-11-02 | 13:57:04 | 11937 | 17.9 | 0.108 | 0.659 |
| 2020-11-02 | 15:25:24 | 11928 | 23.2 | 0.124 | 0.668 |
| 2020-11-03 | 11:31:10 | 11887 | 24.4 | 0.130 | 0.687 |
| 2020-11-03 | 12:09:20 | 12006 | 21.8 | 0.107 | 0.672 |
| 2020-11-06 | 16:45:40 | 13346 | 5.7 | 0.095 | 0.674 |
| 2020-11-06 | 17:41:55 | 13360 | 6.8 | 0.099 | 0.638 |
| 2020-11-10 | 11:52:19 | 13425 | 12.5 | 0.096 | 0.647 |
| 2020-11-10 | 12:55:17 | 13390 | 4.2 | 0.096 | 0.643 |
| 2020-11-10 | 17:51:03 | 13116 | 6.3 | 0.092 | 0.636 |
| 2020-11-11 | 16:40:10 | 13151 | 6.7 | 0.095 | 0.643 |
| 2020-11-11 | 18:24:41 | 13053 | 5.5 | 0.099 | 0.674 |
| 2020-11-11 | 19:23:35 | 13019 | 8.5 | 0.098 | 0.662 |

*Author contributions.* HS designed and built the microdrop device and contributed to experiments, data analysis, and writing. AD perfomed experiments for analyser characterisation, data analysis, and writing. AFB built the crusher application, performed experiments and data analysis, and contributed to the writing. AS contributed to data analysis and visualisation. JM contributed to experiments, data analysis and interpretation. All authors contributed to the revision of the final submitted manuscript.

*Competing interests.* The authors declare no competing interests.

*Acknowledgements.* The authors acknowledge FARLAB, University of Bergen, Norway for provision of analysers and laboratory space. Enver Alagoz is acknowledged for help with the thermal regulation of the microdrop device. We thank Microdrop GmbH for support regarding the software control of the dispensing heads. Andreas Arp (Microdrop GmbH, Germany) is gratefully acknowledged for a helpful discussion on dispenser head characteristics. AF acknowledges support from Juan de la Cierva Fellowship (IJC2019040065-I) granted by the Spanish Ministry of Science and Innovation and co-funded by the European Development Fund and the European Social Fund. This work was partly funded by the European research Council under the H2020 Work programme (Grant no. 773245) and the Norwegian Research Council (Grant no. 245907, 262353/F20) and its Centres of Exellence funding scheme (Grant no. 262618).

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
