# Peer review of "A flexible device to produce a gas stream with precisely controlled water vapour mixing ratio and isotope composition based on microdrop dispensing technology"

_Atmospheric Measurement Techniques, 2023_

## Referee Comment (RC1)

**General.**

This is a very thorough paper outlining a novel method for dealing with one of the main challenges associated with using laser-based spectroscopy on very small samples- namely the concentration dependence of isotopes. It is a significant body of work that provides a framework for evaluating performance challenges associated with calibrating CRDS systems under a variety of conditions to make the most precise and accurate measurements. The authors provide several applications, including the measurements of inclusions in stalagmites which provides a very concrete example of how this methodology could transform the current methods used for the analysis of fluid inclusions. This paper has significant impacts on the future design and implementation of critical measurements of water vapor isotopes both in the lab and in the field.

Laser-based instruments have provided an enormous step change in our ability to make important measurements in all aspects of the hydrosphere and the atmosphere. But our ability to thoroughly calibrate and account for a number of artifacts, including a mole fraction or concentration dependence for the isotopes; sample carry over; and interaction with surfaces to which the sample is exposed along the measurement route, has remained a challenge. This paper addresses a number of those.

The authors have shown the details of applying existing technology from another field (inkjet printers). While this technology has appeared in previous publications – notably the 2008 paper by St. Clair et al. which is referenced by this paper, this remains to be an important next step in validation and proof of concept for its application with isotopic calibration on small sample sizes. It goes well beyond the scope and purpose of St.Clair (2008).

Key findings include how the micro-dropper device is able to produce a water background of constant isotopic compositions that can be used to obtain accurate and precise $\delta 18$ O and $\delta 2$ H measurements over a variety of concentrations, controlled by parameters of the Dispenser Heads. The inclusion of the comparison to the Standard Delivery Module (SDM, Picarro Inc, Sunnyvale, USA) is a good benchmark and will no doubt be of interest to readers.
The examples of the fluid inclusions would appear to provide validation of methodology that would stand to be an improvement over current methods of measuring fluid inclusions. While a comparison with samples analyzed with mass spectrometer methods would have been useful, the validation provided by the analysis of standards introduced via capillary pipets is convincing.

**Specific**

Line 51: In the introduction, a discussion of similar techniques used to introduce and vaporize liquid water and highlights those used for CFA analyses (Gkinis et al., 2010). To be complete, consider also referencing Jones et al. ,2017 which employs a nebulizer, that solved some of the challenges found in the WVISS system outlined in Aemisegger et al., (2012).

Line 99:  A long with Figure 1, a picture of at least the apparatus in general and/or the critical components (eg. The microdrop dispenser heads; evaporation chamber) would greatly help.

Line 103: With regard to the use of the Microdrop, dispenser head (DH),  JM St. Clair et al., 2008 (Reference on line 575) uses the Microdrop part MD-K-130-010, while the authors chose to use Microdrop GmbH, Germany, Part Nr. MK-K-130.  It could be useful to explain the choice of microdrop dispensers, and whether a heated or non-headed head was used and why.

Line 150:  Clarify "To this end, we first calibrate the raw mixing 150 ratio signal with a calibration curve obtained from a dewpoint hygrometer."  Consider adding "To this end, we first calibrate the raw mixing 150 ratio signal **measured by the Picarro** with a calibration curve obtained from a dewpoint hygrometer."  Also, it would be helpful to name the brand/model dewpoint hygrometer.

Line 154:  Performance characteristics for DH1 and DH2 are mentioned, but not DH3.  Consider mentioning why these two were selected for testing, and not DH3.  Perhaps because these two were the ones chosen for the system, based on performance?  Also, can operating frequencies and dispersion parameters be specified, or are they DH dependent, and constitute unique operating parameters to be determined for each unit?

In the section on the application of this method, it would be useful to have comparisons of this method vs. the traditional extraction/ mass spec method as validation of the method.

Line 332: A significant side bar finding of this study is the statement that " We find that different analyzers require different polynomials for fitting in the $\Delta\delta$ and the ln(x) direction. This may come as a surprise to some adopters of CRDS systems for isotopic measurement and is an important finding.

Line 370:  To my knowledge, Affolter et al., (2014) does not mention memory (or sample carry over) explicitly. So, it could be appropriate to address memory in this paper more directly? Also, does sample carry over affect $\delta^{18}O$ differently than $\delta D$ in this system? Deuterium excess would be an interesting parameter to include if possible.

Line 440:  Long term sustainability and reliability are important qualities of any system like this. Consider expanding upon the statement "While in general working reliably over several hours at a time, these problems are important to be aware of, and one needs to search for remedies in future development of the calibration system".  This is an important step forward. Knowing how long this system can be expected to function reasonably well without manual intervention is useful.  I would encourage authors to speculate on areas where reliability might be improved to potentially lengthen the time it can run unattended.

Line 502.  It is stated that "Prominently, Synflex tubing has been shown to render $\delta2$ H measurements meaningless". This (rather important) conclusion does not appear to be

supported by evidence provided in this manuscript. If this has been shown elsewhere, a reference would help.

**Appendix A: Crusher application details:** Figures A1 and A2 are referred to in the main text, so I am not sure why they are in the appendix, unless there are space limitations?   No details beyond figures are found here.  Consider either moving appendix figures to the main body of the text or changing to "Additional crusher application figures"?  Editorial call.

**Some small Typos or wording:**

Line 17:  needs period at end of sentence.
Line 268 & 276:  Not sure which version AMT prefers:  English version of 'artefact' vs. artifact. Editor's choice?
Line 322:   20'000 should be 20,000.
Line 349:   og -> of

**Figures**

Figure 1.   This schematic explains the set up adequately, but it would help tremendously to add a figure contains images of the micro-drop heads, and/or the evaporation chamber, as this appears to be the heart of the innovation here and deserves some focus.

Figure 2.  These images are very helpful.  Consider adding to the caption how the images were obtained or with what optical imaging system.

Figure 3.   See comment for line 154. Consider adding a note here on why just 2 of the 3 heads were tested.  Perhaps 3 were evaluated and two selected for the vaporizer. In 5 b) there are many fewer points above about 10 Hz for DH1. An explanation would help.

Figure 4.   Both a) and b) show bimodal distribution.  According to the text, the day-to-day standard deviations are shown in red and blue.  Grey histograms are for performance over the entire time period.  The difference is not large (~9 or 10μ) but groupings are distinctly different. An explanation of these groupings might help.  Perhaps it is due to the physics of discrete droplet formation that the volume (or diameter) of droplets is not a continuum, and is constrained by the droplet formation process itself?  This might be obvious to those familiar with the DH technology, but a sentence of explanation would help those who are not.

Figure 5. Allen deviation plots are very helpful, as is the comparison with the SDM.

Figure 8.  This is an important figure.  Consider adding a panel o

Fig. 11:  This is an excellent and key figure, showing the comparison of correction functions of Weng et al. (2020). The caption refers to dashed grey lines in the figure – but there are no

dashed lines as mentioned as far as I can tell, but there arepossibly blue and black lines? (I am thinking the caption may have been created with a previous version of the plots?) Either make the colors more distinct, or make one set of lines dashed, as the caption reads.

Figure 13. Excellent figure. IF I understand this method correctly, peak values are integrated between the green markers, which show the start and the end of the peak. For both $\delta^{18}O$ and $\delta D$ there is a significant drop (lighter isotopically) just as the peak in mixing ratio arrives in the analyzer. Presumably, integration between the green markers (including this dip) yields the correct value for water standards introduced with capillary tubes? Maybe I missed something about the calibration? Based on the performance of the analyzer as shown in figure 9, It would seem that integration of the sample might be more stable later? Maybe sentence in the text or caption could address this.

Figure 14. Image of glass pipets is helpful. Presumably the glass pipets are introduced in the crusher unit, if so, this could be mentioned in the caption.

Figure A1. Caption could be improved by listing the names of the internal (or inhouse, secondary) standards and their assigned values with ± uncertainty, and explanation of the analysis method used to assign their values to the VSMOW-SLAP scale. Perhaps adding "note y-axis scales differ" would also help.

1. Does the paper address relevant scientific questions within the scope of AMT?  YES
2. Does the paper present novel concepts, ideas, tools, or data? YES
3. Are substantial conclusions reached?  YES
4. Are the scientific methods and assumptions valid and clearly outlined? YES
5. Are the results sufficient to support the interpretations and conclusions? YES
6. Is the description of experiments and calculations sufficiently complete and precise to allow their reproduction by fellow scientists (traceability of results)? YES, with some caveats as noted in comments
7. Do the authors give proper credit to related work and clearly indicate their own new/original contribution? YES, mostly except where noted.
8. Does the title clearly reflect the contents of the paper? YES
9. Does the abstract provide a concise and complete summary? YES
10. Is the overall presentation well-structured and clear? YES
11. Is the language fluent and precise? YES
12. Are mathematical formulae, symbols, abbreviations, and units correctly defined and used? YES
13. Should any parts of the paper (text, formulae, figures, tables) be clarified, reduced, combined, or eliminated? YES, minor edits are recommended, but nothing major.
14. Are the number and quality of references appropriate? Generally, YES
15. Is the amount and quality of supplementary material appropriate? YES

---

## Referee Comment (RC2)

**Review of "A flexible device to produce a gas stream with precisely controlled water vapour mixing ratio and isotope composition based on microdrop dispensing technology"**

This paper presents a new device based on two microdrop dispenser heads that allows to inject a controlled amount of water with known isotopic composition into a heated evaporation chamber flushed by a dry carrier gas for subsequent analysis with a laser spectrometer. The authors argue that the novelty of this device is its flexibility in terms of producing air samples with a range of different water vapour mixing ratios and isotope compositions varying between the two end-members of the liquid water samples feeding the two dispenser heads. The authors show two applications of the new device: 1) the detailed characterisation of the well-known water vapour mixing ratio dependence of isotope measurements with laser spectrometers and 2) the production of a stable background water isotope signal for the precise measurement of microfluid inclusions in stalagmites.

The paper is interesting and well-illustrated. The major innovation is the combination of two dispenser heads instead of one, which gives more flexibility in the generation of water vapour samples with a range of different isotope compositions and water vapour concentrations. While most of the text is well-written and easy to follow, there are a few parts that are written in a misleading way, where I suggest that the reader guidance and accuracy of the description could be improved (see my minor comments below).

The paper should be published in AMT after the following two major points and a longer list of minor comments as well as a few technical aspects have been addressed:

M1: Given that the key innovation of the proposed system is based on the combination of two dispenser heads for being able to produce a series of gas streams that cover a range of different water vapour mixing ratios and isotope compositions, a solid assessment of the uncertainty of the predicted reference properties ([$H_2O$], $\delta^2H$, $\delta^{18}O$) should be provided. Currently, this aspect is very difficult to assess. From Fig. 8 it seems that the RMSE of the obtained mixtures of standards is very large and therefore the system's characteristic uncertainty in the reference isotope composition and mixing ratio when using the two dispenser heads simultaneously seems to me an important issue.

M2: I am not convinced that the relatively complex water vapour mixing ratio – isotope dependency correction is really needed in the second application for micro fluid inclusions in stalagmites, because the water vapour mixing ratio of the background airstream is well above 10'000 ppmv and the variations due to the microfluid injection peak is in a range of values where this correction is not usually needed. Could the authors show the impact of this correction on their aliquot of a natural carbonate and/or their samples used to mimic real fluid inclusions? It's a major point because the authors argue that this is an important additional processing step that was neglected in the previous literature (L. 430-437).

Minor comments:

1) L. 1, L. 22 and at other places: "a flow of **air-vapour** mixture" is a bit confusing: not clear what vapour of which gas and air does a priori comprehend water vapour as well. The terminology used in the title seems more precise to me: "a gas stream of a pre-determined water vapour mixing ratio and isotope composition".

2) L. 6: "water vapour mixing ratio", just mixing ratio can be any gas or isotope.

3) Abstract: the abstract is a bit short and lacks key information on the new device. It would be more informative to add a few key numbers in the abstract such as the range of flow rates tested, explain what limits the flow rates in the device (key for more recent fast-response CRDS instruments operated at high flow rates). The range of water vapour mixing ratios and isotope delta values tested as well as the quality of the delivered calibration or background gas stream could be mentioned (precision, stability).

4) L. 15: I think Graf et al. 2019 is a nice example for subcloud processes involved during precipitation but not weather systems in general. Thurnherr et al. 2020 cover a broad range of latitudes and different types of weather systems, which would be a better fit here.

5) L. 16: "analyser properties" is a bit unspecific, be more precise.

6) L. 17: "the variability of the calibration system", what does that mean?

7) L. 15-19: Total uncertainty resulting from a variety of instrument characteristics at different water vapour mixing ratios and the characteristics of inlet systems were tested extensively in the two early publications Sturm and Knohl, 2010; and Aemisegger et al. 2012. Application in the field with aircraft-based measurements (Sodemann et al. 2017) and for near-surface humidity gradients (Seidl et al. 2023) were carried out in the more recent ones. This is a bit misleadingly written and should become clearer from this section of the text.

8) L. 29ff: I am not convinced that the microdrop system that is proposed in this paper alleviates all the mentioned problems: given the small liquid reservoir, the isotope composition of the standard liquid in the glass vials should be monitored as well for a reliable normalisation to the VSMOW-SLAP scale. What would be more convincing to me at this stage is to mention primarily that in the bubbler system the reference isotope composition of the vapour phase has to be predicted from equilibrium fractionation, which requires precise temperature regulation of the liquid phase in the bubbler. This problem is overcome in other devices by complete evaporation. Monitoring the liquid isotope composition is a must also in systems with complete evaporation including an SDM standard bag. When using large water reservoirs of several litres the close-up monitoring of the liquid is not so essential in bubblers because the changes are much smaller than the precision of the reference measurement with IRMS or laser spectrometry.

9) L. 50-51: "limited possibility to regulate the mixing ratio of the water vapour in the airflow", which is not really needed for the mentioned application.

10) L. 64-65: The mention of flow rates in between the water vapour mixing ratio range and the different isotope composition is disturbing the logical flow of information in

the reader's mind: Maybe something like the following would help: "… that provides the combination of a precise stream of water vapour across a range of water vapour mixing ratios and isotope compositions as well as operation at various flow rates between 50 and 250 sccm." -> group the range of mixing ratios and isotope compositions together, flow rates is a different aspect also from an operational perspective.

11) L. 71: "and as a component in specific laboratory applications" this is a bit unspecific. Please be more concrete.

12) L. 73: Here it would be very nice to be more precise about the gas stream of a preset background humidity and isotope composition for fluid inclusion analysis. A reader who is unfamiliar with this technique is lost here (see also my minor comment 53).

13) L. 75-89: This paragraph is very dense with information and a bit difficult to follow. The first sentence starts with introducing the advantage of the microdrop generator in covering "a wider range of applications with CRDS analysers" than with currently available devices. But there are a countable number of applications, and the currently available devices are optimised for the tasks at hand. I rather think the advantage of the microdrop generator is the flexibility of it to produce a wide range of (predetermined) water vapour mixing ratios and isotope compositions. The application to microfluid inclusions in stalagmites is nice but this high flexibility is actually not needed there.

14) L. 77: Why are the authors so strict about the water vapour mixing ratio uncertainty? Is this useful in one of their applications? I thought they target the isotope composition of different vapour/liquid/fluid inclusion samples and not the absolute water concentration.

15) L. 81: I believe the first publications with fast-response analysers with flow rates above 100 sccm are Thurnherr et al. 2020 ACP with a detailed assessment of the impact of the flow rate on the water vapour mixing ratio dependence on isotope measurements in their supplement and Bailey et al. 2023 ESSD. From these publications with flow rates of about 300 sccm through the cavity it becomes clear that flow rate limitations of calibration systems is an important challenge.

16) L. 82: This sentence should be removed or moved to the discussion in the conclusions (see also my minor comment 74), the response times of the inlet is not tested in this publication and cannot be tested with the proposed microdrop dispenser because the flow rates are much too small (would need several liters per min to be of interest for this sort of application).

17) L. 88: The requirement of being field compatible seems ambitious and I don't fully understand it. How well do the dispenser heads work in very cold environments with vibrations and at low pressure? In principle drift correction in the field could be done with simpler existing systems, the advantage of this system resides in the possibility to characterise the water vapour mixing ratio – isotope ratio dependency in detail. This is more a laboratory than a field application.

18) L. 92: SDMs can go well below 6000 ppmv.

19) L. 94: What does "Injections with the autosampler are feasible with a specific method" mean?

20) L. 95: This joins my major comment M1: unless you are able to very precisely predict the isotope composition of your mixture with the two dispenser heads you cannot really overcome this limitation from other systems.

21) L. 101: this doesn't become clear from the schematic in Fig. 1.

22) L. 103: this is a very small amount of water which can be rapidly impacted by evaporative enrichment in the field!

23) L. 106: "retention time of water vapour" are interactions with the chamber walls meant here? If yes, to me "memory effect" would be clearer.

24) L. 140: how robust are the voltage and frequency settings and their associated efficiency to prolonged usage?

25) L. 148: "We can used this relation between drop size and mixing ratio to compute the effective drop size…" I am a bit lost here. Do you mean the relation between the frequency and the mixing ratio can be used to estimate the effective drop size?

26) L. 154 and Fig. 3: is it only my print out that make the curves in Fig. 3 not look linear?

27) Fig. 3b: Why is the H2O mixing ratio coverage worse with the settings for panel b than in panel a?

28) L. 161: This seems to be key, indeed. Can the uncertainty from this fitting procedure be quantified in ppmv for a series of runs? And how frequently does this readjustment of dispensing parameters need to be done?

29) Fig. 4: I am not sure that I really understand the meaning of this figure. What does all dispersion parameters mean? Could this Figure be placed a bit more logically in the discussion. A part from a useful diagnostic, I don't see the use of it? Is it possible that only showing the blue and red distribution would help the reader because these are the selected settings. In general, I am not sure that the reader is interested in the full distribution with all your tested dispersion parameter setting.

30) L. 192: here and elsewhere: it is "Allan" variance not "Allen" and the adequate reference would be Allan, 1966 Allan, D. W.: Statistics of atomic frequency standards, Proc. IEEE, 54, 221–230, doi:10.1109/PROC.1966.4634, 1966.

31) Eq. 1: I think the authors mean $y_{i+1}(\tau)$ not $y_i + 1(\tau)$.

32) L. 206: The SDM is optimised for the precision of the $\delta^{18}O$ and $\delta^2H$ not for the mixing ratio. For a fair comparison with a high quality mixing ratio signal, a dew point generator should be used.

33) L. 225: this is understandable only if the basic concept of fluid inclusion peak measurements on a stable background has been explained in the intro. Maybe a reference to Section 7 could also be added here.

34) L. 254: "The linearity of this mixing is provided over a range of mixing ratios". I can't really evaluate this statement based on Fig. 8a. How about coloring the data points with the mixing ratio?

35) L. 255: "At lower humidity than 5000 ppmv deviations from the linear mixing become apparent" Also here, I can't really see this in Fig. 8b, which shows the RMSE of $\delta^{18}O$ and $\delta^2H$ from the observed mixing line at different water vapour mixing ratios. The RMSE is by the way very large implying that the predicted reference isotope

composition of the gas stream has a large uncertainty making it very difficult to me to effectively these mixed samples as reference calibration samples.

36) L. 260: what is Mix000 and Mix 100? This is the first time these labels appear.

37) L. 260: Where can I see this bending down (please add a reference to the figure that is meant here).

38) L. 263: the phase space you are showing in Sec. 6 is the correction function in the ln(mixing ratio) vs. $\delta^2$H (or $\delta^{18}$O) phase space.

39) L. 265: the microdrop vapour generator was already connected to the analyser for all the evaluations presented in the previous section. Here you switch from characterising the microdrop vapour generator to using it to characterise CRDS analysers in their water vapour mixing ratio and isotope composition dependency.

40) L. 267: The water vapour mixing ratio and isotope composition dependency is of spectroscopic origin and due to uncertainties associated with the baseline. Even if a system is optimised for measurements at lower humidity levels than 20'000 ppmv this problem occurs.

41) L. 273: This is misleading. The spectroscopic origin of the concentration dependency has been mentioned already in early laser spectrometric instrument evaluation work such as from Sturm and Knohl, 2010 and Iannone et al. 2009.

42) L. 274: matrix gas -> carrier gas

43) L. 296-308: This is very difficult to follow, could the respective sequences be clearly indicated in Fig. 9? This paragraph should be in stronger dialogue with what we see in Fig. 9 for the reader to be able to follow.

44) L. 313: Here I got lost, where do I have to look above or below?

45) L. 315: Can the authors be more precise about the number of outliers they filtered out? This is an important quality measure for the microdrop generator. Because an ideal calibration system needs as little instrument measurement time as possible so minimising the occurrence of "outliers" is a key aim when designing a robust calibration system.

46) L. 320: What is the share of new data points from the microdrop generator vs. the data points from Weng et al. (2020) in this analysis shown in Fig. 10? Could the data points from the microdrop generator be highlighted in a different color or shape?

47) L. 331: the delta value is specified und used in equations 2 and 3.

48) L. 336: this final step was a bit obscure to me. I thought that the correction function is 0 per definition at the reference mixing ratio level?

49) Table 3: the RMSE is most likely very dependent on the mixing ratio? Maybe a Figure showing this dependence as a function of mixing ratio would be more informative.

50) L. 350-355: this paragraph was again not clear to me. Which subsample of available measurements?

51) Fig. 11: The vertical axis is not the raw but the reference $\delta^2$H and $\delta^{18}$O, right? There are no dashed grey lines (at least not in my print out) and what are the blue lines? Would this figure be better readable if it was shown in this format in the appendix (to illustrate the limited concentration effects at high water vapour mixing ratios) and with

a zoom in to 0-5000 ppmv (which is the interesting part) in the paper? Right now one does not see much about the important effects at low water vapour mixing ratios.

52) L. 357: "a constant vapour stream" with known water vapour mixing ratio and isotope composition (very important!)

53) L. 357-366: This information should already be provided in the introduction.

54) L. 369: "memory effects can be removed entirely" is misleading and a bit optimistic. Isn't the advantage of the CRDS technique mainly the ability to measure $\delta^2$H and $\delta^{18}$O (possibly even d17O) quasi-simultaneously with much reduced sample preparation effort?

55) L. 372: "higher throughput and better reproducibility" than what?

56) L. 384: Fig. A2 -> Fig. 12?

57) L. 387-393: connect this paragraph better to Fig. 13, they should be in better text-illustration dialogue to help the reader follow.

58) L. 395: As mentioned in M2 I wonder about the necessity of this step given the high mixing ratio of the background air stream.

59) L. 397: "Mixing ratio -isotope ratio corrections…" this has already been mentioned in point i) just above and appears as an unnecessary repetition here.

60) L. 399: what is meant by "size-$\delta$" relationship?

61) L. 405-414: I got lost in this paragraph. As the authors write at L. 410 this appears redundant. Fig. A1 is not very convincing in showing a consistent dependency of the isotope signals on the injected amount.

62) Fig. A1: To which mixing ratio levels do these injection volumes correspond?

63) L. 407: "drift in values", what do you mean by values?

64) L. 412: the last sentence of this paragraph "Finally,…" is particularly obscure to me. Could the authors illustrate this with a figure?

65) L. 431: Here I am a bit puzzled: the authors write that water vapour mixing ratio – isotope dependency corrections are necessary below 5000-10'000 ppmv (L. 270). But the background gas stream can be designed in such a way that the mixing ratio is larger than this and ideally even close to the optimal water vapour mixing ratio operation level of 20'000 ppmv? So why does this correction matter? Can the authors show how different the isotope signal estimates get when applying vs. when not applying the water vapour mixing ratio correction for their aliquot or the samples used to mimic real fluid inclusions?  This is also my major point M2.

66) L. 432: "more correct" than what?

67) L. 434: Why is the precision of the water vapour mixing ratio mentioned here? Isn't what matters for precise fluid inclusion measurements the precision of the isotope signals?

68) L. 455: for which application would horizontally mounted DHs be an interesting option?

69) L. 462: Flow rates between 300 and 600 sccm through the cavity are now used in different applications (aircraft-based and flux measurements) and present many

advantages (e.g. faster response times, better signal to noise ratio at shorter averaging times) but this range of flow rates cannot be covered the microdrop dispenser system. Why is that so, and is an extension to these higher flow rates possible with the given operational range of dispenser head triggering frequency?

70) L. 475: remind the reader that the Picarro SDM is the commercially available benchmark used for characterisation of the microdrop dispenser.

71) L. 475: In the long-term precision (relevant for calibration) the microdrop dispenser is not better than the SDM for the isotope signals (which we are interested in). The dispenser is better than the SDM for water vapour mixing ratio, for which however a dew point generator is needed anyway for precise calibration. Together with the many small problems that occur with air bubbles in the dispenser head, the small amount of liquid that needs to be exchanged regularly, this questions the utility of this system for regular field calibrations. It is very useful for an in-depth characterisation of the water vapour mixing ratio – isotope signal dependency in the lab but apparently not so much for prolonged measurements in the field.

72) L. 477: I don't understand what this sentence implies scientifically. Which specifications are meant here? The dispenser head and mass flow controller specifications or the objectives set by the authors for the system in Section 2? Maybe a reference to section 2 could be added here.

73) L. 495: Could this simply be a numbered list of implications? It would be easier to read.

74) L. 502: Here is the place where the sentence from L. 82 could be brought in: larger flow rates are needed to test the inlet system and the overall response time of different water vapour isotope measurement field setups.

75) L. 502: Again this is misleading, response times of laser spectrometric water vapour isotope measurements were not the topic of this publication. If one just reads the conclusions, this sounds like a result from this study. References to the studies Sturm and Knohl, 2010 AMT (tested Synflex tubings in the lab) and Tremoy et al. 2014 JGR (tested Synflex tubings in the field) should be made.

Technical points:

1) L. 17: **.** Currently
2) L. 67: technology, **which** allows
3) L. 75: "water" vapour
4) L. 76-80: Grammatically and for keeping the reader's attention having (ii) in the same sentence as (i) would be much more convenient.
5) L. 103-104: "one 12 ml glass vial that holds a liquid water standard and which is mounted next to …". It is not the evaporation chamber that holds the liquid water standard, right?
6) L. 161: demonstrates
7) L. 161: "…  suitable dispensing parameters"
8) L. 214: microdrop
9) Caption Fig. 4: dispening -> dispersion
10) L. 348: og -> of

11) L. 369: CDRS -> CRDS

12) L. 464: visco~u~sity

13) L. 509: perfo~r~med

14) P. 27, Fig. A1: is "assigned" really the term you want to use here on the y-axis? To me this seems to be the "reference".

---

## Author Comment (AC1)

General.

This is a very thorough paper outlining a novel method for dealing with one of the main challenges associated with using laser-based spectroscopy on very small samples- namely the concentration dependence of isotopes. It is a significant body of work that provides a framework for evaluating performance challenges associated with calibrating CRDS systems under a variety of conditions to make the most precise and accurate measurements. The authors provide several applications, including the measurements of inclusions in stalagmites which provides a very concrete example of how this methodology could transform the current methods used for the analysis of fluid inclusions. This paper has significant impacts on the future design and implementation of critical measurements of water vapor isotopes both in the lab and in the field.

Laser-based instruments have provided an enormous step change in our ability to make important measurements in all aspects of the hydrosphere and the atmosphere. But our ability to thoroughly calibrate and account for a number of artifacts, including a mole fraction or concentration dependence for the isotopes; sample carry over; and interaction with surfaces to which the sample is exposed along the measurement route, has remained a challenge. This paper addresses a number of those.

The authors have shown the details of applying existing technology from another field (inkjet printers). While this technology has appeared in previous publications – notably the 2008 paper by St. Clair et al. which is referenced by this paper, this remains to be an important next step in validation and proof of concept for its application with isotopic calibration on small sample sizes. It goes well beyond the scope and purpose of St.Clair (2008).

Key findings include how the micro-dropper device is able to produce a water background of constant isotopic compositions that can be used to obtain accurate and precise $\delta18$ O and $\delta2$ H measurements over a variety of concentrations, controlled by parameters of the Dispenser Heads. The inclusion of the comparison to the Standard Delivery Module (SDM, Picarro Inc, Sunnyvale, USA) is a good benchmark and will no doubt be of interest to readers.

The examples of the fluid inclusions would appear to provide validation of methodology that would stand to be an improvement over current methods of measuring fluid inclusions. While a comparison with samples analyzed with mass spectrometer methods would have been useful, the validation provided by the analysis of standards introduced via capillary pipets is convincing.

We thank the reviewer for their thorough feedback and the constructive comments.

Specific

Line 51: In the introduction, a discussion of similar techniques used to introduce and vaporize liquid water and highlights those used for CFA analyses (Gkinis et al., 2010). To be complete, consider also referencing Jones et al. ,2017 which employs a nebulizer, that solved some of the challenges found in the WVISS system outlined in Aemisegger et al., (2012).

Thank you, we will add this useful reference to the introduction

Line 99: A long with Figure 1, a picture of at least the apparatus in general and/or the critical components (eg. The microdrop dispenser heads; evaporation chamber) would greatly help.

We will add photographs to either Fig. 1 or an appendix figure.

Line 103: With regard to the use of the Microdrop, dispenser head (DH), JM St. Clair et al., 2008 (Reference on line 575) uses the Microdrop part MD-K-130-010, while the authors chose to use Microdrop GmbH, Germany, Part Nr. MK-K-130. It could be useful to explain the choice of microdrop dispensers, and whether a heated or non-headed head was used and why.

As far as we currently know, the additional part number of St. Clair refers to the nozzle size. We will clarify and update the description accordingly.

Line 150: Clarify "To this end, we first calibrate the raw mixing ratio signal with a calibration curve obtained from a dewpoint hygrometer." Consider adding "To this end, we first calibrate the raw mixing ratio signal measured by the Picarro with a calibration curve obtained from a dewpoint hygrometer." Also, it would be helpful to name the brand/model dewpoint hygrometer.

We will modify this sentence as suggested and add information about the dewpoint hygrometer (Optisonde, GE Inc., USA)

Line 154: Performance characteristics for DH1 and DH2 are mentioned, but not DH3. Consider mentioning why these two were selected for testing, and not DH3. Perhaps because these two were the ones chosen for the system, based on performance? Also, can operating frequencies and dispersion parameters be specified, or are they DH dependent, and constitute unique operating parameters to be determined for each unit?
In the section on the application of this method, it would be useful to have comparisons of this method vs. the traditional extraction/ mass spec method as validation of the method.

DH3 was essentially purchased at a later date and used as a spare. We will provide more details about the choice of dispensing parameters, which are indeed unit dependent, but non-unique. We understand the potential value of a comparison to traditional methods, but do not currently have an extraction line available. instead, we use the glass capillaries as a validation method.

Line 332: A significant side bar finding of this study is the statement that " We find that different analyzers require different polynomials for fitting in the $\Delta\delta$ and the $\ln(x)$ direction. This may come as a surprise to some adopters of CRDS systems for isotopic measurement and is an important finding.

We consider making this finding (which has already in principle been observed in several other studies) more prominent in the Conclusions.

Line 370: To my knowledge, Affolter et al., (2014) does not mention memory (or sample carry over) explicitly. So, it could be appropriate to address memory in this paper more directly? Also, does sample carry over affect d18O differently than dD in this system? Deuterium excess would be an interesting parameter to include if possible.

This is a very interesting comment. We will consider adding d-excess to Fig. 13 with a corresponding discussion of memory.

Line 440: Long term sustainability and reliability are important qualities of any system like this. Consider expanding upon the statement "While in general working reliably over several hours at a time, these problems are important to be aware of, and one needs to search for remedies in future development of the calibration system". This is an important step forward. Knowing how long this system can be expected to function reasonably well without manual intervention is useful. I would encourage authors to speculate on areas where reliability might be improved to potentially lengthen the time it can run unattended.

We will expand upon this statement in the discussion section. Since submission of the manuscript, we already made important progress in identifying a key limiting parameter affecting the duriation of unattended operation, namely the amount of dissolved gas in the dispensed liquid. Proper degassing of the calibration liquid ensures substantially extended operation times, in practice advancing from hours to days of continuous operation.

Line 502. It is stated that "Prominently, Synflex tubing has been shown to render $\delta2\,H$ measurements meaningless". This (rather important) conclusion does not appear to be supported by evidence provided in this manuscript. If this has been shown elsewhere, a reference would help.

We will include a reference to Tremoy et al., 2009, which was by mistake dropped from the submitted draft manuscript.

Appendix A: Crusher application details: Figures A1 and A2 are referred to in the main text, so I am not sure why they are in the appendix, unless there are space limitations? No details beyond figures are found here. Consider either moving appendix figures to the main body of the text or changing to "Additional crusher application figures"? Editorial call.

We will consider including these figures in the main manuscript, which may indeed be more appropriate and allow for better readability.

Some small Typos or wording:

Line 17: needs period at end of sentence.
Line 268 & 276: Not sure which version AMT prefers: English version of 'artefact' vs. artifact. Editor's choice?
Line 322: Line 349:
Figures
20'000 should be 20,000. og -> of
This schematic explains the set up adequately, but it would help tremendously to add

We will correct these typos in the revised manuscript.

Figure 1.
a figure contains images of the micro-drop heads, and/or the evaporation chamber, as this appears to be the heart of the innovation here and deserves some focus.

As mentioned above, we will add photographs to either Fig. 1 or an appendix figure.

Figure 2. These images are very helpful. Consider adding to the caption how the images were obtained or with what optical imaging system.

We will include in the caption how these photographs were obtained (high-frequency camera available from Microdrop GmbH).

Figure 3. See comment for line 154. Consider adding a note here on why just 2 of the 3 heads were tested. Perhaps 3 were evaluated and two selected for the vaporizer. In 5 b) there are many fewer points above about 10 Hz for DH1. An explanation would help.

DH3 was essentially purchased at a later date and used as a spare. We will update the text to expain the number of points for each DH.

Figure 4. Both a) and b) show bimodal distribution. According to the text, the day-to-day standard deviations are shown in red and blue. Grey histograms are for performance over the entire time period. The difference is not large (~9 or 10µ) but groupings are distinctly different. An explanation of these groupings might help. Perhaps it is due to the physics of discrete droplet formation that the volume (or diameter) of droplets is not a continuum, and is constrained by the droplet formation process itself? This might be obvious to those familiar with the DH technology, but a sentence of explanation would help those who are not.

This distinct grouping is indeed a consequence of the DH geometry, which allows to dispense stable drops of a particular size for each head.

Figure 5. Allen deviation plots are very helpful, as is the comparison with the SDM. Figure 8. This is an important figure. Consider adding a panel o

Unfortunately, it appears that part of the reviewer comment went missing.

Fig. 11: This is an excellent and key figure, showing the comparison of correction functions of Weng et al. (2020). The caption refers to dashed grey lines in the figure – but there are no dashed lines as mentioned as far as I can tell, but there arepossibly blue and black lines? (I am thinking the caption may have been created with a previous version of the plots?) Either make the colors more distinct, or make one set of lines dashed, as the caption reads.

We will correct the caption/lines in the figure, and redraw the data points acquired by Weng et al. (2020) in a different color.

Figure 13. Excellent figure. If I understand this method correctly, peak values are integrated between the green markers, which show the start and the end of the peak. For both d18O and dD there is a significant drop (lighter isotopically) just as the peak in mixing ratio arrives in the analyzer. Presumably, integration between the green markers (including this dip) yields the correct value for water standards introduced with capillary tubes? Maybe I missed something about the calibration? Based on the performance of the analyzer as shown in figure 9, It would seem that integration of the sample might be more stable later? Maybe sentence in the text or caption could address this.

We will carefully revise this section to clarify the calibration and data processing.

Figure 14. Image of glass pipets is helpful. Presumably the glass pipets are introduced in the crusher unit, if so, this could be mentioned in the caption.

Yes, this information will be added to the caption.

Figure A1. Caption could be improved by listing the names of the internal (or inhouse, secondary) standards and their assigned values with ± uncertainty, and explanation of the analysis method used to assign their values to the VSMOW-SLAP scale. Perhaps adding "note y- axis scales differ" would also help.

We will add the names and assigned values of our internal lab standards here, and include a note on the difference of y-axis scales.

---

## Author Comment (AC2)

**Review of "A flexible device to produce a gas stream with precisely controlled water vapour mixing ratio and isotope composition based on microdrop dispensing technology"**

This paper presents a new device based on two microdrop dispenser heads that allows to inject a controlled amount of water with known isotopic composition into a heated evaporation chamber flushed by a dry carrier gas for subsequent analysis with a laser spectrometer. The authors argue that the novelty of this device is its flexibility in terms of producing air samples with a range of different water vapour mixing ratios and isotope compositions varying between the two end-members of the liquid water samples feeding the two dispenser heads. The authors show two applications of the new device: 1) the detailed characterisation of the well-known water vapour mixing ratio dependence of isotope measurements with laser spectrometers and 2) the production of a stable background water isotope signal for the precise measurement of microfluid inclusions in stalagmites.

The paper is interesting and well-illustrated. The major innovation is the combination of two dispenser heads instead of one, which gives more flexibility in the generation of water vapour samples with a range of different isotope compositions and water vapour concentrations. While most of the text is well-written and easy to follow, there are a few parts that are written in a misleading way, where I suggest that the reader guidance and accuracy of the description could be improved (see my minor comments below).

We are very greatful to the reviewer for their attentive reading of our manuscript, and the detailed and specific constructive comments.

The paper should be published in AMT after the following two major points and a longer list of minor comments as well as a few technical aspects have been addressed:

M1: Given that the key innovation of the proposed system is based on the combination of two dispenser heads for being able to produce a series of gas streams that cover a range of different water vapour mixing ratios and isotope compositions, a solid assessment of the uncertainty of the predicted reference properties ([H2O], $\delta^2H$, $\delta^{18}O$) should be provided. Currently, this aspect is very difficult to assess. From Fig. 8 it seems that the RMSE of the obtained mixtures of standards is very large and therefore the system's characteristic uncertainty in the reference isotope composition and mixing ratio when using the two dispenser heads simultaneously seems to me an important issue.

This is a very valuable comment, we agree that the aspect of the uncertainty of the mixtures is important in the context of this device. We will add a section/figure to make this aspect more readily accessible in the revised manuscript.

M2: I am not convinced that the relatively complex water vapour mixing ratio – isotope dependency correction is really needed in the second application for micro fluid inclusions in stalagmites, because the water vapour mixing ratio of the background airstream is well above 10'000 ppmv and the variations due to the microfluid injection peak is in a range of values where this correction is not usually needed. Could the authors show the impact of this correction on their aliquot of a natural carbonate and/or their samples used to mimic real fluid inclusions? It's a major point because the authors argue that this is an important additional processing step that was neglected in the previous literature (L. 430-437).

Fluid inclusions contain a variable amount of liquid in the samples. Therefore, the mixing ratio in the background air stream has to be adjusted, and can not always be at 10'000 or even 20'000 ppmv, but may have to be for example as low as 5000 ppmv. With the strong variations that occur during the crushing procedure in mixing ratio, there can indeed be where a significant impact in the mixing ratio - isotope ratio dependency. Furthermore, the stability of the mixing ratio is important for this application to separate the background water vapour clearly from the water vapour released from the crushed sample. We will consider the best way to provide more support to this point, either from acutal sample processing, or from a simulation of the impact for hypothetical samples.

Minor comments:

1) L. 1, L. 22 and at other places: "a flow of air-vapour mixture" is a bit confusing: not clear what vapour of which gas and air does a priori comprehend water vapour as well. The terminology used in the title seems more precise to me: "a gas stream of a pre- determined water vapour mixing ratio and isotope composition".

We will clarify the terminology upfront for the remainder of the manuscript in order to be both precise and avoid excessive wordiness,

2) L. 6: "water vapour mixing ratio", just mixing ratio can be any gas or isotope.

We will correct this according to the reply to comment 1) above.

3) Abstract: the abstract is a bit short and lacks key information on the new device. It would be more informative to add a few key numbers in the abstract such as the range of flow rates tested, explain what limits the flow rates in the device (key for more recent fast-response CRDS instruments operated at high flow rates). The range of water vapour mixing ratios and isotope delta values tested as well as the quality of the delivered calibration or background gas stream could be mentioned (precision, stability).

While we want to avoid too much detail and complexity in the abstract, we will include several of the specific aspects mentioned by the reviewer.

4) L. 15: I think Graf et al. 2019 is a nice example for subcloud processes involved during precipitation but not weather systems in general. Thurnherr et al. 2020 cover a broad range of latitudes and different types of weather systems, which would be a better fit here.

We will either include the additional reference or a set of references here to substantiate this statement.

5) L. 16: "analyser properties" is a bit unspecific, be more precise.

We will add a statement to this sentence specifying spectroscopic baseline effects.

6) L. 17: "the variability of the calibration system", what does that mean?

This expression is meant to encompass the variability of the water vapour mixing ratio and isotope composition of the calibration systems under normal and perturbed operating conditions (e.g. due to bubble bursting in a SDM). We will modify the writing to clarify this aspect here.

7) L. 15-19: Total uncertainty resulting from a variety of instrument characteristics at different water vapour mixing ratios and the characteristics of inlet systems were tested extensively in the two early publications Sturm and Knohl, 2010; and Aemisegger et al. 2012. Application in the field with aircraft-based measurements (Sodemann et al. 2017) and for near-surface humidity gradients (Seidl et al. 2023) were carried out in the more recent ones. This is a bit misleadingly written and should become clearer from this section of the text.

The intention here was to have the two more recent references refer to measurements under very dry ambient conditions (aircraft and polar environment). We will rephrase this section for clarity.

8) L. 29ff: I am not convinced that the microdrop system that is proposed in this paper alleviates all the mentioned problems: given the small liquid reservoir, the isotope composition of the standard liquid in the glass vials should be monitored as well for a reliable normalisation to the VSMOW-SLAP scale. What would be more convincing to me at this stage is to mention primarily that in the bubbler system the reference isotope composition of the vapour phase has to be predicted from equilibrium fractionation, which requires precise temperature regulation of the liquid phase in the bubbler. This problem is overcome in other devices by complete evaporation. Monitoring the liquid isotope composition is a must also in systems with complete evaporation including an SDM standard bag. When using large water reservoirs of several litres the close-up

monitoring of the liquid is not so essential in bubblers because the changes are much smaller than the precision of the reference measurement with IRMS or laser spectrometry.

We agree that the present system does not resolve all problems, and it was not our intention to claim this at this point, the intention was rather to bring forward aspects that contribute to variability in the performance of existing calibration systems. We will rephrase to avoid such a potential misunderstanding. We agree on the point of temperature control of the bubbler system, and will emphasize this point. However, we also like to mention that a large reservoir for a bubbler can impose substantial difficulty during field operation due to weight and form factor.

9) L. 50-51: "limited possibility to regulate the mixing ratio of the water vapour in the airflow", which is not really needed for the mentioned application.

This is true for the used analyzer system, but can very quickly be a problem if the gas flow rate through the cavity is different, for example in flux measurements. We will consider adding a clarifying sentence.

10) L. 64-65: The mention of flow rates in between the water vapour mixing ratio range and the different isotope composition is disturbing the logical flow of information in the reader's mind: Maybe something like the following would help: "... that provides the combination of a precise stream of water vapour across a range of water vapour mixing ratios and isotope compositions as well as operation at various flow rates between 50 and 250 sccm." -> group the range of mixing ratios and isotope compositions together, flow rates is a different aspect also from an operational perspective.

Thank you for this suggestion, we will revise accordingly.

11) L. 71: "and as a component in specific laboratory applications" this is a bit unspecific. Please be more concrete.

The specific applications are stated in the following sentence. We will rephrase for clarity.

12) L. 73: Here it would be very nice to be more precise about the gas stream of a preset background humidity and isotope composition for fluid inclusion analysis. A reader who is unfamiliar with this technique is lost here (see also my minor comment 53).

This sentence mainly serves as an outlook of what is to come. We will add a forward reference here to the respective section (Sec. 6 and 7) to clarify this intention.

13) L. 75-89: This paragraph is very dense with information and a bit difficult to follow. The first sentence starts with introducing the advantage of the microdrop generator in covering "a wider range of applications with CRDS analysers" than with currently available devices. But there are a countable number of applications, and the currently available devices are optimised for the tasks at hand. I rather think the advantage of the microdrop generator is the flexibility of it to produce a wide range of (predetermined) water vapour mixing ratios and isotope compositions. The application to microfluid inclusions in stalagmites is nice but this high flexibility is actually not needed there.

Our intention was indeed to point out the flexibility of the device, we will revise this section accordingly.

14) L. 77: Why are the authors so strict about the water vapour mixing ratio uncertainty? Is this useful in one of their applications? I thought they target the isotope composition of different vapour/liquid/fluid inclusion samples and not the absolute water concentration.

The background humidity for fluid inclusion analysis has in previous applications been provided by peristaltic pumps, which induced substantial noise on the analysis (Affoltern et al., xxxx). We will point this aspect out more clearly in the introduction as a motivation for the precision of the mixing ratio.

15) L. 81: I believe the first publications with fast-response analysers with flow rates above 100 sccm are Thurnherr et al. 2020 ACP with a detailed assessment of the impact of the flow rate on the water vapour mixing ratio dependence on isotope measurements in their supplement and Bailey et al. 2023 ESSD. From these publications with flow rates of about 300 sccm through the cavity it becomes clear that flow rate limitations of calibration systems is an important challenge.

Thank you for pointing out these references, which we will include in the revision.

16) L. 82: This sentence should be removed or moved to the discussion in the conclusions (see also my minor comment 74), the response times of the inlet is not tested in this publication and cannot be tested with the proposed microdrop dispenser because the flow rates are much too small (would need several liters per min to be of interest for this sort of application).

Flow rates of above 150 sccm are indeed possible with the presented system as presented in Fig. 7. While we show tests of long-term stability with 250 sccm, the flow regulator currently allows up to 500 sccm of flow rate. It is correct that downstream dilution will be necessary to provide higher flow rates. We note however that some inlets are operated in this range of flow rates (Chazette et al., 2021). We will revise this section and the discussion to provide the intended information at the correct location.

17) L. 88: The requirement of being field compatible seems ambitious and I don't fully understand it. How well do the dispenser heads work in very cold environments with vibrations and at low pressure? In principle drift correction in the field could be done with simpler existing systems, the advantage of this system resides in the possibility to characterise the water vapour mixing ratio – isotope ratio dependency in detail. This is more a laboratory than a field application.

This requirement comes from the desire to have one multi-purpose system, which could lower cost compared to requiring several dedicated components. It may also for example be desireable to verify the baseline characteristics a system before shipping it back from a field deployment. There are of course a wide range of possible field applications, and there are certain limits that the system presented here can not operate under without additional modification. For example, operation at freezing level would require active heating control within a heat insulation, or the use of temperature-controlled dispenser heads (which are available from the same manufacturer). Low pressure is no limitation to the dispenser head, as they are equipped with a differential pressure regulation with respect to ambient conditions. We have not done vibration testing, but note that there are no actual mechanical parts in the dispenser head that could for example break or get stuck during shock. Our system is not necessarily more complicated than for example an SDM, and the absence of mechanical parts allows for in principle more extensive operating duration. We also have since submission learned that gas bubbles an important aspect that so far has stopped dispensing after several hours. We can now control this aspect much better by de-gassing our standard liquid before operation, and can achieve continuous operation over several days, rather than hours. We will revise the text here and in the discussion to better explain, motivate and detail the field compatibility requirement.

18) L. 92: SDMs can go well below 6000 ppmv.

We acknowledge this is possible, but with limitations. Here is an excerpt from the user manual of the Picarro SMD A0101 User's Guide from April 2010, pg. 14:

"The concentration of vapor will be determined by the user programmed liquid flow rate. A rate of 0.02 microliters/second corresponds to approximately 6000ppmv. The vapor concentration is a linear function of the liquid flow rate. Rates higher than 0.08 microliters/second (24000ppmv) are prevented by the software in order to prevent accidental saturation of the analyzer.
The precision of the isotopic ratio measurement is specified for a vapor concentration of 6000 to 20000ppmv. The precision will suffer significantly below 6000ppmv, increasing the measurement duration will compensate to some degree. The dry air source, such as Drierite® condition air with a 200- 300ppmv water concentration, can contribute significantly to the measured isotope rate when operating at standard vapor concentrations below 6000ppmv .".

We will clarify in the revision that 6000 ppmv is the lower recommended limit of operation of the SDM.

19) L. 94: What does "Injections with the autosampler are feasible with a specific method" mean?

This sentence was mistyped, we will clarify that it is possible to obtain a range of isotope ratios from liquid injections with some manual work.

20) L. 95: This joins my major comment M1: unless you are able to very precisely predict the isotope composition of your mixture with the two dispenser heads you cannot really overcome this limitation from other systems.

See our reply to comment M1.

21) L. 101: this doesn't become clear from the schematic in Fig. 1.

Thanks for pointing this out. As also requested by Reviewer 1, we will add a photograph to Fig 1 or the appendix that clarifies the design of the evaporation chamber.

22) L. 103: this is a very small amount of water which can be rapidly impacted by evaporative enrichment in the field!

The 12 ml glass vial is a default size, but it is possible to operate the DHs from larger reservoirs, we will add a corresponding statement here or in the discussion.

23) L. 106: "retention time of water vapour" are interactions with the chamber walls meant here? If yes, to me "memory effect" would be clearer.

Yes, we will include the term memory effect, as also suggested by Reviewer 1.

24) L. 140: how robust are the voltage and frequency settings and their associated efficiency to prolonged usage?

According to manufacturer information, these settings are constant for each DH once defined, but there are multiple possible combinations that result in different drop sizes. The piezo crystal may degrade over time if very high voltages are involved, which is not the case for our operation. We will add this information with a reference to personal communication with an engineer from Microdrop GmbH.

25) L. 148: "We can used this relation between drop size and mixing ratio to compute the effective drop size..." I am a bit lost here. Do you mean the relation between the frequency and the mixing ratio can be used to estimate the effective drop size?

Yes, this was a typo, we will correct the sentence accordingly.

26) L. 154 and Fig. 3: is it only my print out that make the curves in Fig. 3 not look linear?

The lines in Fig. 3 are strictly linear.

27) Fig. 3b: Why is the H2O mixing ratio coverage worse with the settings for panel b than in panel a?

We have simply run fewer frequencies for this setting, as the linearity was much better.

28) L. 161: This seems to be key, indeed. Can the uncertainty from this fitting procedure be quantified in ppmv for a series of runs? And how frequently does this readjustment of dispensing parameters need to be done?

We have since submission come to understand that this assessment does not need to be done very frequently, maybe on a yearly basis, to ensure the DHs have not been ageing, as is expected.

It is possible to automate this characterisation from the control software. In the revision, we will add information for the uncertainty for both run series.

29) Fig. 4: I am not sure that I really understand the meaning of this figure. What does all dispersion parameters mean? Could this Figure be placed a bit more logically in the discussion. A part from a useful diagnostic, I don't see the use of it? Is it possible that only showing the blue and red distribution would help the reader because these are the selected settings. In general, I am not sure that the reader is interested in the full distribution with all your tested dispersion parameter setting.

It appears that Reviewer 1 appreciated this information, as precise humidity control is a great concern in fluid inclusion analysis. We will therefore clarify in the Figure caption that all dispersion parameters corresponds to the data displayed in Fig. 3.

30)L. 192: here and elsewhere: it is "Allan" variance not "Allen" and the adequate reference would be Allan, 1966 Allan, D. W.: Statistics of atomic frequency standards, Proc. IEEE, 54, 221–230, doi:10.1109/PROC.1966.4634, 1966.

Thank you for this correction and reference which we will include in our revision.

31) Eq. 1: I think the authors mean $y_{i+1}(\boxed{?})$ not $y_{i+1}(\boxed{?})$.

Yes, thank you for this correction.

32) L. 206: The SDM is optimised for the precision of the $\boxed{?}\boxed{?}\boxed{?}\boxed{?}$ and $\boxed{?}2H$ not for the mixing ratio. For a fair comparison with a high quality mixing ratio signal, a dew point generator should be used.

While it is true that the SDM is not optimized for mixing ratio stability, this is an important aspect for the fluid inclusion application and thus part of the assessment here. We will however consider including a comparison of mixing ratio stability only from a dew point generator for the revision.

33)L. 225: this is understandable only if the basic concept of fluid inclusion peak measurements on a stable background has been explained in the intro. Maybe a reference to Section 7 could also be added here.

A reference to Sec. 7 and Fig. 13 are already included, but we will in the revision clarify better upfront how the fluid inclusion measurements work.

34) L. 254: "The linearity of this mixing is provided over a range of mixing ratios". I can't really evaluate this statement based on Fig. 8a. How about coloring the data points with the mixing ratio?

We will implement this suggestion in the revised version.

35) L. 255: "At lower humidity than 5000 ppmv deviations from the linear mixing become apparent" Also here, I can't really see this in Fig. 8b, which shows the RMSE of $\boxed{?}\boxed{?}\boxed{?}\boxed{?}$ and $\boxed{?}2H$ from the observed mixing line at different water vapour mixing ratios. The RMSE is by the way very large implying that the predicted reference isotope composition of the gas stream has a large uncertainty making it very difficult to me to effectively these mixed samples as reference calibration samples.

It appears that Fig. 8 was not as clear as we hoped. We will make an attempt to better support the statements in the text with this Figure. We will double-check the scale of the RMSE which indeed indicates very large values.

36) L. 260: what is Mix000 and Mix 100? This is the first time these labels appear.

Mix000 is purely from DH1, and Mix100 purely from DH2 (the number is the percent fraction from DH2). In the revision, we will define these abbreviations as they are introduced.

37) L. 260: Where can I see this bending down (please add a reference to the figure that is meant here).

We will add a reference here to Fig. 9.

38) L. 263: the phase space you are showing in Sec. 6 is the correction function in the ln(mixing ratio) vs. δ2H (or δ18O) phase space.

We are unfortunately not sure how to interpret this comment, the logarithmic transform in Sec. 6 is used to obtain a more linear display of the dependency characteristics.

39) L. 265: the microdrop vapour generator was already connected to the analyser for all the evaluations presented in the previous section. Here you switch from characterising the microdrop vapour generator to using it to characterise CRDS analysers in their water vapour mixing ratio and isotope composition dependency.

The intention here was to contrast two applications, one where the microdrop vapour generator is directly connected to the CRDS, and one where there are additional applicances in between. Apparently this statement can be misunderstood and we will consider rephrasing in the revision.

40) L. 267: The water vapour mixing ratio and isotope composition dependency is of spectroscopic origin and due to uncertainties associated with the baseline. Even if a system is optimised for measurements at lower humidity levels than 20'000 ppmv this problem occurs.

This is true. What we wanted to point out here is that Picarro analyzers are in fact already corrected for baseline effects in the factory with a target humidity of abour 20'000 ppmv, and the remaining dependency is thus a residual of this factory baseline correction. We will rephrase for clarity.

41) L. 273: This is misleading. The spectroscopic origin of the concentration dependency has been mentioned already in early laser spectrometric instrument evaluation work such as from Sturm and Knohl, 2010 and Iannone et al. 2009.

We think this might be a misunderstanding, it was not our intention to mislead the reader. What we mean to say is that the additional dependency of the previously identified mixing-ratio dependency (e.g., Sturm and Knohl, 2010, Iannone et al., 2009) was shown to be in addition dependent on isotope ratio by Weng et al., 2020. We will rephrase to avoid potential misunderstanding.

42) L. 274: matrix gas -> carrier gas

ok

43) L. 296-308: This is very difficult to follow, could the respective sequences be clearly indicated in Fig. 9? This paragraph should be in stronger dialogue with what we see in Fig. 9 for the reader to be able to follow.

We will add annotations and rewrite to connect Fig. 9 and this paragraph better.

44) L. 313: Here I got lost, where do I have to look above or below?

We will rephrase and add a reference to the relevant section.

45) L. 315: Can the authors be more precise about the number of outliers they filtered out? This is an important quality measure for the microdrop generator. Because an ideal calibration system needs as little instrument measurement time as possible so minimising the occurrence of "outliers" is a key aim when designing a robust calibration system.

We will include the number of outliers here. However, we note that after the potential of leaks was detected, care was taken to avoid such artifacts and no more outlier removal was required thereafter.

46) L. 320: What is the share of new data points from the microdrop generator vs. the data points from Weng et al. (2020) in this analysis shown in Fig. 10? Could the data points from the microdrop generator be highlighted in a different color or shape?

We will highlight the additional data points in the revised Figure 10 by a different color.

47) L. 331: the delta value is specified and used in equations 2 and 3.

Since the humidity is typically not exactly at 20'000 ppmv, but bracketed by values nearby, we interpolate between neighbouring data points. Interpolation with a polynomial makes thereby use of most of the data points.

48) L. 336: this final step was a bit obscure to me. I thought that the correction function is 0 per definition at the reference mixing ratio level?

The 2D polynomial fit is not constrained to zero at the reference level and thus result in an offset, which we correct for by this additional final step.

49) Table 3: the RMSE is most likely very dependent on the mixing ratio? Maybe a Figure showing this dependence as a function of mixing ratio would be more informative.

We will investigate the dependency with respect to mixing ratio and consider to include a figure or additional lines in Table 3 for sub-sets of the mixing ratio.

50) L. 350-355: this paragraph was again not clear to me. Which subsample of available measurements?

This was a Monte-Carlo approach, where we draw a non-unique sub-sample from the actual distribution of data points. We will give more detail and attempt to rephrase for clarity.

51) Fig. 11: The vertical axis is not the raw but the reference $\delta^2H$ and $\delta^{18}O$, right? There are no dashed grey lines (at least not in my print out) and what are the blue lines? Would this figure be better readable if it was shown in this format in the appendix (to illustrate the limited concentration effects at high water vapour mixing ratios) and with a zoom in to 0-5000 ppmv (which is the interesting part) in the paper? Right now one does not see much about the important effects at low water vapour mixing ratios.

Vertical axis is indeed the raw delta values, we will correct this. Blue lines show contour levels for the correction found here, and solid black lines are the correction function of Weng et al., 2020, this will be corrected in the caption.

52) L. 357: "a constant vapour stream" with known water vapour mixing ratio and isotope composition (very important!)

Yes, we will add this important information.

53) L. 357-366: This information should already be provided in the introduction.

We understand from the comments of Reviewer 2 that this information is necessary in the introduction and will revise the introduction and Section 7 accordingly.

54) L. 369: "memory effects can be removed entirely" is misleading and a bit optimistic. Isn't the advantage of the CRDS technique mainly the ability to measure $\delta^2H$ and $\delta^{18}O$ (possibly even d17O) quasi-simultaneously with much reduced sample preparation effort?

We will moderate and rephrase this statement.

55) L. 372: "higher throughput and better reproducibility" than what?

During fluid inclusion analysis, several samples are typically analysed from the same growth layer of a stalagmite. We will add a statement that this is a comparison to methods that involve collection of the sample by a cold trap for subsequent analysis on a mass spectrometer.

56) L. 384: Fig. A2 -> Fig. 12?

Correct.

57) L. 387-393: connect this paragraph better to Fig. 13, they should be in better text- illustration dialogue to help the reader follow.

We will rephrase this section to integrate better with Fig. 13.

58) L. 395: As mentioned in M2 I wonder about the necessity of this step given the high mixing ratio of the background air stream.

See our reply to comment M2 above.

59) L. 397: "Mixing ratio -isotope ratio corrections..." this has already been mentioned in point i) just above and appears as an unnecessary repetition here.

We will revise for brevity.

60) L. 399: what is meant by "size-?" relationship?

This refers to a relation between the amount of liquid injected and the delta value during calibration. We will rephrase this expression.

61) L. 405-414: I got lost in this paragraph. As the authors write at L. 410 this appears redundant. Fig. A1 is not very convincing in showing a consistent dependency of the isotope signals on the injected amount.

This section relies on the analytical methods published earlier by Affolter et al., (2014). While the step may appear redundant, it is important here to maintain the principle of identical treatment of sample and standards. We will revise for clarity.

62) Fig. A1: To which mixing ratio levels do these injection volumes correspond?

We will add this information to the figure caption.

63) L. 407: "drift in values", what do you mean by values?

This refers to a drift in the background, we will rephrase for clarity.

64) L. 412: the last sentence of this paragraph "Finally,..." is particularly obscure to me. Could the authors illustrate this with a figure?

We will either refer here to already published method description or include a supplementary figure to clarify this part of the procedure.

65) L. 431: Here I am a bit puzzled: the authors write that water vapour mixing ratio – isotope dependency corrections are necessary below 5000-10'000 ppmv (L. 270). But the background gas stream can be designed in such a way that the mixing ratio is larger than this and ideally even close to the optimal water vapour mixing ratio operation level of 20'000 ppmv? So why does this correction matter? Can the authors show how different the isotope signal estimates get when applying vs. when not applying the water vapour mixing ratio correction for their aliquot or the samples used to mimic real fluid inclusions? This is also my major point M2.

See our reply to comment M2 above.

66) L. 432: "more correct" than what?

This sentence will be rephrased.

67) L. 434: Why is the precision of the water vapour mixing ratio mentioned here? Isn't what matters for precise fluid inclusion measurements the precision of the isotope signals?

As explained in our reply to M2, the stability of the mixing ratio is important for this application to separate the background water vapour clearly from the water vapour released from the crushed sample.

68) L. 455: for which application would horizontally mounted DHs be an interesting option?

Due to the design of the evaporation chamber, the DHs are mounted horizontally here. We will rephrase this sentence for clarity.

69) L. 462: Flow rates between 300 and 600 sccm through the cavity are now used in different applications (aircraft-based and flux measurements) and present many advantages (e.g. faster response times, better signal to noise ratio at shorter averaging times) but this range of flow rates cannot be covered the microdrop dispenser system. Why is that so, and is an extension to these higher flow rates possible with the given operational range of dispenser head triggering frequency?

The current limitation to 500 sccm for the device presented here originates mainly from the specification of the gas flow regulator. It may be possible to obtain higher flow rates with the present device, but we did not have the necessary pumps available to produce such high cavity flow rates. Higher flow will require higher dispensing frequencies, and/or dispenser heads that can produce larger drops. We will add some clarification of this aspects to the discussion.

70) L. 475: remind the reader that the Picarro SDM is the commercially available benchmark used for characterisation of the microdrop dispenser.

Will be added in the revision.

71) L. 475: In the long-term precision (relevant for calibration) the microdrop dispenser is not better than the SDM for the isotope signals (which we are interested in). The dispenser is better than the SDM for water vapour mixing ratio, for which however a dew point generator is needed anyway for precise calibration. Together with the many small problems that occur with air bubbles in the dispenser head, the small amount of liquid that needs to be exchanged regularly, this questions the utility of this system for regular field calibrations. It is very useful for an in-depth characterisation of the water vapour mixing ratio – isotope signal dependency in the lab but apparently not so much for prolonged measurements in the field.

We can partly understand this assessment of the reviewer based on the presented information. However, since submission of the manuscript, we can report on a much longer uninterrupted operation time of the device thanks to de-gassing of the calibration liquid before use. We will update the relevant parts of the manuscript accordingly.

72) L. 477: I don't understand what this sentence implies scientifically. Which specifications are meant here? The dispenser head and mass flow controller specifications or the objectives set by the authors for the system in Section 2? Maybe a reference to section 2 could be added here.

Yes, we will add a reference to Section 2.

73) L. 495: Could this simply be a numbered list of implications? It would be easier to read.

We will consider reorganising as suggested.

74) L. 502: Here is the place where the sentence from L. 82 could be brought in: larger flow rates are needed to test the inlet system and the overall response time of different water vapour isotope measurement field setups.

Thank you for this helpful suggestion, this paragraph will be revised accordingly.

75) L. 502: Again this is misleading, response times of laser spectrometric water vapour isotope measurements were not the topic of this publication. If one just reads the conclusions, this sounds like a result from this study. References to the studies Sturm and Knohl, 2010 AMT (tested Synflex tubings in the lab) and Tremoy et al. 2014 JGR (tested Synflex tubings in the field) should be made.

The reference to Tremoy et al., 2014 went missing during the editing, it was certainly not our intention to mislead the reader, see also reply to Reviewer 1. We will add the mentioned references in the revision.

Technical points:

1) L. 17: . Currently
2) L. 67: technology, which allows
3) L. 75: "water" vapour
4) L. 76-80: Grammatically and for keeping the reader's attention having (ii) in the same sentence as (i) would be much more convenient.
5) L. 103-104: "one 12 ml glass vial that holds a liquid water standard and which is mounted next to ...". It is not the evaporation chamber that holds the liquid water standard, right?
6) L. 161: demonstrates
7) L. 161: "... a suitable dispensing parameters"
8) L. 214: microdrop
9) Caption Fig. 4: dispening -> dispersion
10) L. 348: og -> of
11) L. 369: CDRS -> CRDS
12) L. 464: viscousity
13) L. 509: performed
14) P. 27, Fig. A1: is "assigned" really the term you want to use here on the y-axis? To me this seems to be the "reference".

We will implement all technical changes as suggested.

---

## Author Response (AR1)

**Revision of "A flexible device to produce a gas stream with precisely controlled water vapour mixing ratio and isotope composition based on microdrop dispensing technology" by Sodemann et al., submitted to AMT**

Dear Editor,

we have now completed substantial revisions of our manuscript submitted to AMT. We reply below to the numerous, detailed and constructive comments by the two reviewers. We have addressed all points, and feel that we could accomodate these or reply in adequate form. We have in addition made several minor edits to improve overall readability, and added several newly available references.

In our reply below, we show the reviewer comments in italic, whereas our reply is in normal font at an inset.

With best regards, on behalf of all authors,
Harald Sodemann

*Reviewer #1*

*General.*

*This is a very thorough paper outlining a novel method for dealing with one of the main challenges associated with using laser-based spectroscopy on very small samples- namely the concentration dependence of isotopes. It is a significant body of work that provides a framework for evaluating performance challenges associated with calibrating CRDS systems under a variety of conditions to make the most precise and accurate measurements. The authors provide several applications, including the measurements of inclusions in stalagmites which provides a very concrete example of how this methodology could transform the current methods used for the analysis of fluid inclusions. This paper has significant impacts on the future design and implementation of critical measurements of water vapor isotopes both in the lab and in the field.*

*Laser-based instruments have provided an enormous step change in our ability to make important measurements in all aspects of the hydrosphere and the atmosphere. But our ability to thoroughly calibrate and account for a number of artifacts, including a mole fraction or concentration dependence for the isotopes; sample carry over; and interaction with surfaces to which the sample is exposed along the measurement route, has remained a challenge. This paper addresses a number of those.*

*The authors have shown the details of applying existing technology from another field (inkjet printers). While this technology has appeared in previous publications – notably the 2008 paper by St. Clair et al. which is referenced by this paper, this remains to be an important next step in validation and proof of concept for its application with isotopic calibration on small sample sizes. It goes well beyond the scope and purpose of St.Clair (2008).*

*Key findings include how the micro-dropper device is able to produce a water background of constant isotopic compositions that can be used to obtain accurate and precise $\delta^{18}O$ and $\delta^2H$ measurements over a variety of concentrations, controlled by parameters of the Dispenser Heads. The inclusion of the comparison to the Standard Delivery Module (SDM, Picarro Inc, Sunnyvale, USA) is a good benchmark and will no doubt be of interest to readers.*

*The examples of the fluid inclusions would appear to provide validation of methodology that would stand to be an improvement over current methods of measuring fluid inclusions. While a comparison with samples analyzed with mass spectrometer methods would have been useful, the validation provided by the analysis of standards introduced via capillary pipets is convincing.*

We thank the reviewer for their thorough feedback and the constructive comments.

*Specific*

*Line 51: In the introduction, a discussion of similar techniques used to introduce and vaporize liquid water and highlights those used for CFA analyses (Gkinis et al., 2010). To be complete, consider also referencing Jones et al. ,2017 which employs a nebulizer, that solved some of the challenges found in the WVISS system outlined in Aemisegger et al., (2012).*

Thank you, we have added this useful reference to the introduction.

*Line 99: A long with Figure 1, a picture of at least the apparatus in general and/or the critical components (eg. The microdrop dispenser heads; evaporation chamber) would greatly help.*

We have added drawings of the evaporation chamber as an appendix figure. In addition, we included a close-up photograph of the dispensing heads inserted into the evaporation chamber for illustration.

*Line 103: With regard to the use of the Microdrop, dispenser head (DH), JM St. Clair et al., 2008 (Reference on line 575) uses the Microdrop part MD-K-130-010, while the authors chose to use Microdrop GmbH, Germany, Part Nr. MK-K-130. It could be useful to explain the choice of microdrop dispensers, and whether a heated or non-headed head was used and why.*

We have investigated further regarding the part number. The -010 refers to the viscosity range of liquids to be used for dispensing. We have clarified and updated the description of the dispensing head in comparison to St Clair et al., 2009:

"The two tubes reach 3 mm into the interior of the chamber, and each holds a dispenser head (Microdrop GmbH, Germany, Part Nr. MK-K-130-020) with an inner nozzle diameter of 50μm for liquids with a viscosity below 20 mPas. For our prototype design, we chose an unheated head with a medium-size nozzle, similar to the design of St Clair et al. (2008)"

*Line 150: Clarify "To this end, we first calibrate the raw mixing ratio signal with a calibration curve obtained from a dewpoint hygrometer." Consider adding "To this end, we first calibrate the raw mixing ratio signal measured by the Picarro with a calibration curve obtained from a dewpoint hygrometer." Also, it would be helpful to name the brand/model dewpoint hygrometer.*

We have modified this sentence as suggested and added information about the dewpoint hygrometer (Optisonde, GE Inc., USA).

*Line 154: Performance characteristics for DH1 and DH2 are mentioned, but not DH3. Consider mentioning why these two were selected for testing, and not DH3. Perhaps because these two were the ones chosen for the system, based on performance? Also, can operating frequencies and dispersion parameters be specified, or are they DH dependent, and constitute unique operating parameters to be determined for each unit? In the section on the application of this method, it would be useful to have comparisons of this method vs. the traditional extraction/ mass spec method as validation of the method.*

DH3 was essentially purchased at a later date and used as a spare. We already provided some details about the choice of dispensing parameters, which are indeed unit dependent in Sec. 4.2. More details have been added to the text:

"The default factory calibration parameters can be modified to obtain different drop sizes."

and to the caption of Table 2.

"As DH3 was purchased as a spare at a later time, we herein focus on results obtained with DH1 and DH2."

We also add newly available information about the temporal stability of these characteristics:

"According to manufacturer information, unless when operating at much higher voltages, the piezo-electric characteristics of each DH are thereby expected to be constant over time."

We understand the potential value of a comparison to traditional methods, but do not currently have an extraction line available. Instead, we use the glass capillaries as a validation method.

*Line 332: A significant side bar finding of this study is the statement that " We find that different analyzers require different polynomials for fitting in the Δδ and the ln(x) direction. This may come as a surprise to some adopters of CRDS systems for isotopic measurement and is an important finding.*

We made this finding (which has already in principle been observed in several other studies) more prominent by adding it into the respective paragraph in the Conclusions:

"Along the ln(x) axis, a 2nd- order polynomial was sufficient for all for analysers investigated here. Along the Δδ axis, for two of the four analyzers a 2nd-order polynomial was sufficient, while the other two had better fitting results with a 3rd-order polynomial."

*Line 370: To my knowledge, Affolter et al., (2014) does not mention memory (or sample carry over) explicitly. So, it could be appropriate to address memory in this paper more directly? Also, does sample carry over affect d18O differently than dD in this system? Deuterium excess would be an interesting parameter to include if possible.*

The reference to Affolter et al., 2014, was incorrect here, it should have been a reference to de Graaf et al., 2021, who directly address memory effects. This has been corrected in the revised manuscript.

*Line 440: Long term sustainability and reliability are important qualities of any system like this. Consider expanding upon the statement "While in general working reliably over several hours at a time, these problems are important to be aware of, and one needs to search for remedies in future development of the calibration system". This is an important step forward. Knowing how long this system can be expected to function reasonably well without manual intervention is useful. I would encourage authors to speculate on areas where reliability might be improved to potentially lengthen the time it can run unattended.*

We have expanded upon this statement in the discussion section. Since submission of the manuscript, we already made important progress in identifying a key limiting parameter affecting the duraition of unattended operation, namely the amount of dissolved gas in the dispensed liquid. Proper degassing of the calibration liquid ensures substantially extended operation times, in practice advancing from hours to days of continuous operation. An explanation of this critical aspect and gas removal procedures have been added in Sec. 4.3.

*Line 502. It is stated that "Prominently, Synflex tubing has been shown to render δ2 H measurements meaningless". This (rather important) conclusion does not appear to be supported by evidence provided in this manuscript. If this has been shown elsewhere, a reference would help.*

We now included a reference to Tremoy et al., 2009, which was by mistake dropped from the submitted draft manuscript.

*Appendix A: Crusher application details: Figures A1 and A2 are referred to in the main text, so I am not sure why they are in the appendix, unless there are space limitations? No details beyond figures are found here. Consider either moving appendix figures to the main body of the text or changing to "Additional crusher application figures"? Editorial call.*

We considered including these figures in the main manuscript, since we mention these in the main text. However, we found that the flow of the manuscript is better if these additional figures remain in the appendix, since they are not discussed extensively. In particular, the readers can focus more on the example figure showing an analysis sequence with the crusher and microdrop device.

*Some small Typos or wording:*

*Line 17: needs period at end of sentence.*
*Line 268 & 276: Not sure which version AMT prefers: English version of 'artefact' vs. artifact. Editor's choice?*
*Line 322: 20'000 should be 20,000.*
*Line 349: og -> of*

We have corrected these typos in the revised manuscript.

*Figure 1. This schematic explains the set up adequately, but it would help tremendously to add a figure contains images of the micro-drop heads, and/or the evaporation chamber, as this appears to be the heart of the innovation here and deserves some focus.*

As mentioned above, we have add schematics in an appendix figure, including a photograph of the dispenser heat ports on the evaporation chamber.

*Figure 2. These images are very helpful. Consider adding to the caption how the images were obtained or with what optical imaging system.*

We have included in the caption how these photographs were obtained with a USB camera:

"Images have been obtained with a time-synchronized USB-b/w-camera with 10x objective (part #MD-O-539-USB, Microdrop GmbH, Germany)"

*Figure 3. See comment for line 154. Consider adding a note here on why just 2 of the 3 heads were tested. Perhaps 3 were evaluated and two selected for the vaporizer. In 5 b) there are many fewer points above about 10 Hz for DH1. An explanation would help.*

DH3 was essentially purchased at a later date and mostly used as a spare (see reply to comment on line 154). In the original figure, there were more steps below 10 Hz, since we mainly focused on the sensitivity of the delta scale at low humidity in the tests shown there. We now combined this Figure with Figure 4, and show a analysis sequence with a more even distribution of dispensing frequencies.

*Figure 4. Both a) and b) show bimodal distribution. According to the text, the day-to-day standard deviations are shown in red and blue. Grey histograms are for performance over the entire time period. The difference is not large (~9 or 10μ) but groupings are distinctly different. An explanation of these groupings might help. Perhaps it is due to the physics of discrete droplet formation that the volume (or diameter) of droplets is not a continuum, and is constrained by the droplet formation process itself? This might be obvious to those familiar with the DH technology, but a sentence of explanation would help those who are not.*

In light of the comments by reviewer 2, we have removed the histograms with settings that are non-optimal, and only cite this as a perturbing factor in the revised manuscript.

*Figure 5. Allen deviation plots are very helpful, as is the comparison with the SDM. Figure 8. This is an important figure. Consider adding a panel o*

Unfortunately, it appears that part of the reviewer comment went missing. Fig. 8 has been revised in the light of comments from reviewer 2.

*Fig. 11: This is an excellent and key figure, showing the comparison of correction functions of Weng et al. (2020). The caption refers to dashed grey lines in the figure – but there are no dashed lines as mentioned as far as I can tell, but there are possibly blue and black lines? (I am thinking the caption may have been created with a previous version of the plots?) Either make the colors more distinct, or make one set of lines dashed, as the caption reads.*

We corrected the caption in the figure, and now plot the data points from Weng et al., 2020 as dashed lines.

*Figure 13. Excellent figure. If I understand this method correctly, peak values are integrated between the green markers, which show the start and the end of the peak. For both d18O and dD there is a significant drop (lighter isotopically) just as the peak in mixing ratio arrives in the analyzer. Presumably, integration between the green markers (including this dip) yields the correct value for water standards introduced with capillary tubes? Maybe I missed something about the calibration? Based on the performance of the analyzer as shown in figure 9, It would seem that integration of the sample might be more stable later? Maybe sentence in the text or caption could address this.*

The sequence shown in Fig. 9 does indeed show a more stable signal at the end of a step change. However, in Fig. 9, we change dispensing frequency, after which the dispensing system needs to find a new equilibrium state. In the crushing application shown in Fig. 11 (now Fig. 13), the microdrop device remains without frequency changes, and only the sample water is introduced into the background stream during crushing. The entire water peak from the crushed sample needs to be integrated as the sample signal is never at a constant humidity. This is now stated more clearly in the text.

We speculate that the negative dip may be due to fractionation during the wetting of the newly exposed calcite surface during the crushing. This is now mentioned in the text as a speculation.

"During the release of sample water into the background air stream, we note an initial dip to more depleted isotope composition, which we speculate can be due to surface effects between the newly exposed surface area of the calcite sample and the background vapour stream that favour deposition of heavy isotopes. Thereafter, the water peak from the sample arrives on top of the background. In order to quantify the total water released from the sample, integration is done between the green dots."

*Figure 14. Image of glass pipets is helpful. Presumably the glass pipets are introduced in the crusher unit, if so, this could be mentioned in the caption.*

We added this information to the caption: "For measurements, the entire pipet is then crushed in the crushing unit."

*Figure A1. Caption could be improved by listing the names of the internal (or inhouse, secondary) standards and their assigned values with ± uncertainty, and explanation of the analysis method used to assign their values to the VSMOW-SLAP scale. Perhaps adding "note y- axis scales differ" would also help.*

We have added the names and assigned values of our internal lab standards to the figure caption, and included a note on the difference of y-axis scales. This figure has also been changed in light of comments by reviewer #2.

**Reviewer #2**

*This paper presents a new device based on two microdrop dispenser heads that allows to inject a controlled amount of water with known isotopic composition into a heated evaporation chamber flushed by a dry carrier gas for subsequent analysis with a laser spectrometer. The authors argue that the novelty of this device is its flexibility in terms of producing air samples with a range of different water vapour mixing ratios and isotope compositions varying between the two end-members of the liquid water samples feeding the two dispenser heads. The authors show two applications of the new device: 1) the detailed characterisation of the well-known water vapour mixing ratio dependence of isotope measurements with laser spectrometers and 2) the production of a stable background water isotope signal for the precise measurement of microfluid inclusions in stalagmites.*

*The paper is interesting and well-illustrated. The major innovation is the combination of two dispenser heads instead of one, which gives more flexibility in the generation of water vapour samples with a range of different isotope compositions and water vapour concentrations. While most of the text is well-written and easy to follow, there are a few parts that are written in a misleading way, where I suggest that the reader guidance and accuracy of the description could be improved (see my minor comments below).*

We are very grateful to the reviewer for their attentive reading of our manuscript, and the detailed and specific constructive comments.

*The paper should be published in AMT after the following two major points and a longer list of minor comments as well as a few technical aspects have been addressed:*

*M1: Given that the key innovation of the proposed system is based on the combination of two dispenser heads for being able to produce a series of gas streams that cover a range of different water vapour mixing ratios and isotope compositions, a solid assessment of the uncertainty of the predicted reference properties ($[H2O]$, $\delta^2 H$, $\delta^{18}O$) should be provided. Currently, this aspect is very difficult to assess. From Fig. 8 it seems that the RMSE of the obtained mixtures of standards is very large and therefore the system's characteristic uncertainty in the reference isotope composition and mixing ratio when using the two dispenser heads simultaneously seems to me an important issue.*

> This is a very valuable comment, we agree that the aspect of the uncertainty of the mixtures is important in the context of this device. We have made substantial edits to better explain these reference properties, and in particular Fig. 8. We discovered that there was an error in the calculation of the RMSE. In the revised manuscript, the uncertainties are now two orders of magnitude lower than the erroneous numbers in the initial submission. We also note that we mainly demonstrate the overall feasibility of mixing from two dispenser heads. Several known factors contribute to the quantified uncertainty, and while it is possible to obtain a more precise mixing with further optimisation and processing, we opt to leave such performance tuning to future work.

*M2: I am not convinced that the relatively complex water vapour mixing ratio – isotope dependency correction is really needed in the second application for micro fluid inclusions in stalagmites, because the water vapour mixing ratio of the background airstream is well above 10'000 ppmv and the variations due to the microfluid injection peak is in a range of values where this correction is not usually needed. Could the authors show the impact of this correction on their aliquot of a natural carbonate and/or their samples used to mimic real fluid inclusions? It's a major point because the authors argue that this is an important additional processing step that was neglected in the previous literature (L. 430-437).*

> Fluid inclusions contain a variable amount of liquid in the samples. Therefore, the mixing ratio in the background air stream has to be adjusted, and can not always be at 10'000 or even 20'000 ppmv, but may have to be for example as low as 5000 ppmv. With the strong variations that occur during the crushing procedure in mixing ratio, there can indeed be a significant impact in the mixing ratio - isotope ratio dependency. We have seen this in a series of standard water injections in the crushing line (Fig. R1). Standard waters have been injected on a humid air background between 10'000 and 15'000 ppmv. The difference between corrected and uncorrected integrated peak values is consistent with the range of corrections in the HIDS2254 shown in Fig. 10 in the revised manuscript. A difference of 0.1-0.2 permil in δ18O and 0.5 to 1.0 permil for δD for this specific analyzer can cause consistent offsets in comparison with other laboratories, and does represent an important factor in our opinion. As mentioned above, samples may require lower background mixing ratios, which would then for this analyzer increase the correction impact. Furthermore, different analyzers have a different shape of the correction, and thus a different sensitivity in the typical range of operations for fluid inclusion measurements. Furthermore, the stability of the mixing ratio is important for this application to separate the background water vapour clearly from the water vapour released from the crushed sample. We have made edits in various places of the manuscript to highlight this reasoning, and provide more evidence to support this point from a simulation of the impact for hypothetical samples. Furthermore, we include the measurement series of standard DI2 in the Fig. R2 as a new figure (Fig. 13) to lend support to this important finding in the revised manuscript.

[Figure]

**Figure R1:** difference between corrected and uncorrected raw measurements from injections of different amounts ("size") of liquid water standards in the crushing line with different H2O background mixing ratios.

*Minor comments:*

*1) L. 1, L. 22 and at other places: "a flow of air-vapour mixture" is a bit confusing: not clear what vapour of which gas and air does a priori comprehend water vapour as well. The terminology used in the title seems more precise to me: "a gas stream of a pre- determined water vapour mixing ratio and isotope composition".*

> We have now used the expression in the title at the start of the abstract and the start of the introduction for consistency.

*2) L. 6: "water vapour mixing ratio", just mixing ratio can be any gas or isotope.*

> revised as suggested.

3) Abstract: the abstract is a bit short and lacks key information on the new device. It would be more informative to add a few key numbers in the abstract such as the range of flow rates tested, explain what limits the flow rates in the device (key for more recent fast-response CRDS instruments operated at high flow rates). The range of water vapour mixing ratios and isotope delta values tested as well as the quality of the delivered calibration or background gas stream could be mentioned (precision, stability).

> While we want to avoid too much detail and complexity in the abstract, we have rewritten parts of the abstract to include several of the specific aspects mentioned by the reviewer.

*4) L. 15: I think Graf et al. 2019 is a nice example for subcloud processes involved during precipitation but not weather systems in general. Thurnherr et al. 2020 cover a broad range of latitudes and different types of weather systems, which would be a better fit here.*

We have included the Graf et al 2019 reference as an example for a detailed study of a cold-front passage. In the revision, we have also included a reference to Thurnherr et al., 2020.

*5) L. 16: "analyser properties" is a bit unspecific, be more precise.*

We have rephrased this sentence, specifying spectroscopic baseline effects.

*6) L. 17: "the variability of the calibration system", what does that mean?*

This expression is meant to encompass the variability of the water vapour mixing ratio and isotope composition of the calibration systems under normal and perturbed operating conditions (e.g. due to bubble bursting in a SDM). We will have rephrased to clarify this aspect:

"Currently, variability of the gas stream produced by the calibration system, both in terms of mixing ratio and isotope composition, together with inlet and instrument characteristics, are important contributors to the total uncertainty of atmospheric water vapour isotope measurements."

*7) L. 15-19: Total uncertainty resulting from a variety of instrument characteristics at different water vapour mixing ratios and the characteristics of inlet systems were tested extensively in the two early publications Sturm and Knohl, 2010; and Aemisegger et al. 2012. Application in the field with aircraft-based measurements (Sodemann et al. 2017) and for near-surface humidity gradients (Seidl et al. 2023) were carried out in the more recent ones. This is a bit misleadingly written and should become clearer from this section of the text.*

The intention here was to have the two more recent references refer to measurements under very dry ambient conditions (aircraft and polar environment). We have split the sentence and rephrased for clarity:

"Separating different contributions of uncertainty is in particular critical at low humidities, such as for airborne measurements (Sodemann et al., 2017) and in cold environments (Casado et al., 2016, Seidl et al., 2023)."

*8) L. 29ff: I am not convinced that the microdrop system that is proposed in this paper alleviates all the mentioned problems: given the small liquid reservoir, the isotope composition of the standard liquid in the glass vials should be monitored as well for a reliable normalisation to the VSMOW-SLAP scale. What would be more convincing to me at this stage is to mention primarily that in the bubbler system the reference isotope composition of the vapour phase has to be predicted from equilibrium fractionation, which requires precise temperature regulation of the liquid phase in the bubbler. This problem is overcome in other devices by complete evaporation. Monitoring the liquid isotope composition is a must also in systems with complete evaporation including an SDM standard bag. When using large water reservoirs of several litres the close-up monitoring of the liquid is not so essential in bubblers because the changes are much smaller than the precision of the reference measurement with IRMS or laser spectrometry.*

We agree that the present system does not resolve all problems, and it was not our intention to claim this at this point, the intention was rather to bring forward aspects that contribute to variability in the performance of existing calibration systems. We agree on the point of temperature control of the bubbler system, and now mainly emphasize this point.

However, we also mention that a large reservoir for a bubbler can impose substantial difficulty during field operation due to weight and form factor:

"Despite its overall simplicity, there are several drawbacks with bubbler designs. First, precise temperature control is required to predict the isotope composition of the water vapour from evaporation under equilibrium fractionation. Additionally, a reservoir of up to several liters of water may be needed to limit the impact of drift from the changing isotope composition in the liquid over time. Depending on the measurement platform and ambient conditions, handling of such amounts of liquid may be a hindrance during field deployments."

*9) L. 50-51: "limited possibility to regulate the mixing ratio of the water vapour in the airflow", which is not really needed for the mentioned application.*

This is true for the used analyzer setup, but would be a problem if one were to produce variable gas flow rates. We have rephrased this sentence, also in the light of a comment from reviewer 1:

"Both designs have been operated over extended time ranges (e.g., Bonne et al., 2019), but have not been constructed to regulate the mixing ratio of water vapour in the gas flow."

*10) L. 64-65: The mention of flow rates in between the water vapour mixing ratio range and the different isotope composition is disturbing the logical flow of information in the reader's mind: Maybe something like the following would help: "... that provides the combination of a precise stream of water vapour across a range of water vapour mixing ratios and isotope compositions as well as operation at various flow rates between 50 and 250 sccm." -> group the range of mixing ratios and isotope compositions together, flow rates is a different aspect also from an operational perspective.*

Thank you for this suggestion, we have revised accordingly.

*11) L. 71: "and as a component in specific laboratory applications" this is a bit unspecific. Please be more concrete.*

We have rephrased, including a reference to Affolter et al., 2014:

"Due to its flexibility, the device is suitable for a range of applications, including instrument characterisation, calibration of water vapour isotope measurements, and as a component in specific analytical setups, such as a crushing line for fluid inclusion isotope analysis in cave deposits (Affolter et al., 2014)."

*12) L. 73: Here it would be very nice to be more precise about the gas stream of a preset background humidity and isotope composition for fluid inclusion analysis. A reader who is unfamiliar with this technique is lost here (see also my minor comment 53).*

We have decided to move part of Section 6 (fluid inclusion analysis) into the introduction to clarify upfront how a precisely regulated mixing ratio is important for the analytical setup during crushing. We also specifically name fluid inclusion analysis as an application example in the introduction, including a reference to Affolter et al., 2014. We hope these revisions more clearly highlight this as a topic from the start of the manuscript.

*13) L. 75-89: This paragraph is very dense with information and a bit difficult to follow. The first sentence starts with introducing the advantage of the microdrop generator in covering "a wider*

*range of applications with CRDS analysers" than with currently available devices. But there are a countable number of applications, and the currently available devices are optimised for the tasks at hand. I rather think the advantage of the microdrop generator is the flexibility of it to produce a wide range of (predetermined) water vapour mixing ratios and isotope compositions. The application to microfluid inclusions in stalagmites is nice but this high flexibility is actually not needed there.*

> Our intention was indeed to point out the flexibility of the device (i.e., that the device covers a range of applications), We have slightly revised this section for clarity, and added a bullet point list that clearly states the instrument specifications.

*14) L. 77: Why are the authors so strict about the water vapour mixing ratio uncertainty? Is this useful in one of their applications? I thought they target the isotope composition of different vapour/liquid/fluid inclusion samples and not the absolute water concentration.*

> The background humidity for fluid inclusion analysis has in previous applications been provided by peristaltic pumps, which induced substantial noise on the analysis (Affoltern et al., 2014). As mentioned in reply to comment #11 above, we have pointed out this aspect more clearly in the introduction as a motivation for the precision of the mixing ratio.

*15) L. 81: I believe the first publications with fast-response analysers with flow rates above 100 sccm are Thurnherr et al. 2020 ACP with a detailed assessment of the impact of the flow rate on the water vapour mixing ratio dependence on isotope measurements in their supplement and Bailey et al. 2023 ESSD. From these publications with flow rates of about 300 sccm through the cavity it becomes clear that flow rate limitations of calibration systems is an important challenge.*

> Thank you for pointing out these references, which we have include in the revision:
>
> "At even larger flow of about 300 sccm through the cavity the flow rate limitations of calibration systems emerge as an important challenge (Thurnherr et al., 2020; Bailey et al., 2023)."

*16) L. 82: This sentence should be removed or moved to the discussion in the conclusions (see also my minor comment 74), the response times of the inlet is not tested in this publication and cannot be tested with the proposed microdrop dispenser because the flow rates are much too small (would need several liters per min to be of interest for this sort of application).*

> It is correct that downstream dilution will be necessary to provide higher flow rates with the system, but also note that some inlets are operated in this range of flow rates (e.g., Chazette et al., 2021). Nonetheless, we have rephrased now stating that inlet evaluation would require downstream dilution, and return to this point in the discussion section:
>
> "An important limitation of the current setup is the maximum flow rate of about 250-500 sccm. Even larger flow rates would be needed characterise the response times of entire inlet lines with additional flush pumps used in semi-permanent installations for water isotope analysis (e.g., Steen Larsen et al., 2013; Bonne et al., 2014; Galewsky et al., 2016). While we have not tested such an application here, is technically fairly straightforward to create higher flow rates by diluting the water vapour stream from the microdrop device with dry carrier gas, for example from a gas tank or a dry air generator. Such characterisation studies are left for future work."

*17) L. 88: The requirement of being field compatible seems ambitious and I don't fully understand it. How well do the dispenser heads work in very cold environments with vibrations and at low*

*pressure? In principle drift correction in the field could be done with simpler existing systems, the advantage of this system resides in the possibility to characterise the water vapour mixing ratio – isotope ratio dependency in detail. This is more a laboratory than a field application.*

We think we have been unclear what we mean by field application. In our understanding, this basically entails that the device is sufficiently small and easy to handle such that it can be packed and shipped without special requirements, and for instance operated on a ship or in some station setup along with the analyzer. This requirement comes from the desire to have one multi-purpose system, which could lower cost compared to requiring several dedicated components. It may also for example be desireable to verify the baseline characteristics a system before shipping it back from a field deployment. There are certain limits to the operating conditions that the system presented here can not operate at without additional modification, for example, operation at sub-freezing temperature levels. Low pressure is no limitation to the dispenser head, as they are equipped with a differential pressure regulation with respect to ambient conditions. We have not done vibration testing, but note that there are no actual mechanical parts in the dispenser head that could for example break or get stuck during shock. Our system is not necessarily more complicated than for example an SDM, and the absence of mechanical parts allows for in principle more extensive operating duration. We also have since submission learned that gas bubbles an important aspect that so far has stopped dispensing after several hours. We can now control this aspect much better by de-gassing our standard liquid before operation, and can achieve continuous operation over several days, rather than hours. We have revised the text here and in the discussion to better explain, motivate and detail the field compatibility requirement from our perspective.

*18) L. 92: SDMs can go well below 6000 ppmv.*

We acknowledge this is possible, but with limitations. Here is an excerpt from the user manual of the Picarro SMD A0101 User's Guide from April 2010, pg. 14:

"The concentration of vapor will be determined by the user programmed liquid flow rate. A rate of 0.02 microliters/second corresponds to approximately 6000 ppmv. The vapor concentration is a linear function of the liquid flow rate. Rates higher than 0.08 microliters/second (24000 ppmv) are prevented by the software in order to prevent accidental saturation of the analyzer. The precision of the isotopic ratio measurement is specified for a vapor concentration of 6000 to 20000 ppmv. The precision will suffer significantly below 6000 ppmv, increasing the measurement duration will compensate to some degree. The dry air source, such as Drierite® condition air with a 200- 300 ppmv water concentration, can contribute significantly to the measured isotope rate when operating at standard vapor concentrations below 6000 ppmv .".

We have clarified in the revision that 6000 ppmv is the lower recommended limit of operation of the SDM:

"For example, the SDM provides a more limited range of humidities (6'000-24'000 ppmv according to manufacturer specifications, even though lower humidities are possible), and at most 20 min of operation before a new cycle is started."

*19) L. 94: What does "Injections with the autosampler are feasible with a specific method" mean?*

We now clarify the intended message of these sentences, namely that it is possible to obtain a range of isotope ratios from liquid injections with some manual work. We have reprased:

"To some degree, autosampler injections can be used to cover a range of isotope composition over different mixing ratios. However, this involves significant manual intervention and preparations, as different mixtures between water standards have to be prepared and analysed beforehand, and injection amounts have to be adjusted to obtain the desired range of water vapour mixing ratios in the vapour stream from the vapouriser module."

*20) L. 95: This joins my major comment M1: unless you are able to very precisely predict the isotope composition of your mixture with the two dispenser heads you cannot really overcome this limitation from other systems.*

See our reply to comment M1.

*21) L. 101: this doesn't become clear from the schematic in Fig. 1.*

Thanks for pointing this out. As also requested by Reviewer 1, we have added drawings of the evaporation chamber to the appendix, which hopefully clarifies the design of the evaporation chamber.

*22) L. 103: this is a very small amount of water which can be rapidly impacted by evaporative enrichment in the field!*

The 12 ml glass vial is a default size, but it is possible to operate the DHs from larger reservoirs which would resemble more to the 45 ml bags used in the SDM. We added a corresponding statement in the discussion:

"Another limitation for field operations is the need to refill the calibration standard reservoirs at regular time intervals. The currently used 12 ml vials hold only about 1/4 of the storage bags of the SDM (45 ml). For longer field deployments, a simple modification would be to use larger glass vials with the same thread size as reservoirs for the DHs."

*23) L. 106: "retention time of water vapour" are interactions with the chamber walls meant here? If yes, to me "memory effect" would be clearer.*

We have rephrased this sentence to include the term memory effect, as also suggested by Reviewer 1.

*24) L. 140: how robust are the voltage and frequency settings and their associated efficiency to prolonged usage?*

According to manufacturer information, once they have been found, the piezo-electric parameters (voltage and peak duration) can be considered as constant for each DH. There are multiple possible combinations of these parameters that result in different drop sizes. This is information is consistent with our experience with the dispensing system. The manufacturer informed us that the piezo crystal may degrade over time if very high voltages are involved, which is however not the case for our operation. We have added this information in the discussion section of the revised draft.

*25) L. 148: "We can used this relation between drop size and mixing ratio to compute the effective drop size..." I am a bit lost here. Do you mean the relation between the frequency and the mixing ratio can be used to estimate the effective drop size?*

Yes, this was a typo, we have corrected the sentence accordingly.

*26) L. 154 and Fig. 3: is it only my print out that make the curves in Fig. 3 not look linear?*

The lines in Fig. 3 are indeed strictly linear. Fig. 3 has been replaced by a new figure.

*27) Fig. 3b: Why is the H2O mixing ratio coverage worse with the settings for panel b than in panel a?*

We have simply run fewer frequencies for this setting, as the focus was on low humidity range. We now show two different sets of sequences of H2O mixing ratio, which span a wider range of data points.

*28) L. 161: This seems to be key, indeed. Can the uncertainty from this fitting procedure be quantified in ppmv for a series of runs? And how frequently does this readjustment of dispensing parameters need to be done?*

We have since submission come to understand that this assessment does not need to be done very frequently, maybe only on a yearly basis, to ensure the DHs have not been ageing more than expected at the voltages we use. Most day-to-day scatter is then due to partial bubble formation, which rather requires empty/fill cycles for their removal. This statement has been removed from the revised manuscript, and bubble formation is now discussed as primary cause of variability throughout the manuscript.

*29) Fig. 4: I am not sure that I really understand the meaning of this figure. What does all dispersion parameters mean? Could this Figure be placed a bit more logically in the discussion. A part from a useful diagnostic, I don't see the use of it? Is it possible that only showing the blue and red distribution would help the reader because these are the selected settings. In general, I am not sure that the reader is interested in the full distribution with all your tested dispersion parameter setting.*

We have combined Fig. 4 with Fig. 3, and only show the distributions of drop sizes for a well-tuned parameter combination. The discussion of this Figure in Sec. 4.2 has therefore been rewriten.

*30) L. 192: here and elsewhere: it is "Allan" variance not "Allen" and the adequate reference would be Allan, 1966 Allan, D. W.: Statistics of atomic frequency standards, Proc. IEEE, 54, 221–230, doi:10.1109/PROC.1966.4634, 1966.*

Thank you for this correction and reference which we included in our revision.

*31) Eq. 1: I think the authors mean yi+1(tau) not yi+1(tau).*

Yes, thank you for this correction.

*32) L. 206: The SDM is optimised for the precision of the δ18O and δ2H, not for the mixing ratio. For a fair comparison with a high quality mixing ratio signal, a dew point generator should be used.*

A comparison to dew point generator signal would be a technical reference for mixing ratio only, but we want to present results from a device that provides both stable water vapour mixing ratios and isotope composition. While it is true that the SDM is not optimized for mixing ratio stability, this is an important aspect for the fluid inclusion application and thus part of the assessment here. We think that this requirement already comes forward better thanks to the rephrased introduction, and made only small additional edits to this section.

*33) L. 225: this is understandable only if the basic concept of fluid inclusion peak measurements on a stable background has been explained in the intro. Maybe a reference to Section 7 could also be added here.*

> A reference to Sec. 7 and Fig. 13 are already included. Due to the changes in the revised introduction and the more stringent narrative about fluid inclusion analysis, we made no further changes here.

*34) L. 254: "The linearity of this mixing is provided over a range of mixing ratios". I can't really evaluate this statement based on Fig. 8a. How about coloring the data points with the mixing ratio?*

> We have implemented this suggestion in the revised version of Fig. 8 and adjusted the text accordingly.

*35) L. 255: "At lower humidity than 5000 ppmv deviations from the linear mixing become apparent" Also here, I can't really see this in Fig. 8b, which shows the RMSE of δ18O and δ2H from the observed mixing line at different water vapour mixing ratios. The RMSE is by the way very large implying that the predicted reference isotope composition of the gas stream has a large uncertainty making it very difficult to me to effectively these mixed samples as reference calibration samples.*

> It appears that Fig. 8 was not as clear as we hoped. We will make an attempt to better support the statements in the text with this Figure. We have found a calculation error in the RMSE for Fig. 8b, and the numbers now make more sense. We adjusted the writing correspondingly.

*36) L. 260: what is Mix000 and Mix 100? This is the first time these labels appear.*

> Mix000 corresponds to a frequency ratio of 0.0, while Mix 100 is a frequency ratio of 1.0. In the revision, this section has been revised and the labels no longer appear.

*37) L. 260: Where can I see this bending down (please add a reference to the figure that is meant here).*

> This section has been revised and that formulation has been removed.

*38) L. 263: the phase space you are showing in Sec. 6 is the correction function in the ln(mixing ratio) vs. δ2H (or δ18O) phase space.*

> We are unfortunately not sure how to interpret this comment. The logarithmic transform in Sec. 6 is used to obtain a more linear display of the dependency characteristics.

*39) L. 265: the microdrop vapour generator was already connected to the analyser for all the evaluations presented in the previous section. Here you switch from characterising the microdrop vapour generator to using it to characterise CRDS analysers in their water vapour mixing ratio and isotope composition dependency.*

> The intention here was to contrast two applications, one where the microdrop vapour generator is directly connected to the CRDS, and one where there are additional applicances in between. Apparently this statement can be misunderstood and we have rephrased in the revision:

"We now present a first application example for the use of the microdrop vapour generator. In this application example, the device is directly connected to the input port of the analyser with an open split. We then..."

*40) L. 267: The water vapour mixing ratio and isotope composition dependency is of spectroscopic origin and due to uncertainties associated with the baseline. Even if a system is optimised for measurements at lower humidity levels than 20'000 ppmv this problem occurs.*

This is true. What we wanted to point out here is that Picarro analyzers are in fact already corrected for baseline effects in the factory with a target humidity of abour 20'000 ppmv, and the remaining dependency is thus a residual of this factory baseline correction, but have now simplified and rephrased for clarity.

*41) L. 273: This is misleading. The spectroscopic origin of the concentration dependency has been mentioned already in early laser spectrometric instrument evaluation work such as from Sturm and Knohl, 2010 and Iannone et al. 2009.*

We think this might be a misunderstanding, it was not our intention to mislead the reader. What we mean to say is that the additional dependency of the previously identified mixing-ratio dependency (e.g., Sturm and Knohl, 2010, Iannone et al., 2009) was shown to be in addition dependent on isotope ratio by Weng et al., 2020. We have rephrased to avoid potential misunderstandings.

*42) L. 274: matrix gas -> carrier gas*

ok

*43) L. 296-308: This is very difficult to follow, could the respective sequences be clearly indicated in Fig. 9? This paragraph should be in stronger dialogue with what we see in Fig. 9 for the reader to be able to follow.*

We have added labels to Fig. 9 and rewrote the paragraph to better describe the calibration sequence displayed in Fig. 9.

*44) L. 313: Here I got lost, where do I have to look above or below?*

We have simplified the writing, removed some details, and rephrased this paragraph.

*45) L. 315: Can the authors be more precise about the number of outliers they filtered out? This is an important quality measure for the microdrop generator. Because an ideal calibration system needs as little instrument measurement time as possible so minimising the occurrence of "outliers" is a key aim when designing a robust calibration system.*

The number of outlier runs was initally relatively large, but reduced in the process by preventing for example leaks. As we have by now better ways to de-gas the standard waters, there need for outlier removal should be reduced substantially.

*46) L. 320: What is the share of new data points from the microdrop generator vs. the data points from Weng et al. (2020) in this analysis shown in Fig. 10? Could the data points from the microdrop generator be highlighted in a different color or shape?*

About two thirds of the data points are added from the microdrop device, which were reduced by 50% when restricted to the highest data quality. We used about an equal

number of data points from the microdrop device and Weng et al., 2020 in construction the correction function for analyzer HIDS2254. We are now highlighting the data points by Weng et al., 2020, in the revised Figure 10 in red (many points are obscured by overlapping blue points and the fitted surface).

*47) L. 331: the delta value is specified and used in equations 2 and 3.*

corrected.

*48) L. 336: this final step was a bit obscure to me. I thought that the correction function is 0 per definition at the reference mixing ratio level?*

The 2D polynomial fit is not constrained to zero at the reference level and thus result in an offset, which we correct for by this additional final step.

*49) Table 3: the RMSE is most likely very dependent on the mixing ratio? Maybe a Figure showing this dependence as a function of mixing ratio would be more informative.*

We now include additional lines in Table 3 with the RMSE values for 3 sub-sets of the mixing ratio. We also corrected an error in the calculation of the RMSE values.

*50) L. 350-355: this paragraph was again not clear to me. Which subsample of available measurements?*

This was a Monte-Carlo approach, more specifically a bootstrap resampling, where we draw a non-unique sample from the original dataset. We rephrased for clarity:
"Here, we used a Monte-Carlo approach to determine the standard deviation of the correction values. Thereby, a bootstrap resampling with 50 repetitions was used to draw non-unique samples from the entire dataset."

*51) Fig. 11: The vertical axis is not the raw but the reference δ2H and δ18O, right? There are no dashed grey lines (at least not in my print out) and what are the blue lines? Would this figure be better readable if it was shown in this format in the appendix (to illustrate the limited concentration effects at high water vapour mixing ratios) and with a zoom in to 0-5000 ppmv (which is the interesting part) in the paper? Right now one does not see much about the important effects at low water vapour mixing ratios.*

Vertical axis is indeed the raw delta values, we will correct the figure. Blue lines showed contour levels for the correction by Weng et al., 2020. We now plot the data points from Weng et al., 2020 as dashed lines, and update the caption accordingly. We are not sure how useful it is to show a zoom on specific details of a particular analyzer, and thus decided to keep the range of the mixing ratios as it is. We also noted that the Weng et al. (2020) correction was plottet reversed, this has now been corrected.

*52) L. 357: "a constant vapour stream" with known water vapour mixing ratio and isotope composition (very important!)*

We have added this important information specification to the sentence.

*53) L. 357-366: This information should already be provided in the introduction.*

We have moved this paragraph to the Introduction.

*54) L. 369: "memory effects can be removed entirely" is misleading and a bit optimistic. Isn't the advantage of the CRDS technique mainly the ability to measure δ2H and δ18O (possibly even d17O) quasi-simultaneously with much reduced sample preparation effort?*

> We moderated and rephrased this statement. Furthermore, the reference to Affolter et al., 2014, was incorrect here, it should have been a reference to de Graaf et al., 2021, who directly address memory effects. This has been corrected in the revised manuscript.

*55) L. 372: "higher throughput and better reproducibility" than what?*

> This statement refers to the referred studies, which compared fluid inclusion analysis with a moist background to methods with a dry background and different IRMS methods. This has been added in the revision.

*56) L. 384: Fig. A2 -> Fig. 12?*

> Corrected.

*57) L. 387-393: connect this paragraph better to Fig. 13, they should be in better text-illustration dialogue to help the reader follow.*

> We have rephrased this section to integrate better with Fig. 13 (now Fig. 15).

*58) L. 395: As mentioned in M2 I wonder about the necessity of this step given the high mixing ratio of the background air stream.*

> See our reply to comment M2, the background air stream may only be raised to a lower level, for example 5000 ppmv, depending on the amount of water vapour released from a sample.

*59) L. 397: "Mixing ratio -isotope ratio corrections..." this has already been mentioned in point i) just above and appears as an unnecessary repetition here.*

> We have revised for brevity.

*60) L. 399: what is meant by "size-δ" relationship?*

> This refers to a relation between the amount of liquid injected and the delta value during calibration. We have rephrased this expression.

*61) L. 405-414: I got lost in this paragraph. As the authors write at L. 410 this appears redundant. Fig. A1 is not very convincing in showing a consistent dependency of the isotope signals on the injected amount.*

> This section relies on the analytical methods published earlier by Affolter et al., (2014). We have rephrased this section, since it caused misunderstanding. The key point is that it is important here to maintain the principle of identical treatment of sample and standards. In fact, an average of the injections across the entire range of humidities has been used. We now show the overall average for each standard in the revised figure.

*62) Fig. A1: To which mixing ratio levels do these injection volumes correspond?*

> Each injected volume creates a peak on top of the background humidity, so these numbers cannot be translated into one specific mixing ratio.

*63) L. 407: "drift in values", what do you mean by values?*

This refers to a drift in the background, we have rephrased for clarity.

*64) L. 412: the last sentence of this paragraph "Finally,..." is particularly obscure to me. Could the authors illustrate this with a figure?*

We think this can be addressed by rephrasing the sentence, which was not clear in the previous draft. The sentence now reads:

"Finally, the amount of water released by the samples was obtained from a transfer function, that had been constructed previously from a series of injections of known amount of standard water. "

*65) L. 431: Here I am a bit puzzled: the authors write that water vapour mixing ratio – isotope dependency corrections are necessary below 5000-10'000 ppmv (L. 270). But the background gas stream can be designed in such a way that the mixing ratio is larger than this and ideally even close to the optimal water vapour mixing ratio operation level of 20'000 ppmv? So why does this correction matter? Can the authors show how different the isotope signal estimates get when applying vs. when not applying the water vapour mixing ratio correction for their aliquot or the samples used to mimic real fluid inclusions? This is also my major point M2.*

See our reply to comment M2 above, which also applies to this comment.

*66) L. 432: "more correct" than what?*

This sentence has been rephrased:

"First, the characterisation of the analyser allows to correct for the mixing ratio -- isotope ratio dependency of the analyser signal, providing a more accurate integrated signal of each sample than without this correction."

*67) L. 434: Why is the precision of the water vapour mixing ratio mentioned here? Isn't what matters for precise fluid inclusion measurements the precision of the isotope signals?*

As explained in our reply to M2, the stability of the mixing ratio is important for this application to separate the background water vapour clearly from the water vapour released from the crushed sample. In Sec. 7, we now stress that since the background is subtracted from the sample peak, large variations in the background (mixing ratio and isotopes) are propagated directly into the measured values. de Graaf et al. (2020) argue that even minor instabilities in the background can lead to a loss in measurement precision. Another problem with the peristaltic pumps is drift of the background during measurements (Weissbach et al., 2023, their Fig. 2).

*68) L. 455: for which application would horizontally mounted DHs be an interesting option?*

Due to the design of the evaporation chamber, the DHs are mounted horizontally here. We have rephrased this sentence for clarity:

"This in particulary important for our current design with horizontally mounted DHs, where gravitational forces are not aligned with the dispenser axis."

*69) L. 462: Flow rates between 300 and 600 sccm through the cavity are now used in different applications (aircraft-based and flux measurements) and present many advantages (e.g. faster response times, better signal to noise ratio at shorter averaging times) but this range of flow rates cannot be covered the microdrop dispenser system. Why is that so, and is an extension to these higher flow rates possible with the given operational range of dispenser head triggering frequency?*

> The current limitation to 500 sccm for the device presented here originates mainly from the specification of the gas flow regulator. It is possible to obtain higher flow rates with the present device, but we did not have the necessary pumps available to either produce such high cavity flow rates, nor a suitable gas flow regulator with a higher maximum flow rate. Higher flow will require higher dispensing frequencies, and/or dispenser heads that can produce larger drops, which is a straighforward adjustment. In addition, the tubing diameter to provide the gas flow may have to be modified to prevent buildup of overpressure in the evaporation chamber, and the gas heating may have to be modified to provide sufficiently stable carrier gas temperatures. All these aspects would be modifications that then require extensive testing, and exceed the scope of the current study. We have included these details in the discussion section and pick up on these future development options in the implications.

*70) L. 475: remind the reader that the Picarro SDM is the commercially available benchmark used for characterisation of the microdrop dispenser.*

> This has been added in the revision.

*71) L. 475: In the long-term precision (relevant for calibration) the microdrop dispenser is not better than the SDM for the isotope signals (which we are interested in). The dispenser is better than the SDM for water vapour mixing ratio, for which however a dew point generator is needed anyway for precise calibration. Together with the many small problems that occur with air bubbles in the dispenser head, the small amount of liquid that needs to be exchanged regularly, this questions the utility of this system for regular field calibrations. It is very useful for an in-depth characterisation of the water vapour mixing ratio – isotope signal dependency in the lab but apparently not so much for prolonged measurements in the field.*

> We can partly understand this assessment of the reviewer based on the presented information. However, since submission of the manuscript, we can report on a much longer uninterrupted operation time of the device thanks to de-gassing of the calibration liquid before use. After more effectively de-gassing water standards before use in the dispenser heads, we find much better precision than for example the SDM in all characteristic parameters (Fig. 5). This de-gassing is now an aspect that we bring up in Sec. 4.3 and in the Discussions.

> As discussed before, there are different kinds of field operations. By field measurement capability, we do not primarily mean that the device is feasible for unsupervised long-term deployments. From our perspective, this mainly means that the device does not need a temperature controlled environment for operation, is portable, and robust enough to set up in a reasonable time at a different location, very similar to the Picarro analyzer itself. We state now more clearly what we understand by field operations in the specifications, and that there is currently still a need to supervise the device on a regular basis.

*72) L. 477: I don't understand what this sentence implies scientifically. Which specifications are meant here? The dispenser head and mass flow controller specifications or the objectives set by the authors for the system in Section 2? Maybe a reference to section 2 could be added here.*

Yes, we have added a reference to Section 2 here.

*73) L. 495: Could this simply be a numbered list of implications? It would be easier to read.*

We have considered reorganising as suggested, but instead rephrased the entire paragraph. Thereby, we used inline numbering to make the different implications easier to differentiate.

*74) L. 502: Here is the place where the sentence from L. 82 could be brought in: larger flow rates are needed to test the inlet system and the overall response time of different water vapour isotope measurement field setups.*

Thank you for this helpful suggestion, this paragraph has been shortened and revised to include a statement to this effect.

*75) L. 502: Again this is misleading, response times of laser spectrometric water vapour isotope measurements were not the topic of this publication. If one just reads the conclusions, this sounds like a result from this study. References to the studies Sturm and Knohl, 2010 AMT (tested Synflex tubings in the lab) and Tremoy et al. 2014 JGR (tested Synflex tubings in the field) should be made.*

In the revision, we have made clearer that this paragraph describes implications from our study, rather than conclusions. The reference to Tremoy et al., 2011 went missing during the editing, it was certainly not our intention to mislead the reader, see also reply to Reviewer 1. As suggested, we have also added the mentioned reference to Sturm and Knohl (2009) here in the revised draft.

*Technical points:*

*1) L. 17: . Currently*
*2) L. 67: technology, which allows*
*3) L. 75: "water" vapour*
*4) L. 76-80: Grammatically and for keeping the reader's attention having (ii) in the same sentence as (i) would be much more convenient.*
*5) L. 103-104: "one 12 ml glass vial that holds a liquid water standard and which is mounted next to ...". It is not the evaporation chamber that holds the liquid water standard, right?*
*6) L. 161: demonstrates*
*7) L. 161: "... a suitable dispensing parameters"*
*8) L. 214: microdrop*
*9) Caption Fig. 4: dispening -> dispersion*
*10) L. 348: og -> of*
*11) L. 369: CDRS -> CRDS*
*12) L. 464: viscousity*
*13) L. 509: performed*
*14) P. 27, Fig. A1: is "assigned" really the term you want to use here on the y-axis? To me this seems to be the "reference".*

We implemented all technical changes as suggested.

**Additional references**

Jones, T. R., White, J. W. C., Steig, E. J., Vaughn, B. H., Morris, V., Gkinis, V., Markle, B. R., and Schoenemann, S. W.: Improved methodologies for continuous-flow analysis of stable water isotopes in ice cores, Atmos. Meas. Tech., 10, 617–632, https://doi.org/10.5194/amt-10-617-2017, 2017.

de Graaf, S, Vonhof, HB, Levy, EJ, Markowska, M, Haug, GH. Isotope ratio infrared spectroscopy analysis of water samples without memory effects. Rapid Commun Mass Spectrom. 2021; 35:e9055. https://doi.org/10.1002/rcm.9055.

Weissbach, T., Kluge, T., Affolter, S., Leuenberger, M. C., Vonhof, H., Riechelmann, D. F. C., Fohlmeister, J., Juhl, M.-C., Hemmer, B., Wu, Y., Warken, S. F., Schmidt, M., Frank, N., Aeschbach, W., Constraints for precise and accurate fluid inclusion stable isotope analysis using water-vapour saturated CRDS techniques, Chemical Geology, 617, 2023, 121268, https://doi.org/10.1016/j.chemgeo.2022.121268.

Fernandez, A., Løland, M. H., Maccali, J., Krüger, Y., Vonhof, H. B., Sodemann, H., and Meckler, A. N.: Characterization and Correction of Evaporative Artifacts in Speleothem Fluid Inclusion Isotope Analyses as Applied to a Stalagmite From Borneo, Geochemistry, Geophysics, Geosystems, 24, e2023GC010857, https://doi.org/10.1029/2023GC010857, 2023.

Maccali, J., Meckler, A. N., Lauritzen, S. E., Brekken, T., Rokkan, H. A., Fernandez, A., Krüger, Y., Adigun, J., Affolter, S., and Leuenberger, M.: Multi-proxy speleothem-based reconstruction of mid-MIS 3 climate in South Africa, Clim. Past Discuss., 2023, 1-27, 10.5194/cp-2023-1, 2023.